# Distinct autoreactive CD19⁻ plasma cell subsets accumulate in lupus-prone mice

Van Duc Dang [1,2,3,15], Franziska Szelinski [1,2,15], Elodie Mohr [1], Tuan Anh Le [1,2], Jacob Ritter [1,2], Annika Wiedemann [1,2], Marta Ferreira-Gomes [1], Gabriela Maria Guerra[1], Pawel Durek [1], Frederik Heinrich [1], Hector Rincon-Arevalo [1,2,4,5], Ana-Luisa Stefanski [1,2], Eva Schrezenmeier[1,4,6], Van T. Hoang[7], Hong-Nhung Dao[7], Soeren Ocvirk[8], Qingyu Cheng[1,2], Falk Hiepe [1,2], Christian Hipfl[9], Sebastian Hardt[9], Max Löhning [1,2], Liem Thanh Nguyen[7,10], Mir-Farzin Mashreghi [1,11], Simon Fillatreau [12,13,14], Thomas Dörner [1,2] & Andreia C. Lino [1,2] ✉

Plasma cells (PC) participate in the pathogenesis of systemic lupus erythematosus (SLE) through sustained autoantibody and inflammatory cytokine secretion. Current PC-depleting therapies risk eliminating protective long-lived PCs, highlighting the need to identify pathogenic subsets for selective targeting. Here, using single-cell RNA sequencing, B cell receptor repertoire analysis, and genetic models, we identify disease- and organ-specific PCs in lupus-prone mice. We find a substantial expansion of autoreactive CD19⁻ PCs, particularly class-switched CXCR3⁺ and phosphatidylcholine-specific B-1−derived subsets, which exhibit unique gene expression profiles. We show that CD19⁻ PCs originate from CD19⁺ PCs in a unidirectional manner. Peripheral blood from SLE patients shows elevated frequencies of CD19⁻ PCs, implicating these cells in sustaining pathogenic activity. Our findings highlight the emergence of autoreactive CD19⁻ PCs as a critical feature of lupus pathogenesis in mice and underscore the need for therapeutic approaches that extend beyond CD19-targeting to improve treatment strategies in SLE.

As a result of the secretion of autoantibodies and inflammatory cytokines, which mediate tissue-specific and/or systemic inflammation, antibody-secreting cells (ASC) or plasmocytes (PC) containing plasmablasts and plasma cells play a pathogenic role in autoimmune diseases (AD)[1–3]. Although several approaches to eliminate PCs currently exist[3], eliminating all PCs can lead to a loss of protective memory long-lived plasmocytes (LLPC) that reside in the bone marrow (BM)[4–6]. Moreover, PCs contain interleukin (IL)−10- and IL-35-producing cells that exert regulatory functions[3,7–9]. Additionally, PCs contribute to tissue homeostasis, for instance in myelopoiesis in aged BM[10].

Several PC subsets with relevant functions have been recently described in mice. For instance, we found that PCs expressing immune checkpoint receptor LAG-3 are a subset of PCs with regulatory function

in vivo[8]. By single-cell mRNA sequencing combined with cytometry and a genetic pulse-chase mouse model, IgG and IgM LLPCs were found to display CD326[hi] (EpCAM) CXCR3⁻ while IgA LLPCs were Ly6A[hi]Tigit[-11]. Recently, a study using genetic time-stamping approach and RNA sequencing reported that LLPCs in BM increase the expression of Fcεr1γ and IL-13rα1 and decrease MHC-II and SLAMF6 expression[12]. In the context of antibody-mediated ADs, CXCR3⁺ PCs were found in the BM, spleen and inflamed kidneys of mice with lupus[3]. Using an unbiased, large-scale protein screening, we identified a BM-specific PC subset in lupus as CD39[hi]CD326[hi] cells[13].

Several PC subpopulations have been described in humans. For example, CXCR3⁺ PCs were found in chronically inflamed joints and the blood of rheumatoid arthritis (RA) patients[14], accumulate at a

higher frequency in the peripheral blood and lamina propria of patients with ulcerative colitis[15]. Additionally, a subset of human PCs that lack canonical B cell marker CD19 (CD19⁻ PC) is enriched in human BM, while almost absent from tonsils, spleen, and blood. Yet, CD19⁻ PCs are also detected in chronically inflamed tissues in ADs[16,17]. Furthermore, both CD19⁺ and CD19⁻ autoreactive PCs against dsDNA are readily detectable in BM[17]. This observation is also applying to vaccinia-specific LLPCs[18]. We and others used single-cell RNA profiling to study PC subpopulations in healthy human BM and revealed several PC subsets, some of which were associated with CD19⁻ PC subpopulation[19,20].

Here, we show that CD19⁻ PCs accumulate with age in the mouse BM. CD19⁻ bone marrow plasma cells (BMPC) derive from various B cell subsets upon T-dependent and -independent reactions. Next, we combine single-cell RNA sequencing (scRNA-seq), cite cellular indexing of transcriptomes and epitopes by sequencing (CITE-seq) and B cell receptor (BCR) repertoire analysis to investigate PC heterogeneity in BM, spleen, and kidney of mice prior and post systemic lupus erythematosus (SLE) manifestation. Using this approach, we uncover several disease- and organ-specific PCs, including two PCs subsets, both CD19⁻ PCs that are clearly associated to SLE. First, a subset of IgG PCs, expressing the chemokine receptor CXCR3, that accumulates in BM, spleen, and kidney of sick SLE mice. Second, phosphatidylcholine-specific CD19⁻ B-1-derived PCs that accumulate in BM of sick SLE mice. Notably, we show that in vivo CD19⁺ PCs generate CD19⁻ PCs, while the reverse does not occur. Notably, we report that SLE patients' peripheral blood is enriched in CD19⁻ PCs, suggesting that the presence of an established pool of CD19⁻ PCs might compromise the efficacy of treatment targeting CD19, such as in CD19-directed chimeric antigen receptor (CAR)-T cells. Therefore, we reason that the timing of therapeutic intervention is crucial when considering the use of anti-CD19 therapy in lupus. In addition, our results highlight the possibility to develop therapies that specifically target pathogenic PC subsets in lupus.

## Results
### CD19⁻ PCs are enriched in the BM of human and mouse and accumulate with age in mice
We and others identified the CD19⁻ PC subpopulation to be enriched in human BM (Supplementary Fig. 1A)[16–18]. In the meantime, CD19-directed CAR-T cell therapy has proven successful for the treatment of SLE, with titers of protective antibodies remaining stable, while the titers of autoantibodies decreased in most of the treated patients[21–26]. A possible explanation for this observation is that PCs producing autoantibodies are CD19⁺, while those producing protective antibodies are CD19⁻. Comprehensive data that would allow understanding the implications of CD19⁺ and CD19⁻ PCs in autoimmunity and CD19-directed CAR-T cell therapy are not available and hard to obtain in humans. Thus, we evaluated whether the mouse model would be appropriate to study CD19⁻ PCs in detail. First, we analyzed the distribution of CD19⁻ PCs in various mouse lymphoid organs. CD19⁻ PCs were enriched in BM and lamina propria (LP) of 3–6-month-old C57BL/6 mice in comparison with other lymphoid organs such as the spleen, mesenteric lymph nodes (mLN) and Peyer's patches (PP), where they were also present, albeit at lower frequencies (Fig. 1A and Supplementary Fig. 1B). Intriguingly, we found similar frequencies and absolute numbers of CD19⁻ PCs in specific-pathogen-free (SPF) and in germ-free (GF) mice, indicating that these PCs could be triggered by self- or/and food-derived antigens (Supplementary Fig. 1C).

CD19⁻ and CD19⁺ BMPCs expressed high amounts of BLIMP1 (Fig. 1B), that governs PC differentiation and maintenance[27,28], confirming that CD19⁻ were bona fide plasma cells, which was confirmed by their ability to secrete IgM, IgA or IgG antibodies (Fig. 1C and Supplementary Fig. 1D, E). CD19⁻ BMPCs expressed the proliferation

marker Ki-67 to a lesser extent than their CD19⁺ counterparts (Fig. 1D), as well as lower levels of B220 and MHC-II (Fig. 1E) which have been shown to be downregulated during ASC maturation[29–31]. Furthermore, CD19⁻ PCs accumulated with age (Fig. 1F). These results suggested that CD19⁻ PCs are more mature than their CD19⁺ counterparts.

### CD19⁻ BMPCs derive from various B cell subsets upon T-dependent and -independent reactions
The variety of immunoglobulin (Ig) classes produced by CD19⁻ PCs (Fig. 1C) suggested that they could originate from various B cell subsets and be generated along different differentiation pathways.

First, we analyzed the expression of VH11/Vk14 and VH12 BCRs, whose expression is restricted to B-1a cells[8,32], by CD19⁻ and CD19⁺ BMPCs. Approximately 9% of the CD19⁻BMPCs expressed VH11 and Vk14 (Fig. 2A and Supplementary Fig. 2A). Interestingly, no BMPCs expressed VH12 (Fig. 2A).

In a complementary approach, we employed a genetic fate mapping system *Cd21*-cre-ROSA26-STOP-eYFP reporter mouse. In this system, cells expressing CD21 at any time point become eYFP⁺. This approach was based on the observation that B-1a cells exhibit low CD21 expression compared to marginal zone (MZ) and follicular (FO) B cells from the spleen, as well as B-2 and B-1b cells from the peritoneal cavity[8,33]. We found that nearly all MZ and FO from spleen, B-2 and B-1b from the peritoneal cavity were YFP⁺, while most B-1 cells from BM and spleen, as well as B-1a and VH11⁺Vk14⁺ B-1a cells from the peritoneal cavity were YFP⁻. Within the CD19⁻ BMPC population, approximately 80% were eYFP⁺, suggesting that subsets beyond B-1a cells, including MZ, FO, B-2, and B-1b cells contribute to the generation of CD19⁻ BMPCs (Fig. 2B and Supplementary Fig. 2B, C).

Next, we used another genetic fate mapping system—Cγ1-cre-ROSA26-STOP-eYFP reporter mice. In these mice, cells that express Ig gamma1, usually B cells undergoing germinal center (GC) reactions upon immunization with a T cell-dependent antigen, become YFP⁺[34]. Here, we found that older naïve Cγ1-cre-ROSA26-STOP-eYFP mice accumulate YFP⁺ cells within CD19⁻ BMPCs, while more YFP⁺ cells were found in the CD19⁺ counterpart in younger one-month-old mice (Fig. 2C), suggesting that CD19⁻ PCs partly derived from GC and may mature from GC-derived CD19⁺ PCs. In agreement, we found that T cell-deficient mice (*Tcrβδ⁻/⁻*) had reduced frequency and absolute number of CD19⁻ BMPCs compared to wild type (WT) mice (Fig. 2D and Supplementary Fig. 2D).

To further demonstrate that CD19⁻ BMPCs could represent GC-derived LLPCs, we immunized WT mice with 4-hydroxy-3-nitrophenylacetyl keyhole limpet hemocyanin (NP-KLH), a classical T cell-dependent antigen, pulsed with 5-bromo-2′-deoxyuridine (BrdU) at days 7 and 8 post-immunization (Fig. 2E) and analyzed them at days 7, 28 and 90 after immunization. We found that the frequency of CD19⁻ cells among NP-specific PCs increased progressively over time, rising from ~40% on day 7 to ~80% by day 90 post-immunization (Fig. 2F). NP-specific PCs are known to be Igλ[35–38]. Thus, we confirmed by flow cytometric analysis of BMPCs following NP-KLH immunization that the NP-specific PCs were Igλ⁺ (Supplementary Fig. 2E). Furthermore, 80% of the NP⁺BrdU⁺ PCs in the BM were CD19⁻ at day 90 after immunization (Fig. 2G and Supplementary Fig. 2F). Furthermore, at day 90, CD19⁻ NP-specific PCs were mainly B220⁻ cells (Fig. 2H), a hallmark of LLPCs. Additionally, the majority of NP⁺IgG⁺ PCs were CD19⁻ (Fig. 2I).

In summary, CD19⁻ BMPCs are LLPCs generated from various B cell subsets, including B-1 cells, in T-dependent and T-independent manners.

### CD19⁻ PCs accumulate in lupus
Recently, CD19-directed CAR-T cells have been shown to be efficacious and safe in the treatment of severe refractory SLE. Remission correlated with the loss of SLE-associated autoantibodies[21–24]. Few individuals did not show a reduction in some SLE-associated

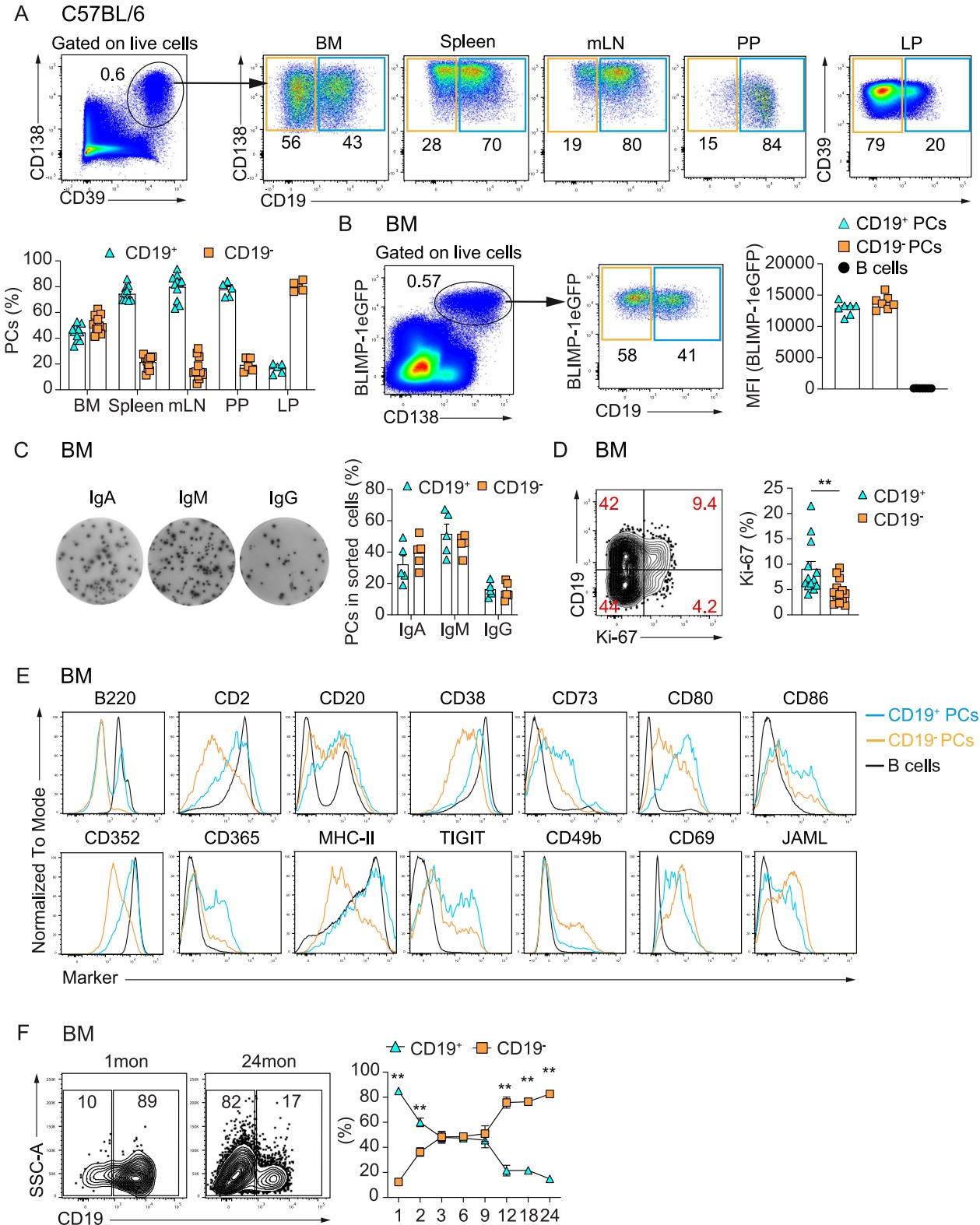

autoantibodies[23,25,26], suggesting that pathogenic PCs in these patients might not express CD19.

To gain insights into the implication of CD19+ and CD19− PCs into lupus pathology, we analyzed peripheral blood PCs from patients with SLE and healthy donors. In agreement with a recent report[39], SLE patients exhibited increased frequencies of both CD19+ and CD19− PCs compared to healthy donors (Fig. 3A). Next, we analyzed B6.Sle123 (Sle123) lupus prone triple congenic mice. These mice spontaneously

develop lupus, make autoantibodies, and suffer from immune complex-mediated glomerulonephritis[40–42]. The disease develops slowly, allowing us to analyze two months old healthy (proteinuria negative) Sle123 and six- to seven-months-old diseased (proteinuria >100 mg/dL) Sle123 mice. We found a striking increase in the frequencies and numbers of CD19− PCs in sick compared to healthy mice (Fig. 3B and Supplementary Fig. 3A). Although CD19− frequencies increase with age (Fig. 1F), it was notable that sick Sle123 mice had

**Fig. 1 | CD19⁻ PCs are enriched in the BM of human and mouse and accumulate with age in mice. A** Representative FACS plots show a CD19⁻ PC subset in mouse bone marrow (BM), spleen, mesenteric lymph nodes (mLN), Peyer's patches (PP) and lamina propria (LP). Cells were gated on total PCs (CD39⁺CD138⁺ or CD39⁺CD130⁺)[13] as in Supplementary Fig. 1B. Graphs show the frequencies of CD19⁺ and CD19⁻ PCs. **B** Representative FACS plots show BLIMP-1eGFP and CD19 expression by BMPCs. Graph shows BLIMP-1eGFP MFI values in CD19⁺(CD19⁺CD138⁺BLIMP-1eGFP⁺), CD19⁻(CD19⁻CD138⁺BLIMP-1eGFP⁺) BMPCs and B cells (CD19⁺CD138⁻BLIMP-1eGFP⁺). **C** Frequencies of IgA-, IgM- and IgG-secreting cells among sorted CD19⁺ and CD19⁻ BMPCs (gating in Supplementary Fig. 1D) by ELISPOT. Each point represents one experiment. **D** Representative FACS plot shows CD19 and Ki-67 expression in BMPCs. Cells were gated on total PCs (CD39⁺CD138⁺) as in (**A**). Graph shows frequency of Ki-67⁺ cells within CD19⁺ and CD19⁻ BMPCs.

Two-tailed Mann–Whitney test (**p = 0.0042). **E** Histograms show differential expression of surface molecules by CD19⁺, CD19⁻ BMPCs, and B cells determined using LEGEND screening of 255 surface proteins. Cells were gated on total PCs (CD39⁺CD138⁺) as in (**A**). B cells were gated as CD19⁺CD39⁻CD138⁻. **F** Representative FACS plots show frequencies of CD19⁺ versus CD19⁻ BMPCs within total PCs from C57BL/6 at 1 and 24 months. Graph shows CD19⁺ and CD19⁻ PCs frequencies in BM across ages. Cells were gated on total PCs (CD138⁺CD39⁺). Two-tailed Mann–Whitney test was used (**p = 0.0022 (1, 12, 24 mo), **p = 0.0043 (2 mo), **p = 0.0079 (18 mo)). P values ≥ 0.05 not shown. Experiments represent naive C57BL/6 (**A**, **C**–**F**) and prdm1eGFP mice (**B**) at 3–6 months (**A**–**E**) and indicated ages (**F**). Data combine three (**A**, **B**, **D**, **F**), five (**C**), and two (**E**) independent experiments; n = 11 (BM, spleen, mLN) and 5 (PP, LP) (**A**); n = 7 (**B**); n = 15 (**C**); n = 13 (**D**); n = 6 (1, 2, 12, 24 mo), 7 (3, 9 mo), 15 (6 mo), and 5 (18 mo) (**F**). Data show mean ± SEM.

---

increased frequencies of CD19⁻ PCs compared to age-matched WT, while we found similar frequencies of these cells in healthy *Sle123* mice compared to two months old age-matched WT (Fig. 3C).

Next, we used the *Sle123* mouse model to study heterogeneity of SLE PCs with an unbiased approach. We isolated total PCs from BM and spleen of healthy and sick *Sle123 mice* for single-cell transcriptomic profiling (Fig. 3D and Supplementary Fig. 3B). In addition, we purified PCs from kidneys of sick mice, as non-lymphoid tissue and the organ affected by SLE pathogenesis. No PCs were present in the kidney of healthy mice[13,43,44]. We obtained 6539 (BM healthy), 8607 (BM sick), 10,982 (spleen healthy), 9133 (spleen sick) and 4776 (kidney sick) cells that after data filtering (Supplementary Fig. 3C–H), distributed in 14 clusters (Fig. 3E). The 14 clusters were identified as bona fide PCs based on the expression of *Prdm1, Irf4, Xbp1, Sdc1, Entpd1,* and *Cd81* (Supplementary Fig. 3I)[13,28,45]. Regarding CD19 RNA and protein expression by SLE-associated PC clusters from *Sle123* mice, we found that Clusters 0, 1, 2, 3, 5, 10 and 11 expressed low amounts of CD19 (Fig. 4A and Supplementary Fig. 4A). Of note, and in agreement with the flow cytometry data (Fig. 3B), we found overall increased percentages of CD19^low PCs in the analyzed organs from sick compared to healthy mice (Fig. 4B and Supplementary Fig. 4B).

To study whether the downmodulation of CD19 in PC subsets was part of a transcriptionally regulated program, we analyzed the expression of genes that have been shown to transcriptionally activate or repress CD19 expression in B cells in genome-wide CRISPR-Cas9 screening aiming at identifying modulators of CD19 expression[46]. We found that CD19^low/− PC subsets expressed low levels of CD19 activators, while the CD19⁺ PC subsets expressed high levels of these molecules (Fig. 4C).

## CD19⁻ PCs are derived from CD19⁺ PCs

Next, using expression of G2M.Score, S.Score, Ki-67 and other transcription factor characteristics of PC differentiation (Supplementary Fig. 5A), we identified clusters 6 and 9 as early PCs/ plasmablasts that are still dividing by contrast with fully mature or LLPCs which do not proliferate[5,47,48]. Thus, clusters 6 and 9 were selected as starting points to construct maturation trajectories, using Monocle3[49]. This approach predicted a maturation trajectory for SLE PCs in which 2, 5, 7, 8, and 12 were predicted as early clusters and 0, 1, 3, 4, 10, and 11 as late clusters (Fig. 5A). We observed that the expression of *Cd19* by PC clusters decreased consistently with the predicted maturation trajectory (Fig. 5A).

Our observation that the CD19⁻ PC subpopulation accumulates with age (Fig. 1F), during an immune response (Fig. 2F) and in lupus mice, prompted us to examine whether CD19⁻ PCs are generated from CD19⁺ PCs, and/or vice-versa. Therefore, we isolated CD19⁺ and CD19⁻PCs from *Sle123* mice, transferred them into *Rag⁻/⁻* mice and followed their development (Fig. 5B). One week after transfer, we found similar frequencies and absolute numbers of total PCs in BM and spleen of mice that received CD19⁺ and CD19⁻ PC (Fig. 5C). Donor cells survived and produced detectable amounts of antibodies in the

recipients (Supplementary Fig. 5B). Remarkably, we found that half of PCs in CD19⁺ PC recipients were CD19⁻ PCs in both BM and spleen, while nearly all PCs in CD19⁻ PC recipients remained CD19⁻ PCs (Fig. 5D). This data supports that CD19⁺ PCs generate CD19⁻ PCs, while CD19⁻ cannot give rise to CD19⁺ cells.

## Single-cell transcriptomic profiling reveals disease- and organ-specific PCs in SLE mice

To further functionally characterize CD19⁻ PCs in the context of lupus, we sorted CD19⁻ and CD19⁺ PCs from sick *Sle123* mice and quantified anti-dsDNA IgM and IgG antibodies using an ELISPOT assay. Notably, we observed increased numbers of dsDNA-specific IgM- and IgG-secreting cells in the CD19⁻ PC fraction compared to CD19⁺ PCs (Fig. 6A).

Single-cell transcriptomic analysis allowed an unbiased approach by which we observed that specific PC clusters were expanded in diseased mice in specific organs (Fig. 3E). In addition to transcripts, we used CITE-seq and analyzed expression of proteins such as LAG-3 (CD223), CXCR3 (CD183), and CD39, as well as CD326, since these molecules identify PC subsets with regulatory functions[8], formed in IFN-γ-rich inflammatory environments[50,51], and that have been associated with SLE[13], respectively. Additionally, we analyzed the expression of CD150, CD80, Ly6C, CD69, CD40, B220, CD19, CD73, and CD117. This selection was informed by a comprehensive screening of 255 surface proteins, which showed that these molecules were expressed only by a fraction of BMPCs (Supplementary Fig. 6A). From this analysis, clusters 0, 1, 2, 5, 10 and 11 were prominent in the analyzed organs of sick mice (Fig. 3E).

For instance, we found cluster 0 represented 46% and 13% of BM- and kidney-derived PCs in sick mice, respectively, while it constituted below 5% in the other conditions (Fig. 3E). Cluster 0 expressed *Ccr10*, produced polymeric IgA and was CD150⁺ (Fig. 6B–D). In a previous study in humans, we found that CCR10 positive plasmablasts contribute to plasmacytosis in SLE[52].

Cluster 1 was notably more frequent in the spleen (42%) and BM (15%) of sick mice, in contrast to its diminished presence in healthy mice (1% and 0.35%, respectively) (Fig. 3E). Cluster 1 expressed *Prg2* (Fig. 6B), which was recently detected in neuropsychiatric SLE[53]. This cluster contained IgM-secreting cells and expressed inhibitory receptor LAG-3 (Fig. 6C, D) that identifies regulatory PCs[8].

We found that cluster 2 accumulated in BM, spleen, and kidney of sick mice (Fig. 3E) and expressed a higher level of *Pld4* and *Grn* (Fig. 6B). *Pld4* has been proposed as a susceptibility gene in SLE[54,55], systemic sclerosis[56] and RA[57]. Additionally, high expression of *Grn* has been linked to SLE pathogenesis[58–60]. This cluster was composed of PCs producing several Ig isotypes, including IgGs, and expressing CXCR3 (Fig. 6C, D), which is a chemokine receptor involved in migration of lymphocytes to inflamed tissues[61–63].

Cluster 5 was barely present in BM and spleen of sick mice but was detected in both these tissues of healthy mice, and in the kidneys

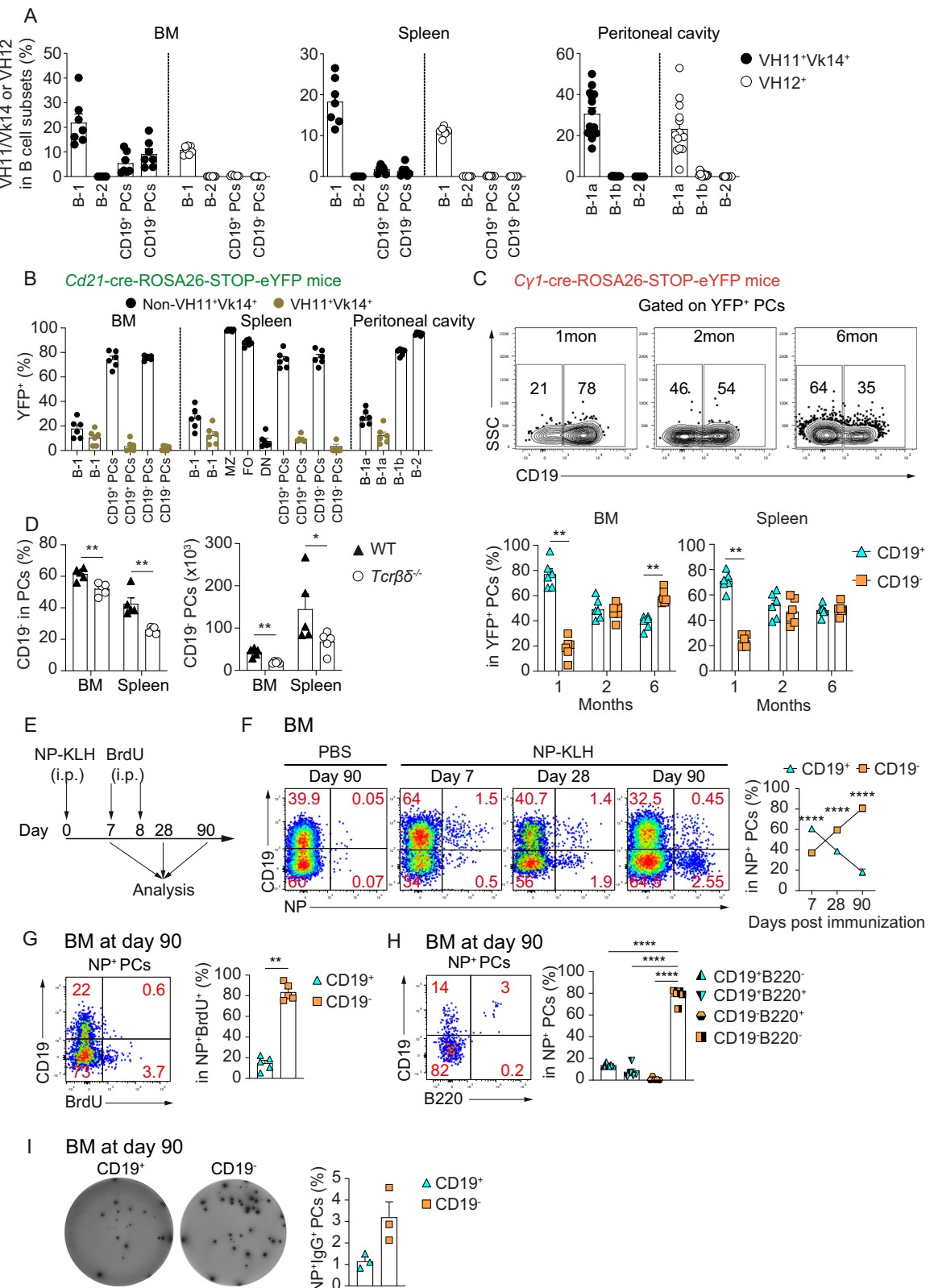

of sick mice (Fig. 3E). Cluster 5 expressed *Anxa2* (Fig. 6B), whose expression correlated with development of lupus nephritis and kidney inflammation[64].

Clusters 10 and 11 clearly accumulated in the BM of sick mice (Fig. 3E). Clusters 10 and 11 shared some expression patterns, but while cluster 10 contained IgM+ cells, cluster 11 consisted of IgM+-, IgA+- and IgG+-expressing cells (Fig. 6B, C). Both clusters co-

expressed CD39, CD326 and Ly6C but cluster 10 expressed CD117 while cluster 11 expressed CD150 (Fig. 6D).

Next, we analyzed expression of genes previously reported in SLE genome-wide associated studies (GWAS) (https://www.ebi.ac.uk/gwas/efotraits/MONDO_0007915) such as IFN-related genes like *Irf5, Tyk2, Slc15a4*, genes associated with antigen processing and presentation (*Tap2, Psmb8, H2-Ab1, H2-Eb2*), NF-κB regulators and ubiquitin-editing

**Fig. 2 | CD19⁻ BMPCs derive from various B cell subsets upon T-dependent and -independent reactions. A** VH11/Vk14 and VH12 frequencies within B cell and CD19⁺ or CD19⁻ PCs in BM, spleen and peritoneal cavity (PeC) of C57BL/6 mice. Subsets were identified as in Supplementary Fig. 2A. PeC B-1a, B-1b, B-2 cells are CD19⁺CD11b⁺CD5⁺, CD19⁺CD11b⁺CD5⁻ and CD19⁺CD11b⁻CD5⁻, respectively[8]. **B** YFP⁺ frequencies in B and PC populations from BM, spleen, PeC of *Cd21*-cre-ROSA26-STOP-eYFP mice, gated as in Supplementary Fig. 2B, C. PeC B-1a, B-1b, B-2 cells were identified as above[8](**C**) FACS plots (top) and graphs (bottom) show CD19⁺ and CD19⁻ PC frequencies in YFP⁺ PCs (CD39⁺CD138⁺YFP⁺) in BM and spleen from *Cy1*-cre-ROSA26-STOP-eYFP mice across ages. **D** Frequency and counts of CD19⁻ PCs in BM and spleen of *Tcrβδ*⁻/⁻ versus WT C57BL/6 controls. Gated as in Fig. 1A. **E** Workflow: C57BL/6 mice were immunized with NP-KLH i.p.; BrdU was injected i.p. on day 7 and 8; BM was analyzed at day 7, 28 and 90. **F** FACS plots show NP-specific and CD19 expression by BMPCs at days 7, 28 and 90. Graph shows CD19⁺ versus CD19⁻

frequencies in NP⁺ BMPCs (CD39⁺CD138⁺NP⁺). **G** FACS plot shows BrdU and CD19 expression in NP⁺ BMPCs. Graph shows CD19⁺ and CD19⁻ frequencies in NP⁺BrdU⁺ BMPCs (CD39⁺CD138⁺NP⁺BrdU⁺) on day 90. **H** FACS plot and graph shows B220 and CD19 frequencies in NP⁺ BMPCs on day 90. **I** Frequency of NP⁺IgG⁺ PCs in sorted CD19⁺ and CD19⁻ BMPCs (sorted as in Supplementary Fig. 1D) by ELISPOT on day 90. Each point represents one experiment. Experiments represent naive (**A–D**) and NP-KLH-immunized mice (**E–I**). Data combine four (*n* = 7/BM and spleen, 14/PeC; **A**), three (*n* = 6; **B**), two (*n* = 6/group; **C**) and two (*n* = 5/group; **D**) independent experiments. Immunization data combine three (**F, I**) and two (**G, H**) experiments (*n* = 17/day 7, 13/day 28 and 10/day 90: **F**; *n* = 5: **G, H**; and *n* = 19: **I**). Two-tailed Mann–Whitney test (**C, D, F, G**) or one-way Anova test (**H**) were used. **p = 0.0022 (**C**), **p = 0.0079 and *p = 0.032 (**D**), ****p < 0.0001 (**F**), **p = 0.0079 (**G**), and ****p < 0.0001 (**H**). *P* values ≥ 0.05 not shown. Data show mean ± SEM.

proteins (*Tnfaip3, Tnip1, Ube2l3*), B cell and Fc receptor signaling (*Blk, Cdkn1b, Fcgr2b, Fcgr3*), autophagy and vesicle trafficking (*Atg5, Tmem39a, Clec16a, Wdfy4*). We found high expression of these genes across several PC subsets, such as cluster 2, which was predominant in sick mice (Fig. 6E). Furthermore, we found that within the same cluster expression was tendentially higher in the sick clusters compared with the healthy ones (Fig. 6E and Supplementary Fig. 6B).

Collectively, these data highlight the diverse nature of SLE PCs and shed light on the potential functional heterogeneity within the context of the disease, providing a deeper understanding of potential pathogenic mechanisms in SLE.

## The most expanded BCR clones in kidney of lupus mice are highly mutated

The presence of disease- and organ-specific PCs in SLE mice prompted us to analyze BCR repertoire of PCs in the context of lupus. We analyzed a total of 32,596 BCRs from the PCs isolated from BM and spleen of healthy and sick mice and kidneys from sick mice (Fig. 3D). We detected 5674 and 7491 clones from BM and spleen of healthy mice; and 9610, 7663, and 2158 clones from BM, spleen, and kidney from sick mice, respectively. Unique clone counts were between 3000 to 5000 for BM and spleen and 1482 for kidney. Clonal abundance curves show that the BM of sick mice displayed a higher peak abundance at lower ranks compared to other tissues (Fig. 7A), indicating an accumulation of PCs that underwent selective clonal expansion. The diversity indices confirmed the overabundance of certain clones during illness (Supplementary Fig. 7A), reducing overall diversity. Interestingly, we found that the BM and spleen of healthy mice share some BCR clones, and this is also the case for BM and spleen of sick mice, while in the sick kidney we found almost only unique clones (Fig. 7B).

Next, we selected the 10 most abundant clones found in each BM, spleen, and kidney of sick mice (Fig. 7C–F and Supplementary Fig. 7B). Using these criteria, we further analyzed 28 clones, as two of the 10 most represented in BM were also among the 10 most represented clones in spleen and kidney. We found representatives of each clone among several clusters (Fig. 7C). In sick mice, the most abundant clones we found in BM and spleen were mainly IgM, while in the kidney there were almost no IgM PCs among the top 10 clones (Fig. 7D). They were mainly of isotype IgG or IgA (Fig. 7D). Clones that were IgG2c or IgA in kidneys of sick mice showed high mutation rates (Fig. 7D, E). Noteworthy, most clones showed low *Cd19* expression, especially in the clusters where they were most abundant (Fig. 7F). The CD19ˡᵒʷ clones that are expanded in the spleen of sick mice, interestingly, were not mutated. This seems to be different for the CD19ˡᵒʷ expressing clones in the kidney, which showed high mutation frequencies (Fig. 7E, F).

## Phosphatidylcholine-specific CD19⁻ B-1 derived PCs accumulate in BM of sick SLE mice

A striking observation from the BCR clonal analysis was that we found the most abundant BCR clone across all analyzed tissues, including in

the healthy BM and spleen. The most abundant clone clearly accumulated in the BM of sick mice (Figs. 7C, 8A). Furthermore, we found a shift towards largely expanded clonotypes in sick BM compared to healthy BM. These largely expanded clones in the BM of sick mice were in cluster 10 (Fig. 8A), in keeping with the overall expansion of cluster 10 in BM of sick mice. Examining V-gene usage and Ig isotypes of cluster 10, we identified a dominance of IGHV11-2/IGKV14-126 and IgM in the BM of sick mice (Fig. 8B, C and Supplementary Fig. 8A). Furthermore, mutation frequency of cluster 10 was the lowest across all clusters (Fig. 8D). Lower mutation rate and VH11/Vk14 BCRs usage is typical of B-1a cells. These cells were previously shown to recognize phosphatidylcholine (PtC)[8,32]. We confirmed using flow cytometric analysis and found significant increase of VH11⁺Vk14⁺ and PtC-specific cells in CD19⁻ PCs of sick mice compared to healthy controls (Fig. 8E, F). Particularly, we found more than 30% CD19⁻ BMPCs carried VH11⁺VK14⁺ BCR and reacted to PtC and nearly every VH11⁺Vk14⁺ and PtC-specific PCs in BM of sick *Sle123* mice did not express CD19 (Fig. 8E, F). Interestingly, VH11⁺Vk14⁺ and PtC-specific cells in WT mice also accumulate in CD19⁻ BMPCs (Supplementary Fig. 8B, C). Strikingly, nearly all VH11⁺Vk14⁺ PtC-specific PCs were CD19⁻ BMPCs in 24-month-old mice (Supplementary Fig. 8D). These data support that VH11⁺Vk14⁺ CD19⁺, PtC-specific CD19⁺ and VH11⁺Vk14⁺ PtC-specific CD19⁺ PCs transitioned into CD19⁻ PCs over time. Altogether, we conclude that in WT mice CD19⁻ PCs encompass self-reactive PCs, as in lupus mice, but the latter are unregulated, thus participating to the disease.

## Discussion

The pathogenic and protective functions of PCs have been documented in multiple models of ADs and clinical settings[3]. Thus, the key issue for the development of selective depletion therapy targeting only pathogenic PCs is to discriminate them from the protective compartment. Here, we made use of an SLE mouse model to study SLE PCs in several organs, including the kidneys. In this mouse model, SLE develops spontaneously. We analyzed PCs from prior and post SLE manifestation.

We identified several SLE PC clusters with relevant immunological functions. Some of these clusters are clearly associated with a specific organ, and remarkably expanded in disease. Among them, cluster 2 that might be considered as pathogenic since PCs in this cluster expressed inflammatory associated chemokine CXCR3, antibodies of IgG isotypes and this cluster was clearly expanded in kidney, spleen and BM of sick mice. Supporting this hypothesis, CXCR3⁺ PC subsets have been identified in the BM, spleen, and inflamed kidneys of other lupus-prone mice, where they contribute to autoantibody production[65,66]. CXCR3 expression on PCs is induced by inflammatory cytokine IFN-γ via a T-bet–dependent mechanism[51]. Consistent with these findings, CXCR3⁺ PCs have been shown to accumulate in the synovial fluid of patients with RA, suggesting their involvement in disease progression[14]. Moreover,

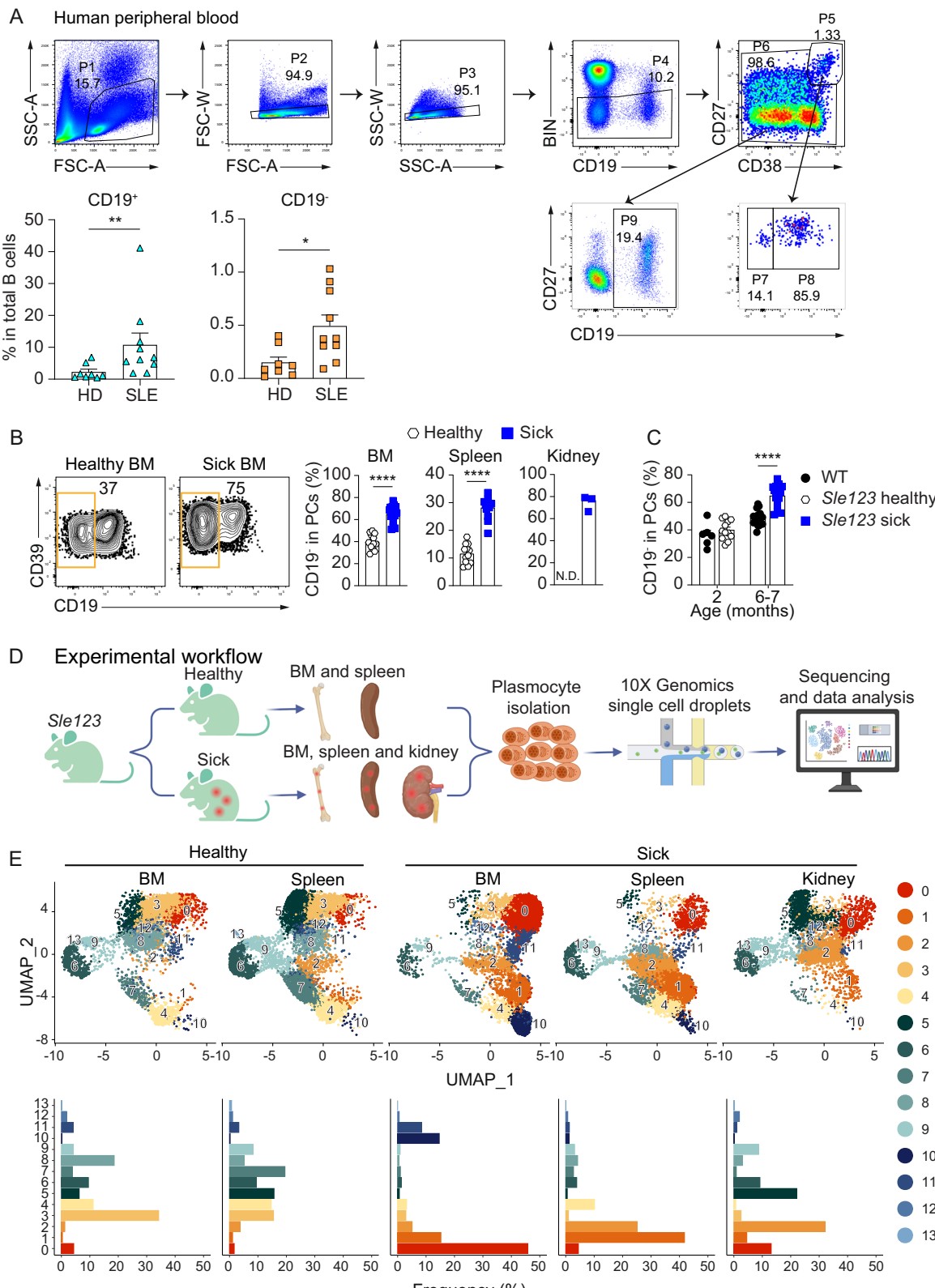

CXCR3+ memory B cells are significantly more frequent in patients with RA compared to healthy donors, further implicating these cells in RA pathogenesis[67]. In addition, CXCR3+ memory B cells and plasmablasts are the main autoreactive B cells in RA[68]. Moreover, CXCR3+ PCs are found at an elevated frequency in the peripheral blood and lamina propria of patients with ulcerative colitis compared to healthy donors[15].

Furthermore, we found that this PC cluster expressed *Pld4*, which has been associated to SLE and other ADs[54,56,57]. In addition, SLE patients were found to have expanded PLD4+ B cells in comparison with healthy donors and this cell population overlaps with DN2 B cells[55], which have been shown to be involved in SLE pathogenesis[69–71]. PLD4 expression is induced by TLR7 stimulation which plays a pivotal

**Fig. 3 | CD19⁻ PCs accumulate in human SLE blood and in target organs of SLE mice. A** Gating strategy for identifying peripheral blood CD19⁺ and CD19⁻ PCs. Frequencies of CD19⁺ and CD19⁻ PCs among total B cells (including PCs) in peripheral blood from healthy donors (HD) and patients with SLE (bottom). The BIN channel included CD3, CD14 and live/dead dye. **B** Representative FACS plots and graphs show frequency of CD19⁻ PCs in indicated organs of sick versus healthy *Sle123* mice. CD19⁻ PCs were gated on total PCs (CD39⁺CD138⁺) as in Fig. 1A[13]. **C** Graph shows frequency of CD19⁻ PCs in WT C57BL/6 versus still healthy *Sle123* mice at age of two months and WT C57BL/6 and sick *Sle123* mice at age of six to seven months. CD19⁻ PCs were gated on total PCs (CD39⁺CD138⁺) as in Fig. 1A. **D** Experimental workflow. Indicated organs were collected from healthy and sick *Sle123* mice for PC isolation (sorting strategy in Supplementary Fig. 3B). Cells were then subjected to 10X Genomics single cell droplet capture, RNA isolation, cDNA synthesis, single-cell sequencing, and data analysis. The scheme was created with BioRender.com. **E** UMAP (top) visualization of PC clusters in indicated organs from healthy and sick *Sle123* mice. Clusters 0-13 are color-coded. Frequencies (bottom) of PCs clusters in indicated organs from healthy and sick *Sle123* mice. Data show eight HDs and ten SLE patients (**A**), four independent experiments (*n* = 12 for BM and spleen of healthy mice and *n* = 14 for BM and spleen of sick mice; kidney, one experiment, *n* = 3 mice/group) (**B**), and three to four independent experiments (*n* = 6, 12, 15 and 14 mice for 2 months WT, healthy *Sle123*, 6-7 months WT and sick *Sle123*, respectively) (**C**). Groups were compared using two-tailed non-parametric Mann–Whitney test (**p = 0.0044 and *p = 0.012 for **A**, and ****p < 0.0001 for **B** and **C**). *P* values ≥ 0.05 are not shown. Data show mean ± SEM (**A**–**C**). For the scRNA-seq experiment, the *Sle123* mice were used, and data represent *n* = 6 mice/group (BM and spleen) and *n* = 13 mice (kidney) (**D, E**).

role in formation of pathogenic B cells and the production of auto-antibodies in SLE[55]. Moreover, this cell cluster also expressed a high amount of *Grn* which has been shown to relate to SLE pathogenesis[58–60]. IgG anti-DNA autoantibodies which drive SLE pathogenesis exhibit evidence of somatic hypermutation (SHM)[72]. Thus, autoreactive PCs generated by SHM from non-autoreactive naïve B cells in GC have been considered as drivers of the development of lupus in both mice[73,74] and humans[75]. Here, we found that cluster 2 PCs accumulated elevated levels of mutated BCR rearrangements and produced IgG in lupus mice. Cluster 2 exhibited higher expression of MHC-II and B220 which have been shown to decrease during ASC maturation[29–31], suggesting that this cluster can contain short-lived PC. Consistent with our findings in mice, studies on human SLE have also suggested a role for short-lived PCs in the pathogenesis of SLE[69,76]. Regarding the expression of CD19 by this cluster in spleen and BM, we detected some CD19 expression. Altogether, cluster 2 seems to harbor pathogenic PCs and requires consideration when targeting pathogenic PCs in SLE is a treatment goal.

Cluster 10 is also worth attention as we found it to be expanded specifically in the BM of sick mice. This cluster is similar to a previously described subset of SLE-specific BM IgM PCs expressing elevated levels of CD39 and CD326[13]. Cluster 10 contained expanded clones, expressed IgM, and recognized PtC. These cells utilized VH11 gene segment and are derived from B-1 cells which have been shown to produce autoreactive antibodies and play a critical role in the pathogenesis of SLE[8,77–80]. VH11 is a germ line gene that has been demonstrated to encode antibodies that recognize dsDNA[81–85]. In PCs from autoimmune FcγRIIB-deficient mice, VH11⁺ cells were found in IgG isotype and exhibited polyreactivity to dsDNA, insulin, LPS, and nucleosomes[86]. In humans, PtC-specific antibodies have also been detected in the sera of individuals with autoimmune or infectious diseases[87–89]. Additionally, SLE patients with hemolytic anemia exhibit higher levels of anti-PtC IgM than those without this hematologic manifestation[90]. Interestingly, soluble PtC has been shown to inhibit the IgM anti-dsDNA reactivity[91], indicating its polyreactive activity. Notably, patients with antiphospholipid syndrome show a significant increase in the frequency of PtC-specific B cells, which correlates with elevated serum levels of anti-PtC IgM and IgG as measured by ELISA[92]. Taken together, these findings support that PtC-specific PCs are polyreactive and might contribute to autoimmune pathogenesis, including SLE.

PCs can become long-lived memory cells and secrete antibodies for extended periods of time[5,12,47]. The biology of LLPCs remains poorly understood due to their rarity and the lack of definitive surface markers distinguishing them from short-lived PCs. Recent advances using inducible mouse models that timestamp PCs upon tamoxifen administration have allowed researchers to track the fate of PCs over time, thereby providing deeper insights into LLPC biology[11,12,31,38,93,94]. LLPCs have been phenotypically characterized as B220^low/–, MHC-II^low/–, SLAMF6 (CD352)^low/–, CD19^low/–, CD93^hi, CD138^hi, Fcεr1γ^hi, and IL-13Rα1^hi compared to newly generated PCs[12,30,31]. Consistent with this, our unbiased large-scale screen of 255 surface proteins revealed that CD19⁻ PCs exhibit reduced expression of MHC-II, B220, and CD352, confirming their LLPC-like phenotype. We further identified that LLPCs downregulated additional surface molecules such as CD2, CD20, CD38, CD73, CD80, CD86, CD365, and TIGIT, while upregulating CD49b, CD69, and JAML. These phenotypic features offer new markers to better study LLPCs in the future. Additionally, trajectory analyses in recent studies suggest that antigen-specific PCs undergo in situ maturation within the BM[11,19,95]. Using kinetic studies and BrdU pulse labeling, we observed that the majority of antigen-specific PCs are CD19⁺ upon arrival in the BM and they become CD19⁻ PCs over time. Adoptive transfer experiments in *Rag*⁻/⁻ mice demonstrated a unidirectional transition from CD19⁺ to CD19⁻ PCs, supporting the in situ maturation model.

Our analysis of the expression of genes shown to be activators and repressors of CD19 expression in B cells[46] suggests that the downmodulation of CD19 during PC maturation is transcriptionally regulated. We found lower levels of CD19 activators in CD19⁻ compared with CD19⁺ counterparts. The latter expressed also high levels of CD19 repressors, while we found low expression of repressors in CD19⁻ PCs. This might be explained by the enrichment of CD19⁻ in sick mice and CD19⁺ in SLE being prompt to generate CD19⁻ PCs. The differentiation of PCs has been shown to be supported by epigenetic remodeling[96]. Thus, it is possible that once downmodulated, CD19 repressors are no longer needed. This is consistent with the fact that CD19⁻ PCs do not generate CD19⁺ PCs

In SLE, some pathogenic PCs producing autoantibodies have been shown to be LLPCs[97], despite anti-dsDNA PCs being usually considered short-lived. Targeting these PCs is a challenge in the treatment of SLE as they are refractory to conventional immunosuppression therapies[98–100]. For instance, Rituximab and Abatacept are ineffective at reducing autoantibody titers in SLE patients[101,102]. The absence of specific markers for pathogenic LLPCs in SLE complicates therapeutic targeting. Our previous study identified SLE-associated PCs with a CD39^hiCD326 (EpCAM)^hi phenotype, which were significantly enriched in diseased compared to healthy mice[13]. This was later supported by an independent group using PC-timestamped mice, identifying LLPCs as CXCR3⁻ EpCAM^hi[11]. In our current analysis, we found that Cluster 10 PCs, which accumulated in sick *Sle123* mice, exhibited a CXCR3⁻ EpCAM^hi phenotype. Additionally, these cells expressed autoreactive BCRs and co-expressed CD39^hiLy6C^hiCD117^hi. These findings may define the phenotype of a pathogenic LLPC subset in SLE and suggest potential targets for the development of more selective therapies.

CD19-directed CAR-T cell therapy has recently gained considerable interest and is considered as a promising treatment option for refractory SLE. Nevertheless, not all patients treated with CD19-directed CAR-T show a reduction in SLE-associated autoantibodies[23,25,26]. This can be related to the fact that, as we showed here, CD19⁺ PCs give rise to CD19⁻ PCs, while the reverse does not occur. It is likely that CD19⁻ PCs are resistant to CD19-directed CAR-T cell therapy. Notably, this therapy has no impact on pre-existing

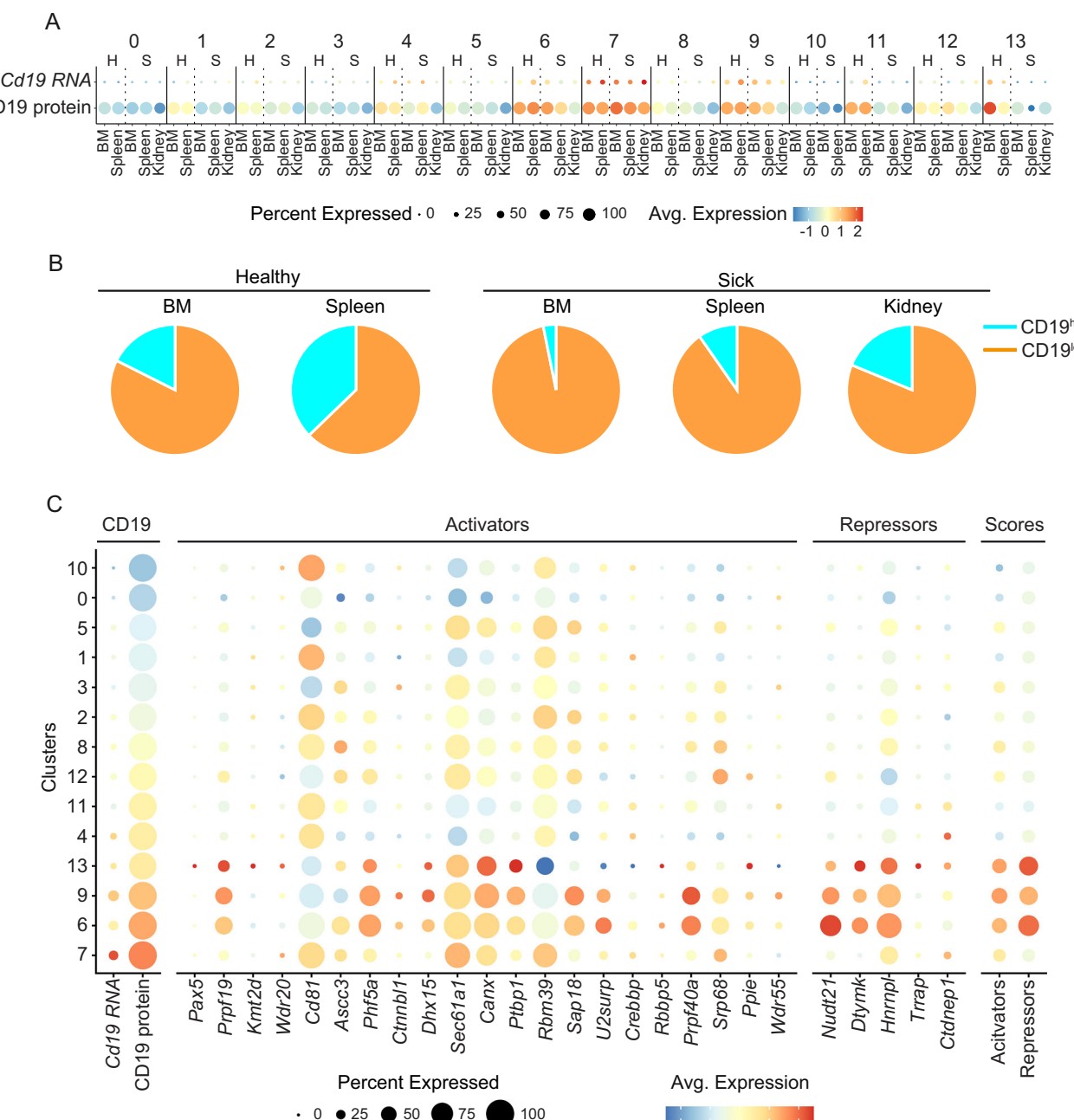

**Fig. 4 | Lupus-associated PC subsets show reduced CD19 expression and distinct regulatory profiles. A** Bubble plot shows the expression of *Cd19* mRNA (top) and CD19 protein (bottom) by the indicated clusters and organs from healthy (H) and sick (S) *Sle123* mice. Color scale shows the average expression. Bubble sizes show the frequency of cells expressing *Cd19* mRNA and CD19 protein by the indicated clusters. Based on CD19 RNA and protein expression, Clusters 4, 6, 7, 8, 9, 12 and 13 were identified as CD19hi clusters while Clusters 0, 1, 2, 3, 5, 10 and 11 expressed low amounts of CD19 and were therefore identified as CD19lo clusters. **B** Pie charts show the frequency of CD19hi (blue; representing concatenation of Clusters 4, 6, 7, 8, 9, 12 and 13) and CD19lo (orange; representing concatenation of Clusters 0, 1, 2, 3, 5, 10 and 11) PCs in indicated organs from healthy and sick *Sle123* mice quantified by scRNA-seq. **C** Bubble plot shows the expression of *Cd19* mRNA (first column) and CD19 protein (second column), as well as of selected activator and repressor genes associated with CD19 regulation as published[46] across the identified clusters. Bubble sizes show the percentage of cells within each cluster expressing the given transcript or protein, and the color scale represents the mean expression level. The two right columns display composite activator and repressor scores for each cluster. Each module score reflects the average scaled expression of the corresponding gene set (activators or repressors) in individual cells. Experiments and analyses were performed with *Sle123* mice. Data represent *n* = 6 mice/ group (BM and spleen) and *n* = 13 mice (kidney).

humoral immune responses, indicating that the PCs producing analyzed antigen-specific antibodies are CD19⁻ PCs[25]. Importantly, we found that in overt lupus, PCs are mostly CD19⁻ that are derived from various sources (T-independent, T-dependent, GC, environmental exposure) at least in a substantial part from CD19⁺ PCs. The current findings bring novel insights into PC biology, and in the context of antibody-mediated diseases and this can lead to changes in clinical decision making. Our data suggests that early intervention using depletion strategies, before CD19⁺ PCs develop into CD19⁻ PCs, is of key relevance to ensure appropriate treatment efficacy.

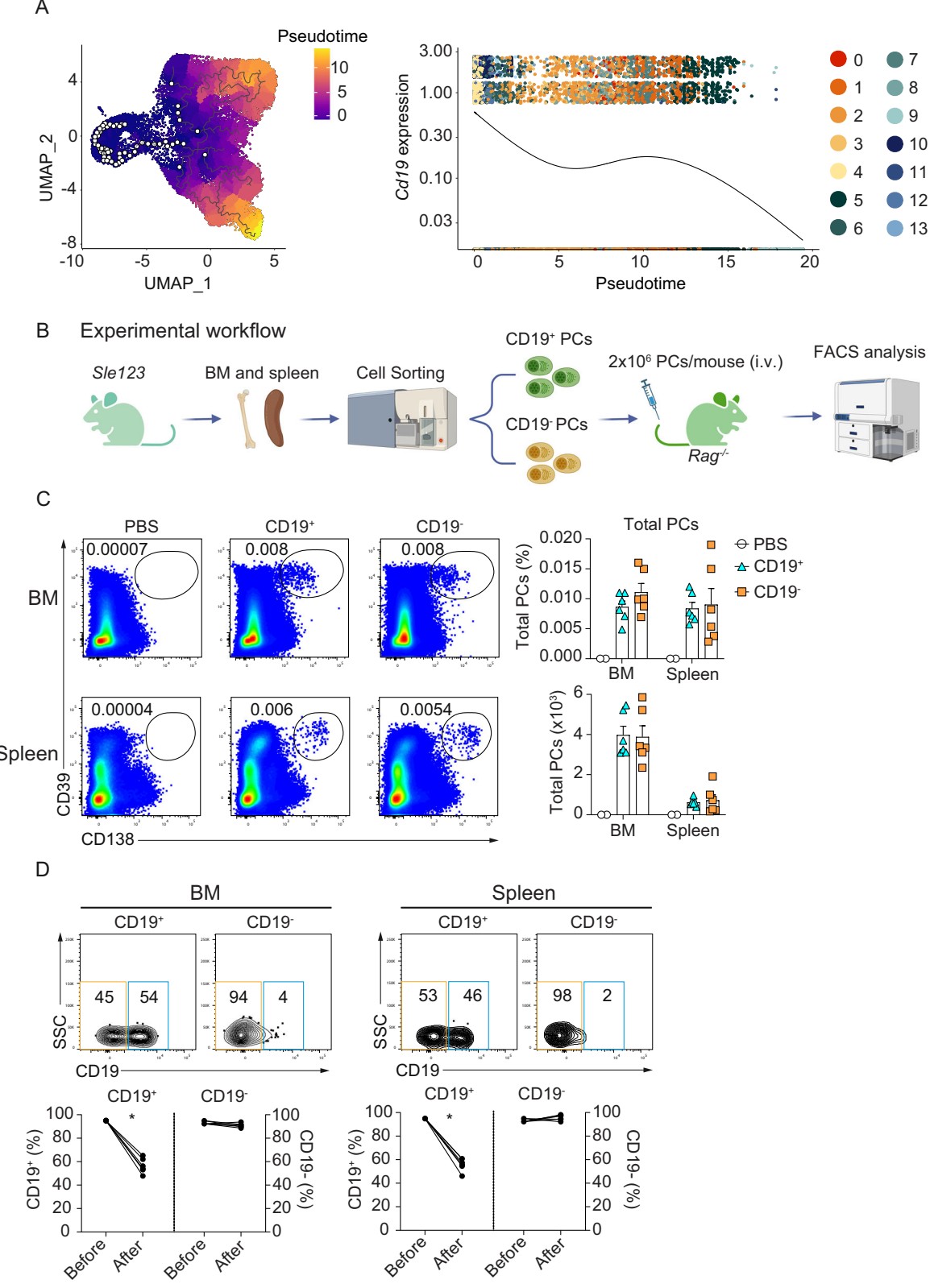

## Methods

### Study design

The aim of this research is to study PC subsets in SLE. First, we investigated the relevance of CD19⁻ PCs, known to contain autoreactive subpopulations in humans, in mouse models. Next, we explored the heterogeneity of PCs within the context of lupus pathogenesis, with a specific focus on their relationship to the CD19⁻ PC subset. For the first part, we assessed their distribution across various lymphoid tissues, evaluating their phenotypic properties, ontogeny, and differentiation under both homeostatic conditions and following immunization. For the second part, we isolated PCs from the bone marrow, spleen, and kidney of mice pre- and post-SLE onset, performing scRNA-seq, CITE-seq, and BCR repertoire analysis, and we examined CD19⁻ PCs in SLE patients. Herein, we reported disease- and organ-specific PC subsets,

**Fig. 5 | CD19⁻ PCs are derived from CD19⁺ PCs. A** UMAP plot (left) shows the predicted path of PC cell maturation analyzed by Monocle3[49] for PC clusters from *Sle123* mice. Color coding indicates pseudotime. The graph (right) shows *Cd19* expression by indicated clusters over pseudotime. **B** Experimental workflow. BM and spleen were collected from *Sle123* mice for sorting CD19⁺ and CD19⁻ PCs. Sorted cells were then transferred into *Rag⁻/⁻* mice (via i.v.) and analyzed 7 days later. As a control, some *Rag⁻/⁻* mice were injected with PBS only. The experimental workflow was created with BioRender.com. **C** Representative FACS plots (left) show the frequency of total PCs at day 7 after adoptive transfer in BM and spleen of *Rag⁻/⁻* mice which received PBS, sorted CD19⁺ or CD19⁻ PCs. The graphs show the frequency (top) and absolute number (bottom) of total PCs at day 7 after adoptive transfer in BM and spleen of *Rag⁻/⁻* mice received PBS, sorted CD19⁺ or CD19⁻ PCs. **D** Representative FACS plots (top) show the frequency of CD19⁺ versus CD19⁻ PCs in total BM and splenic PCs from *Rag⁻/⁻* mice which received either sorted CD19⁺ or CD19⁻ PCs. The corresponding graphs (bottom) show the frequency of CD19⁺ or CD19⁻ in BM and spleen from *Rag⁻/⁻* mice which received either sorted CD19⁺ or CD19⁻ PCs at the time before compared with after transfer. The frequency of CD19⁺ and CD19⁻ PCs at the time before transfer were cell purity after sorting. Data show compilation of *n* = 6 mice/group (BM and Spleen) and *n* = 13 mice (Kidney) (**A**) and two independent experiments (*n* = 6 mice/group) (**C**, **D**). Groups were compared using two-tailed paired *t* test (**D**) (*p = 0.031). *P* values ≥ 0.05 are not shown. Data show mean ± SEM.

as well as insights into their developmental trajectories and functional states.

## Mice

C57BL/6 (Jackson, Strain #:000664), *Cd21*-cre-ROSA26-STOP-eYFP (generated by crossing *Cd21*-cre (Jackson, Strain #:006368) with ROSA26-STOP-eYFP (Jackson, Strain #:00614), *Cγ1*-cre-ROSA26-STOP-eYFP (generated by crossing *Cγ1*-cre (from K. Rajewsky, MDC Berlin, Germany)[34] with ROSA26-STOP-eYFP (Jackson, Strain #:00614), *prdm1*eGFP (from S. Nutt, Walter and Eliza Hall Institute, Melbourne, Australia)[103], *Rag⁻/⁻* (Jackson, Strain #:008449), *Tcrβδ⁻/⁻* (Jackson, Strain #:002122), and *Sle123* (Jackson, Strain #:007228) (all C57BL/6 background) mice were bred under specific pathogen-free conditions at the DRFZ, Charité Universitätsmedizin (Berlin, Germany) and Institut Necker-Enfants Malades (Paris, France). Germ-free (GF) and specific-pathogen free (SPF) C57BL/6 control mice were also bred at the German Institute of Human Nutrition, Potsdam, Germany. Experimental and control animals were bred separately but housed under identical conditions in the same animal facility. Environmental conditions were standardized, with a 12-h light/12-h dark cycle controlled by automated timers with an adjustable twilight phase. The temperature was maintained at 22 ± 2 °C, and relative humidity was kept between 45–65%. *Sle123* mice were examined for proteinuria twice per week using Multistix (Siemens) from week 24 onwards. Sick *Sle123* mice (26-30 weeks old) were euthanized for analysis when proteinuria >100 mg/dL. Young *Sle123* mice (6-10 weeks old) without proteinuria were included as healthy control mice. Other mice were between 9-20 weeks old, unless otherwise stated. Both male and female mice were used in all experiments, with gender-matched controls included, except in experiments involving *Sle123* mice, where only females were used since most males do not develop the lupus phenotype. This approach reflects the higher prevalence of SLE in women compared to man. All animal experiments were reviewed and approved by Landesamt für Gesundheit und Soziales Berlin (LAGeSo, Berlin) under licenses T0344/17, G0072/20, and G0159/23. Mice were euthanized by cervical dislocation.

## Human participants

Human BM samples were obtained from nine patients undergoing total hip arthroplasty. Human spleens were obtained from five immune thrombocytopenia purpura patients under splenectomy surgery. Human tonsils samples were from six patients under routine tonsillectomy. Human peripheral blood samples were collected from 14 healthy donors. For SLE and control cohorts, peripheral blood samples were collected from 10 patients and eight healthy donors, respectively (Supplementary Table 1). All samples were collected in Charité Universitätsmedizin (Berlin, Germany). No compensation was paid. The study was approved by the institutional ethics committee of Charité Universitätsmedizin Berlin in accordance with the Declaration of Helsinki. Written informed consent was obtained from all participants prior to sample collection.

## Antibodies

Mouse specific monoclonal antibodies, DNA barcoded antibodies for Cellular Indexing of Transcriptomes and Epitopes by Sequencing (CITE-seq), human specific monoclonal antibodies, and other antibodies and Streptavidin are provided in Supplementary Tables 2-5.

## Mouse cell isolation

BM, splenocytes, mLN and kidney cells were isolated as previously reported[8,13,104]. Briefly, BM cells were collected by flushing out cells from the femur and tibia using a 1-mL syringe containing PBS supplemented with 0.5% (w/v) bovine serum albumin (BSA) (PBS/BSA) attached with a 26 G needle. Splenocytes and mLN cells were isolated by mashing spleen and mLN through a 70-μm cell strainer (Corning, Cat.# 352350) in a Petri dish with PBS/BSA, and then transferred into a 15-mL tube using a 25 G needle to obtain a single-cell suspension. Red blood cells in BM and spleen samples were lysed using red blood cell lysis buffer (Sigma). After lysis, the cells were resuspended in PBS/BSA. Kidney cells were isolated by mincing with a scalpel, manually crushed on a 70-μm cell strainer with a 3 mL syringe plunger, and then transferred into a 15-mL tube via a 26 G needle.

To collect peritoneal cavity (PeC) cells, mice were injected with 10 mL of cold PBS/BSA into the PeC using a 26 G needle, the peritoneum gently massaged, and an 18 G needle attached to a 10 mL syringe was inserted to collect the fluid containing PeC cells. The sample was then transferred into a 15-mL tube.

For isolation of Peyer's Patches (PP) and laminar propria (LP) lymphocytes, small intestine was cut below pyloric sphincter and above the cecum, and PP were then collected by removing them from intestine with curved scissors and put in HBSS (Gibco, Cat.# 14185-052) + 5% FBS (Corning, Cat.# 35-079-CV) (v/v) + 100 U/mL Penicillin/100 μg/mL Streptomycin (Gibco, Cat.#15140-122) + 10 mM HEPES (Gibco, Cat.#15630056). PP lymphocytes were isolated by mashing through a 70-μm cell strainer (Corning, Cat.# 352350) in a Petri dish with PBS/BSA and then transferred into a 15-mL tube using a 25 G needle to obtain single-cell suspension. The PP-removed intestine was opened longitudinally using fine scissors, put in HBSS (Gibco, Cat.# 14185-052) + 5% FBS (Corning, Cat.# 35-079-CV) (v/v) + 100 U/mL Penicillin/100 μg/mL Streptomycin (Gibco, Cat.#15140-122) + 10 mM HEPES (Gibco, Cat.#15630056) and shaked to remove feces and mucus. The intestine was then cut into smaller pieces, put in HBSS (Gibco, Cat.# 14185-052) + 5% FBS (Corning, Cat.# 35-079-CV) (v/v) + 100 U/mL Penicillin/100 μg/mL Streptomycin (Gibco, Cat.#15140-122) + 10 mM HEPES (Gibco, Cat.#15630056) + 5 mM EDTA and shaked for 20 min. The intestine was then washed, cut into small pieces and digested in HBSS (Gibco, Cat.# 14185-052) + 5% FBS (Corning, Cat.# 35-079-CV) (v/v) + 100 U/mL Penicillin/100 μg/mL Streptomycin (Gibco, Cat.#15140-122) + 10 mM HEPES (Gibco, Cat.#15630056) + 0.5 mg/mL Collagenase D (Sigma-Aldrich, Cat.# 11088866001) + 1.5 g/mL Collagenase type VIII (Sigma-Aldrich, Cat.# C2139) + 30 μg/mL Dnase (Roche, Cat.# 11284932001) for 30 min at 37°C on shaker. Lamina propria lymphocytes were eventually isolated by Percoll gradient centrifugation using Percoll (Sigma-Aldrich, Cat.# 17-0891-02) 75% and 40%.

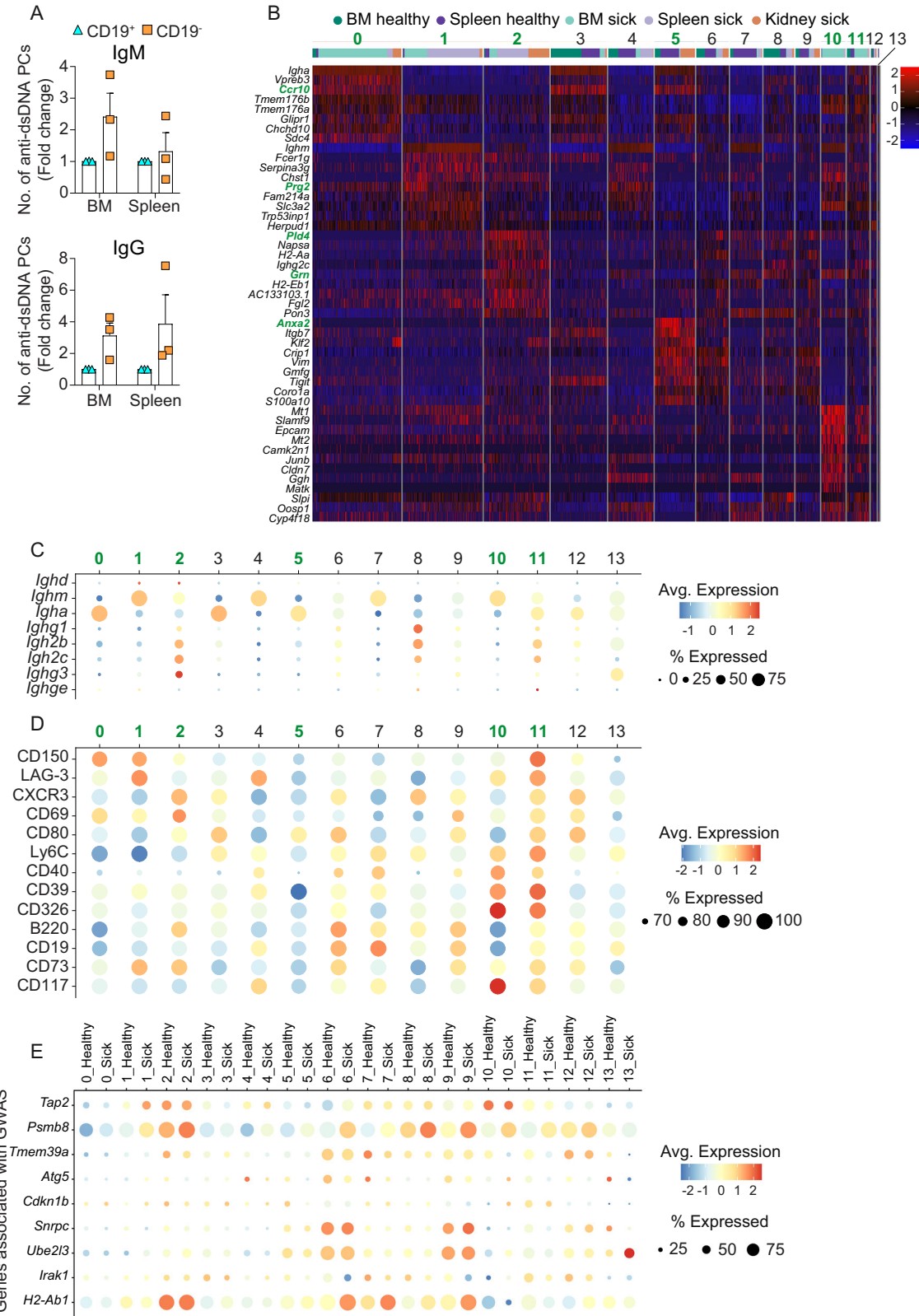

### Mouse cell staining for flow cytometry analysis

Cells were counted using MACSQuant Flow Cytometer (Miltenyi Biotec) prior to staining. Splenocytes, BM, mLN and PeC cells were first incubated with 30 μg/mL anti-Fc receptor antibody (clone 2.4G2) for 15 min to block unspecific binding, followed by fluorochrome- and/or biotin-conjugated antibodies against surface molecules and/or surface binding to phycoerythrin (PE)-conjugated NP or PtC-containing Liposomes labeled with Texas Red (FormuMax, Cat.# F60103F-TR) for 20 min. Biotinylated antibodies were subsequently labeled with fluorochrome-conjugated Streptavidin for another 20 min.

For the screening experiments, surface-stained cells were transferred into LEGENDScreen plate (LEGENDScreen Mouse PE Kit, BioLegend, Cat.# 700005) containing PE-conjugated screening antibody and incubated another 20 min.

**Fig. 6 | Single-cell transcriptomic profiling reveals disease- and organ-specific PCs in SLE mice. A** Graphs show the fold changes in the number of anti-dsDNA IgM⁺ (top) and IgG⁺ (bottom) CD19⁻ relative to CD19⁺ PCs in sick *Sle123* mice, as measured by ELISPOT assay. Data show compilation of three independent experiments ($n = 18$ mice). Data show mean ± SEM. **B** Heatmap shows the expression of top 10 differentially expressed genes (DEG) (row) by clusters 0, 1, 2, 5, 10 and 11 for each cell (column). **C** Bubble plot shows the expression levels of immunoglobulin genes per cluster. **D** Bubble plot of the differential expression of key surface markers obtained by CITE-seq per cluster. **E** Bubble plot shows expression of the subset of genes (that are detected in ≥ 25 % of cells) of the top 50 mouse orthologs of human genes associated with systemic lupus erythematosus (GWAS, MONDO:0007915), ranked by minimal SNP P-value. Bubble size indicates the percentage of cells in that sample with nonzero expression of the given gene, and bubble color encodes the average scaled expression level. **B**–**E** Color scale shows the z-scores of the average expression of a gene within indicated clusters. Bubble sizes show the frequency of cells expressing indicated genes by indicated clusters. Experiments and analyses were performed with *Sle123* mice. Data show compilation of $n = 6$ mice/group (BM and Spleen) and $n = 13$ mice (Kidney). BM (Bone marrow).

Dead cells were excluded by staining with propidium iodide (Sigma-Aldrich, Cat.# P4864).

For Ki-67 intra-nuclear staining, cells were first stained with LIVE/DEAD Fixable Aqua Dead Cell Stain Kit (Invitrogen, Cat. # L34957), followed by Fc receptor block and surface staining, and finally intra-nuclear stained for Ki-67 using Foxp3 staining kit (Ebioscience/Thermo Fisher Scientific, Cat. #00-5523-00).

For BrdU detection, cells after staining with LIVE/DEAD Fixable Aqua Dead Cell Stain Kit (Invitrogen, Cat. # L34957), Fc receptor block and surface markers were processed with BD Pharmingen BrdU Flow Kit (BD Biosciences, Cat.# 552598) according to manufacturer's instructions.

NP- and PtC-specific PCs were detected using the PE-conjugated NP and PtC- containing Liposomes labeled with Texas Red (FormuMax, Cat. #F60103F-TR) thanks to surface binding, respectively.

Stained cells were measured using a BD FACSymphony or BD LSRFortessa.

### Mouse plasma cell purification

For single-cell RNA sequencing experiments, BM cells were collected by flushing out cells from femur and tibia using a 1-mL syringe containing 2 µg/mL Actinomycin D (Sigma-Aldrich, Cat.# SBR00013-1ML) in PBS/BSA attached to a 26 G needle. Splenocytes and kidney cells were isolated by mashing spleen or kidney through a 70-µm cell strainer (Corning, Cat.# 352350) in a Petri dish with 2 µg/mL Actinomycin D (Sigma-Aldrich, Cat.# SBR00013-1ML) in PBS/BSA, and then transferred into a 15-mL tube using a 25 G needle to obtain single-cell suspensions. After Fc receptor block, cells were incubated with anti-CD138-PE antibody, followed by anti-PE microbeads (Miltenyi Biotec). After auto-MACS, the CD138⁺ enriched cells were stained with CITE-seq antibodies, including B220, CD19, CD39, CD40, CD69, CD73, CD80, CD117, CD150, CD183 (CXCR3), CD223 (LAG-3), CD326 (EpCAM), Ly6C and together with flow cytometry antibodies, including CD138, CD81, CD11b, CD11c, and CD3. DAPI was added before sorting to exclude dead cells. Total PCs CD138⁺CD81⁺CD11b⁻CD11c⁻CD3⁻DAPI⁻ were sorted by FACS sorter Aria II. Sorted cells were counted using MACSQuant Flow Cytometer (Miltenyi Biotec) prior to further processing for single-cell RNA sequencing.

For ELISPOT assay and ELISAs, isolation of PC subpopulations was performed as previously reported[8,104]. Briefly, BM cells, splenocytes were stained with anti-CD138-PE antibody and followed by anti-PE microbeads (Miltenyi Biotec). After auto-MACS, the CD138⁺ enriched cells were stained with CD19, CD39, CD138, CD3, CD11b, and CD11c. DAPI was added before sorting to exclude dead cells. CD39⁺CD138⁺CD19⁺CD3⁻CD11b⁻CD11c⁻DAPI⁻ and CD39⁺CD138⁺CD19⁻CD3⁻CD11b⁻CD11c⁻DAPI⁻ were sorted by FACS sorter Aria II.

For adoptive transfer of CD19⁺ and CD19⁻ PC experiments, BM cells and splenocytes were pooled before enrichment for CD138⁺ cells by MACS and then sorted for CD19⁺ and CD19⁻ PCs using a FACS sorter Aria II.

### Quantification of antibody-secreting cells by ELISPOT assay

Quantification of antibody-secreting cells by ELISPOT assay was performed as previously reported[8,104]. Briefly, sorted CD39⁺CD138⁺ CD19⁺CD3⁻CD11b⁻CD11c⁻DAPI⁻ and CD39⁺CD138⁺CD19⁻CD3⁻CD11b⁻CD11c⁻DAPI⁻ were plated at starting concentration of $1.5 \times 10^3$ cells/200 µL/well with four successive five-fold serial dilutions, in Multi-ScreenHTS IP Filter Plate (Millipore, Cat.# MSIPN4510) pre-coated with 50 µL anti-mouse Ig(H + L) chain (Southern Biotechnology Associates). After 3 h of incubation, cells were washed out and the plate was incubated for 1 h with anti-IgA-, IgM- or IgG-alkaline phosphatase (Southern Biotechnology Associates). ELISPOT was developed using BCIP/NBT substrate (Thermo Fisher Scientific, Cat.# 34042) and analyzed by AID EliSpot Reader System. For quantification of NP-specific antibody secreting cells, the plates were coated with 20 µg/mL NP-OVAL (Biosearch Technologies, Cat.# N-5051) at 4 °C overnight. For quantification of dsDNA-specific antibody secreting cells, the plates were pre-coated with 20 µg/mL methylated BSA (Sigma-Aldrich) in PBS for 2 h and subsequently coated with 20 µg/ml calf thymus dsDNA (Sigma Aldrich, Cat.# D4522-5MG) in PBS at 4 °C overnight. After plate blocking, sorted cells were plated and processed as described above.

**Quantification of immunoglobulins by ELISAs.** Sorted CD39⁺CD138⁺CD19⁺CD3⁻CD11b⁻CD11c⁻DAPI⁻ and CD39⁺CD138⁺CD19⁻CD3⁻CD11b⁻CD11c⁻DAPI⁻ were seeded at $5 \times 10^4$ cells per well in the 96-well round-bottom culture plates. Cells were cultured in complete RPMI (RPMI + 10% FCS + Penicillin/Streptomycin), 1 µg/ml LPS (Sigma-Aldrich, Cat.#. L2637-5MG) or 0.5 µM CpG (Invivogen, Cat.# tlrl-1826-1), along with recombinant IL-6 (20 ng/ml) and IL-21 (20 ng/ml). Cultures were incubated at 37 °C and 5% $CO_2$, and supernatants were harvested after 24 h for analysis. Mice sera were prepared as previously reported[13]. IgA, IgM and IgG concentration in culture supernatant and sera were determined by ELISAs. In brief, high binding 96-well ELISA plates (Corning) were coated with 50 µL anti-mouse Ig (H + L)-UNLB, human, adsorbed (SouthernBiotech) diluted in PBS to a final concentration of 1.5 µg/mL at 4ºC overnight. After washing, the plates were blocked with PBS/BSA (3% w/v) (Sigma-Aldrich) at 37ºC. After 2 h of incubation, the plates were washed and mouse IgA, IgM and IgG standards (SouthernBiotech) and diluted culture supernatant or sera were added, incubated for 2 h at 37ºC and washed again. 100 µL of AP-conjugated detection antibodies (SouthernBiotech) were then added, incubated for 1 h at room temperature and wash again to remove unbound antibodies. Assay were then developed using 1-Step PNPP Substrate Solution (Thermo Fisher Scientific) and measured with spectrophotometer (SpectraMax Plus 384, Molecular Devices). Antibody concentration was determined by Softmax Pro software.

### Single-Cell RNA-library preparation and sequencing

Single-Cell RNA-library preparation and sequencing were processed as a previous report[20]. In brief, single-cell suspensions, obtained through cell sorting, were counted, and subjected to the 10x Genomics workflow for cell capturing. scRNA gene expression (GEX), BCR and CITE-Seq library preparation were performed using the Chromium Single-Cell 5′ Library & Gel Bead Kit version 2 (10x Genomics, Cat. # 1000263), as well as the Single-Cell 5′ Feature Barcode Library Kit (10x Genomics, Cat. # 1000256). After cDNA amplification, the Cite-Seq libraries were separately prepared using the Dual Index Kit TN Set A (10x Genomics,

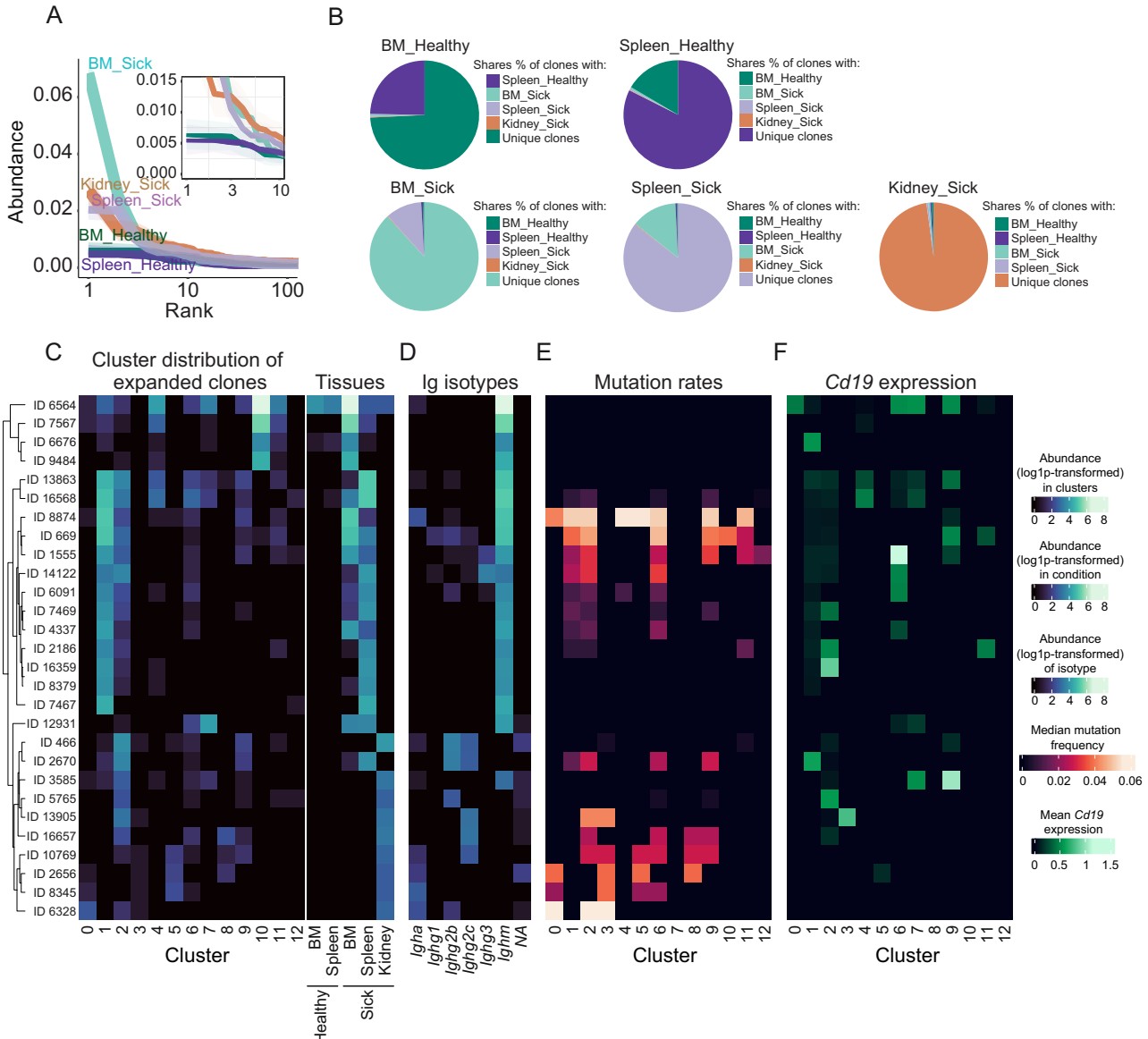

**Fig. 7 | The most expanded BCR clones in kidney of lupus mice are highly mutated. A** Rank-abundance distribution plot of clonotypes displays the relative abundance (%) of clones in descending order across all tissues and conditions. Data points for each organ are color-coded: BM_healthy (green), spleen_healthy (purple), BM_sick (light green), spleen_sick (light purple) and kidney_sick (orange). **B** Pie charts depicting the percentage of shared clonal populations among BM, spleen, and kidney tissues under healthy and diseased conditions. Each chart illustrates the extent of clonal overlap among tissues, as well as the proportion of unique clones. **C** Heatmap shows the abundance of the top 10 expanded clones within each BM, spleen, and kidney tissues in a diseased state across different clusters. Each row represents a unique clone identified by its clone ID, and each column represents a specific cluster (0–12). None of the top clones were present in cluster 13. Color intensity indicates the log1p-transformed abundance within each cluster. Heatmap showing the abundance of the top clones across the different conditions and tissues (BM_healthy, spleen_healthy, BM_sick, spleen_sick, kidney_sick). Color intensity indicates log1p-transformed abundance. **D** Heatmap showing distribution of the abundance of Ig isotype per clone. Color intensity indicates log1p-transformed abundance. **E** Heatmap showing the median mutation frequency of the top clones across the different clusters. Color intensity represents the median mutation frequency, highlighting areas with higher mutation rates. **F** Heatmap illustrating the mean expression levels of *Cd19* of the top clones across different clusters (0–12). None of the top clones were present in cluster 13. Color intensity indicates mean expression of *Cd19*. Experiments and analyses were performed with *Sle123* mice. Data show compilation of *n* = 6 mice/group (BM and Spleen) and *n* = 13 mice (Kidney).

Cat. # 1000250). BCR target enrichment was carried out using the Chromium Single-Cell V(D)J Enrichment Kit for mouse B cells (10x Genomics, Cat. # 1000255). The final GEX and BCR libraries were obtained after fragmentation, adapter ligation and final Index PCR using the Dual Index Kit TT Set A (10x Genomics, Cat. # 1000215). Library quantification was performed using the Qubit HS DNA assay kit (Invitrogen, Cat. # Q33231) and fragment sizes were determined using the Fragment Analyzer with the HS NGS Fragment Kit (1-6000 bp) (Agilent, Cat. # DNF-474-0500).

Sequencing was conducted on a NextSeq2000 device (Illumina), following the sequencing conditions recommended by 10x Genomics for libraries prepared with Next Gem Reagent Kits v2. For 5' GEX and Cite-Seq libraries (read1: 26nt, read2: 90nt, index1: 10nt, index2: 10), NEXTSeq 1000/2000 P3 reagent kits (200 Cycles, Illumina, Cat. # 20040559) were utilized. BCR libraries (read1: 151nt, read2: 151nt, index1: 10nt, index2: 10nt, 2% PhiX spike-in) were sequenced using NEXTSeq 1000/2000 P3 reagent kits (300 Cycles, Illumina, Cat. # 20040559).

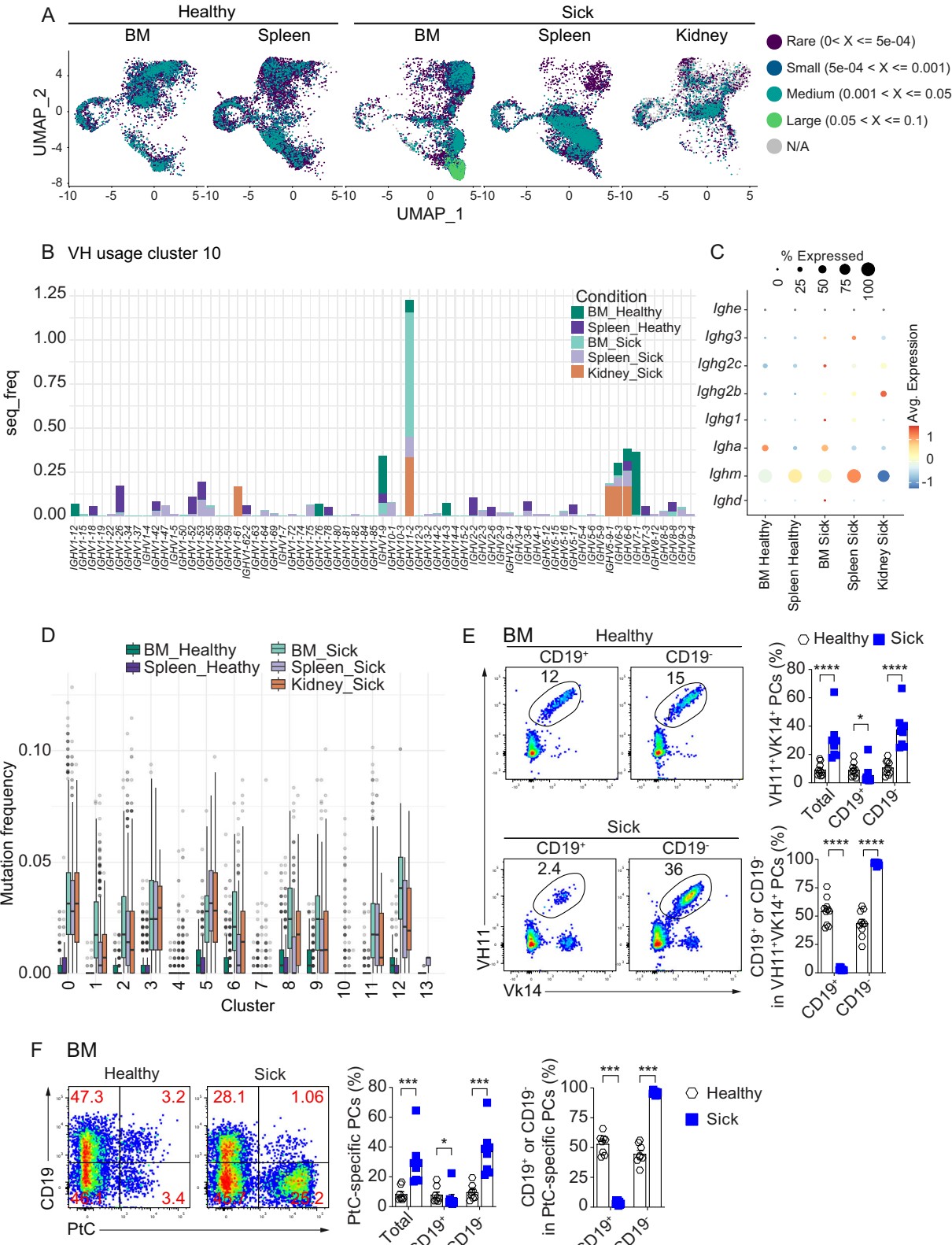

## Single-cell transcriptome and repertoire profiling

The raw sequencing data was processed using cellranger software (version 5.0.0) which included steps like demultiplexing, mapping, cell identification, and gene expression quantification via the cellranger count pipeline. As reference genome mm10-2020-A was used. Downstream data analysis and visualization was performed using R (version 4.3.1) using packages like Seurat[105] and the Immcantation[106,107]

workflows for repertoire profiling. For transcriptome analysis immunoglobulin related genes besides the heavy chain genes *Ighd, Ighm, Igha, Ighg1, Ighg2b, Ighg2c, Ighg3* and *Ighe* were filtered out. Low-quality cells were excluded based on low counts of RNA, overall genes, housekeeping genes, and high percentage of mitochondrial reads.

In our study, we employed the Immcantation framework for comprehensive analysis of B-cell receptor (BCR) repertoires. Initially,

**Fig. 8 | Phosphatidylcholine-specific CD19⁻ B-1 derived PCs accumulate in BM of sick SLE mice. A** UMAPs of the different tissues and conditions showing relative abundance of clonotypes by color gradient. Clonotypes are grouped as rare (0 < X ≤ 5e−04, dark blue), small (5e−04 < X ≤ 0.001, blue), medium (0.001 < X ≤ 0.05, green), and large (0.05 < X ≤ 0.1, light green) relative abundance. **B** Stacked bar plots depict heavy chain variable (V) usage in cluster 10 across tissues and conditions, illustrating changes in VH gene frequencies. **C** Bubble plots show Ig isotype expression of cluster 10 across tissues and conditions. **D** Boxplots display mutation frequencies across 13 identified clusters in BM, spleen, and kidney under healthy and sick conditions. Boxplots show median (center line) and interquartile range (IQR, 25th–75th percentiles), whiskers extend to 1.5×IQR, and outliers as individual points. **E** Representative FACS plots (left) show VH11⁺Vk14⁺ frequencies in CD19⁺ and CD19⁻ BMPCs in healthy and sick *Sle123* mice. CD19⁺ and CD19⁻ BMPCs were gated on total PCs (CD39⁺CD138⁺) as in Fig. 1A. Graphs (right) quantify VH11⁺Vk14⁺ in total, CD19⁺ and CD19⁻ BMPCs and the frequency of CD19⁺ and CD19⁻ PCs in VH11⁺Vk14⁺ BMPCs (CD39⁺CD138⁺VH11⁺Vk14⁺) from healthy and sick *Sle123* mice. **F** Representative FACS plots (left) show frequency of PtC-specific and CD19 expression in BM of healthy and sick *Sle123* mice. CD19⁺ and CD19⁻ BMPCs were gated on total PCs (CD39⁺CD138⁺) as in Fig. 1A. Graphs (right) quantify PtC-specific PCs in total, CD19⁺ and CD19⁻ PCs and the frequency of CD19⁺ and CD19⁻ PCs in PtC-specific PCs (CD39⁺CD138⁺PtC⁺) from healthy and sick *Sle123* mice. Experiments and analyses were performed with *Sle123* mice. Data show compilation of n = 6 mice (BM and Spleen), n = 13 mice (Kidney) (**A**–**D**), and three independent experiments (**E**, **F**) (n = 10 mice for healthy and 9 mice for sick) (**E**) and (n = 8 mice/group) (**F**). Groups were compared using a two-tailed non-parametric Mann–Whitney test (*p = 0.013 and ****p = 0.000022 for **E**, and *p = 0.038 and ***p = 0,000155 for **F**). Data show mean ± SEM.

variable (V), diversity (D), and joining (J) gene assignments were conducted using IgBLAST. The analysis proceeded with a stringent filtering process to retain only productive sequences, while excluding samples containing multiple heavy chains or those with light chain information in the absence of corresponding heavy chain data. Clonal relationships were elucidated through a hierarchical clustering approach, which was informed by the distribution of nearest-neighbor distances among sequences. This method allowed for the distinction between clonally related sequences and unrelated ones. Clonal groups were subsequently defined based on sequence similarities across both heavy (IGH) and light (IGK or IGL) chains. To reconstruct the germline sequence of the unmutated ancestor for each clone we utilized a reference database of known alleles provided by IMGT. This comprehensive analytical workflow enabled a nuanced exploration of BCR repertoire diversity and the clonal evolution of B-cell populations.

**Data processing and quality check.** After combining all datasets from different tissues (Supplementary Fig. 3C) and evaluating cluster stability[108] using the cluster tree (Supplementary Fig. 3D), we selected a clustering resolution of 0.6. Next, we identified and excluded low-quality cells based on criteria such as low counts of RNA, overall genes, housekeeping genes, and high percentage of mitochondrial reads (Supplementary Fig. 3E). Additionally, we removed non-PC contaminating cells based on the expression of *Cd3e* and *Pax5* (Supplementary Fig. 3F, G).

**NP-KLH immunization**
C57BL/6 mice were immunized intraperitoneally with 200 μg of NP-KLH (Biosearch Technologies, Cat.# N-5060) precipitated in alum and analyzed at time points as indicated figure legends.

**BrdU pulse labeling**
C57BL/6 mice were injected i.p. with 2 mg of 5-bromo-2′-deoxyuridine (BrdU) solution (BD Biosciences, Cat.# 552598) at day 7 and 8 after immunizing with 200 μg of NP-KLH (Biosearch Technologies, Cat.# N-5060).

**Adoptive cell transfer**
BM cells and splenocytes from *Sle123* mice were pooled and stained with anti-CD138-PE antibody for 20 min, followed by incubation with anti-PE microbeads (Miltenyi Biotec). CD138⁺ cells were enriched by using an autoMACS. The enriched CD138⁺ cells were stained with CD19, CD39, CD138, CD3, CD11b, and CD11c for 20 min. DAPI was added before sorting to exclude dead cells. CD39⁺CD138⁺CD19⁺ CD3⁻CD11b⁻CD11c⁻DAPI⁻ and CD39⁺CD138⁺CD19⁻CD3⁻CD11b⁻CD11c⁻DAPI⁻ were sorted using a BD FACS sorter Aria II. Sorted CD39⁺ CD138⁺CD19⁺CD3⁻CD11b⁻CD11c⁻DAPI⁻ and CD39⁺CD138⁺CD19⁻CD3⁻ CD11b⁻CD11c⁻DAPI⁻ were washed once with cold PBS, resuspended in PBS, and injected i.v. into *Rag⁻/⁻* mice (2×10⁶ cells/200 μl PBS/mouse). Recipient mice were analyzed seven days later.

**Human mononuclear cell isolation and flow cytometry**
Human BM, spleen, tonsil and peripheral blood mononuclear cells were isolated as previously reported[13,17]. BM (30×10⁶), splenocytes, tonsil cells and peripheral blood (PB) mononuclear (20×10⁶) cells were pre-labeled with live/dead dye (LIVE/DEAD Fixable Blue Dead Cell Stain Kit, Invitrogen) according to the manufacturer's instruction. Labeled cells were then blocked with FcR blocking reagent (Miltenyi Biotec) for 5 min and followed with surface molecule staining with fluorochrome-conjugated antibodies for 20 min. Stained cells were measured using a BD LSRFortessa.

**GWAS analysis**
GWAS summary statistics for SLE (MONDO:0007915) were downloaded on July 10, 2025, and analyzed using R. Human gene symbols were then mapped to their mouse orthologs via the biomaRt[109] interface to Ensembl, and only those genes with valid mouse homologs were retained. The top 50 genes ranked by minimal SNP-value of mouse orthologs are shown split by diagnostic group.

**Statistical analysis**
Quantification and statistical analysis were performed using GraphPad Prism 9 (GraphPad Software). Groups were compared using non-parametric Mann–Whitney test, paired t test or one-way Anova test as indicated in figure legends. Considered p-values were: ᶰˢp ≥ 0.05; *p < 0.05; **p < 0.01; ***p < 0.001; ****p < 0.0001. Flow cytometry data were analyzed with FlowJo software (version 10.9 and 10.10, TreeStar).

**Reporting summary**
Further information on research design is available in the Nature Portfolio Reporting Summary linked to this article.

## Data availability
All data are included in the Supplementary Information or available from the authors, as are unique reagents used in this Article. The raw numbers for charts and graphs are available in the Source Data file. The ScRNA-seq data generated in this study have been deposited in the GEO database under accession code (GSE275094) and are publicly available here: https://www.ncbi.nlm.nih.gov/geo/query/acc.cgi?acc=GSE275094. For further information and requests for resources, contact the corresponding author, Andreia C. Lino (andreialino79@gmail.com). Source data are provided with this paper.

## Code availability
The software utilized in this study for the scRNA analysis is open-source, and no original code concept was developed for data processing and analysis. Cellranger from 10x genomics: https://support.10xgenomics.com/single-cell-gene-expression/software/downloads/latest Seurat packages 4.1.1: https://cloud.r-project.org/web/packages/Seurat/index.html Monocle3 package: https://cole-trapnell-lab.github.io/monocle3/docs/installation/ Immcantation Workflow:

https://immcantation.readthedocs.io/en/stable/# A full list of used packages can be found in Supplementary Table 6.

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

## Acknowledgements

The authors thank J. Kirsch, T. Kaiser and K. Heinrich (Flow Cytometry Core Facility of the DRFZ); Johanna Ndikung, Christin Kabus and Katrin Lehmann (laboratory managers of the DRFZ); Minh Quang Nguyen (VinUniversity) for technical help; Vivien Theissig, Sabine Gruczek, Christina Kasprzak, Lena Reiske and Theres Manthey for assistance with animal care; Klaus Rajewsky (MDC Berlin) for *Cγ1*cre mice and anti-mouse VH12 antibody; Stephen Nutt (WEHI, Australia) for *prdm1*eGFP mice; Kyoko Hayakawa (Fox Chase Cancer Center, USA) for VH11 and Vk14 antibodies. This work was supported by grants from the German Research Foundation LI3540/1-1, 1-4 for A.C.L., TRR130/TP24, Do491/7-5, 10-1, 11-1, 12-1, 14-1 for T.D., SCHR 1658/1-1 for E.S., and MO2934/1-1 for E.M. V.T.H., H.N.D., and L.T.N. were supported by Vingroup Research Fund (Grant no. ISC.19.26). HR-A was supported by the COLCIENCIAS scholarship call 727-2015.

## Author contributions

V.D.D., F.Sz. and A.C.L. developed the study concept. V.D.D., F.Sz., E.M., T.A.L., J.R., A.W., M.F.G., G.M.G., P.D., F.He., H.R.A., A.L.S., E.S., V.T.H., H.N.D., S.O., Q.C., F.Hi., C.H., S.H., M.L., L.T.N., M.F.M., S.Fi., T.D. and A.C.L. performed experiments and contributed to project development. V.D.D., F.Sz. and A.C.L. contributed to the interpretation of the data and wrote the manuscript. All authors read and approved the submitted version of the article.

## Funding

## Competing interests

## Additional information

[1]Deutsches Rheuma-Forschungszentrum, a Leibniz Institute, Charitéplatz 1, 10117 Berlin, Germany. [2]Deparent of Rheumatology and Clinical Immunology, Charité Universitätsmedizin Berlin, corporate member of Freie Universität Berlin and Humboldt-Universität zu Berlin, Berlin, Germany. [3]Faculty of Biology, VNU University of Science, Vietnam National University, Hanoi, Vietnam. [4]Department of Medicine/Nephrology and Medical Intensive Care, Charité-Universitätsmedizin Berlin, corporate member of Freie Universität Berlin and Humboldt-Universität zu Berlin, Berlin, Germany. [5]Grupo de Inmunología Celular e Inmunogenética, Facultad de Medicina, Instituto de Investigaciones Médicas, Universidad de Antioquia UdeA, Medellín, Colombia. [6]Berlin Institute of Health at Charité – Universitätsmedizin Berlin, Berlin, Germany. [7]Vinmec Research Institute of Stem Cell and Gene Technology, College of Health Sciences, VinUniversity, Vinhomes Ocean Park, Hanoi, Vietnam. [8]Intestinal Microbiology Research Group, Department of Molecular Toxicology, German Institute of Human Nutrition Potsdam-Rehbruecke, Nuthetal, Germany. [9]Centre for Musculoskeletal Surgery, Department of Orthopedics, Charité Universitätsmedizin Berlin, Berlin, Germany. [10]Vinmec Health Care System, Hanoi, Vietnam. [11]German Center for Child and Adolescent Health (DZKJ), partner site Berlin, Berlin, Germany. [12]Institut Necker Enfants Malades, INSERM U1151-CNRS UMR 8253, Paris, France. [13]Faculté de Médecine, Université de Paris, Paris, France. [14]AP-HP, Hôpital Necker Enfants Malades, Paris, France. [15]These authors contributed equally: Van Duc Dang, Franziska Szelinski. ✉e-mail: andreialino79@gmail.com

