## [Transparent Peer Review file · Nature Communications]

Distinct Autoreactive CD19⁻ Plasma Cell Subsets Accumulate in Lupus-prone mice

Corresponding Author: Dr Andreia Lino

Version 0:

Reviewer comments:

Reviewer #1

(Remarks to the Author)

In this manuscript, the authors analyzed phenotypes and transcriptomes of plasma cells (PCs) in normal and lupus prone mice. They showed that CD19⁻ PCs are derived from CD19⁺ PCs and accumulate in BM with age in normal mice. In the lupus-prone Sle123 mice, CD19⁻ PCs accumulate in spleen and kidney as well as BM. Single cell transcriptome analysis of PCs from Sle123 mice showed 14 clusters of PCs. One of them expands in sick mice especially in kidney. PCs in this cluster (Cluster 2) are CD19⁻, show gene expression related to lupus and accumulation of somatic mutations in IgV genes, and are clonally different between kidney and spleen/BM. Additionally, this analysis reveals another cluster (Cluster 10) derived from B-1a cells, which expands in BM in sick Sle123 mice. Analysis is nicely performed, and the results are novel and interesting.

1) In the discussion section, the authors suggest a pathogenic role of Cluster 2 because of expression of CXCR3 and other lupus-associated genes. However, these findings are indirect, and more direct evidence such as autoreactivity of these PCs may be required for this conclusion. The authors should discuss this point.

2) Based on the results in this manuscript, Cluster 2 appears to be LLPCs. However, some studies on human SLE suggest a role of SLPCs (doi:10.1038/ni.3175, doi.org/10.1016/j.immuni.2018.08.015). Moreover, a previous study on another lupus-prone mouse line shows that CXCR3⁺ PCs are newly generated PCs but not LLPCs (Reference 53). The authors need to discuss these findings.

3) p.6, l.12-13. The authors mention that MZ, FO, B-2 and B1-b cells could generate CD19⁻ BMPCs. However, the result shows that CD19⁻ BMPCs are derived from both B-1a cells and non-B-1a B cells. There appears no evidence showing generation of CD19⁻ BMPCs from, for example, MZ B cells.

4) p. 6, the last line. The authors mention that CD19⁻ BMPCs are generated by T-dependent and T-independent antigens. Although they show the data on PCs generated by a T-dependent antigen in Fig. 2, I cannot find the data on the T-independent response.

Reviewer #2

(Remarks to the Author)

I consulted B cell immunologist Zhang Biyan (Zhang_Biyan@immunol.a-star.edu.sg) for their expert review of this paper. The following comments were provided.

This interesting study led by Dang et al. sought to characterize the different PC populations based on CD19 expression in naïve, immunized and autoimmune mouse models. The basis of the study is that CD19 CAR-T therapy has been ineffective in clinics and the authors suggest that this could be due to downregulation of CD19 in a further differentiated form of LLPC, resulting in evasion from anti-CD19 therapy and continual maintenance of autoantibody production by this CD19 subset. Indeed, their study had thoroughly characterized the CD19⁻ plasma cell population using conventional plasma cell and plasmablast markers. They also demonstrated using T cell dependent immunization model as well as through staining of B1 cell-associated BCR VH11/Vk14 and VH12 that these CD19⁻ PC can derive from both T cell/GC dependent and T cell independent reactions. Using adoptive transfer model of PC into RAG KO mice, they found that CD19⁻ PC derived unidirectionally from CD19⁺ PC and as such argue that timing of anti CD19 therapy before the majority of CD19⁺ PC turn into CD19⁻ PC could be more efficacious. Single cell analyses of PC as well as B cell clonal analyses revealed interesting

findings that kidneys of lupus mice harbour unique clones of PC of mostly IgA and IgG isotypes while spleen and BM have shared clones of mostly IgM+ PC. Lastly, they found a dominance of B1a cells derived PC in lupus mice among the top 10 PC clones found in bone marrow and spleen which were reactive to T cell independent antigen phosphatidyl choline. These PCs were mostly CD19- in 24 month old mice.

Overall, this is a comprehensive characterization of CD19- PC. The experimental design, methods and statistical analyses used are appropriate to arrive at the conclusions of the paper but there are some concerns that need to be addressed and they are as follows.

Major concerns

- 1) The use of fluorophore conjugated NP with CD138 alone to identify NP-specific PC by flow may lead to false positives. Could the authors also use lambda light chain or two different fluorophore-conjugated NP for more specific identification of NP-specific PC?
- 2) Sca-1 has recently been demonstrated as a reliable marker to identify plasma cell population (Wilmore et al. 2018). Could the authors demonstrate that the CD138+ PC in the paper also express Sca-1?
- 3) Do the two subsets of PC differ in the total amount of Ig secreted? An ELISA measurement of antibodies secreted by in vitro culture of the two different subsets would complement the current set of data.
- 4) In the discussion, it was mentioned that some anti ds-DNA IgG could derive from anti PtC IgM B cells. Indeed, some "natural antibodies" derived from B1 cells have been found to be polyreactive. Could the authors assess if PtC-reactive antibodies secreted by CD19- subset of PCs are polyreactive?
- 5) There is extensive profiling of the CD19- subset of PC in this study. However, there is a lack of functional characterization of the population in relation to disease state of the lupus mice. Also, if the authors could show that specific targeting of CD19- LLPC could be beneficial in the lupus mouse model or perhaps demonstrate, as they purported, that altering the timing of anti CD19 therapy could lead to better disease outcome will further enhance the value of this study.
- 6) CD19- PC is generally understood as a long-lived plasma cell subset. There is lack of reference throughout the paper to many previous studies of LLPC and how this CD19- subset fit into what is known about LLPC in the literature. More of this could be added into the discussion section.

Minor concerns

- 1) There are a number of experiments that had only been done twice. It might be appropriate to do each experiment thrice to ensure reproducibility.
- 2) References are a different Font from the rest of the paper.

Reviewer #3

(Remarks to the Author)

COMMENTS

The authors report a large comprehensive analysis of several plasma cell (PC) subsets in different mouse models. They based their interest on previous findings describing several subsets of PCs both in mice and humans, with different phenotypes, and functions. They argue that new therapies used in autoimmune diseases (systemic lupus) as CD19-directed CAR-T cells when initiated could either miss some pathogenic CD19- PC or impair PCs with regulatory functions. In this view, they first study PCs in BM, spleen, tonsils and blood from human participants, mostly healthy (except for the spleens from patients with immune thrombocytopenia) and confirm the identification of CD19- PCs mostly in BM. Second, they study various lymphoid organs including bone marrow (BM) from C57BL/6 mice at different ages. They use genetic fate mapping systems to follow more precisely CD21+ cells and Ig gamma1 expressing cells. To differentiate B cells responding to T cell dependent and independent antigens, they use NP-KLH immunization. Considering their hypothesis in systemic lupus (SLE), they focus on Sle123 triple congenic mice, and study PCs in non-sick and sick mice, in lymphoid organs and kidney. They found that CD19- BMPCs come from CD19+ cells and increase in numbers with age. These CD19- BMPCs in WT mice originate from diverse B cell subsets, including marginal zone, follicular B cells, B2 and B1-b cells. Moreover, CD19- PCs partly derive from germinal centers (GCs). Using an immunization model with NP-KLH, they demonstrate that CD19- PCs can be IgG+ NP-specific PCs.

Sick Sle123 mice develop more CD19- B cells than their non-sick or WT counterpart. They isolate BM and spleen PCs from sick and non-sick mice, and kidney PCs from sick mice to study heterogeneity of SLE PCs through single cell transcriptomic profiling. They conclude that sick mice are enriched with CD19- PC. Using a transfer model in Rag-/- mice, they conclude that CD19- PC can originate from CD19+ PCs from Sle123 mice.

Through their single cell transcriptomic profiling, CITE-seq, and screening of surface proteins, they differentiate clusters of PCs and based on previous studies, propose links with SLE pathogenesis.

They finally analysed the BCR repertoire in non-sick and sick mice, indicating overabundance or some clones in sick mice, with switch and high mutation rates in affected kidneys. Most clones show CD19 low expression. Looking to the most abundant clones, they identify VH11/Vk14 rearrangements previously associated to the recognition of phosphatidylcholine. Both B cells carrying these rearrangements and recognizing PC were more numerous in CD19- BM PCs in old mice.

Altogether, the main interesting points of this rather heavy analysis in lupus-prone mice are that some PCs can be CD19-, increasing in numbers with age, and more present in mice with lupus, both in lymphoid organs and in organs affected by the disease as kidney. This may explain the failure of targeting CD19 in some patients.

Main points

-This is an important mice study both in wt and lupus-prone mice. It gives very interesting data on the different clusters of PCs and of their kinetics. However, concerning lupus, it remains a mice study and the link with human disease is not obvious. Clearly human data are missing, especially either longitudinal data whatever the authors write in their discussion line 7 ("thus our study allows a longitudinal approach that is not possible in humans") or data in patients treated with anti-

CD19 therapies (Rituximab, CD19-CAR T cells). If this is not possible for the authors, then this is a very elegant mice study with still holes in what happens in patients.

-The interest of phosphatidylcholine specific B cells in the context of SLE is unclear and should be more justified

Minor comments

-Considering that data on lupus come from Sle123 mice, the title should be tempered

-in Methods, in Mouse cell staining for flow cytometry analysis : it should be precise how NP specific PCs and PtC-specific PCs are detected as PCs express usually low levels of surface Ig (surface or intracellular staining ?)

-Figure 1C : IgG Elispot is moderately convincing (lack a positive control to add to supplementary data), also compared to Figure 2I (NP+ IgG Elispot on CD19-)

-Figure 1E : precise the explanation for the lower expression of B220 on CD19- B cells

-As it is shown that CD19- B cells have lower CD19 RNA and a lower level of CD19 protein, instead of a surface downregulation/internalisation of CD19, the authors should develop the reasons of such a decreased transcription

-A general remark is that the manuscript is hard to follow, especially the description of the different PCs clusters

Version 2:

Reviewer comments:

Reviewer #1

(Remarks to the Author)

In the revised manuscript, the authors appropriately addressed the concerns raised by this reviewer.

Reviewer #2

(Remarks to the Author)

The authors have addressed all my major concerns in a satisfactory manner except for the in vivo CD19 depletion experiment to support their hypothesis that altering the timing of anti-CD19 therapy could lead to better disease outcome if their further differentiation into CD19- PC could be prevented.

The attempt to perform in vivo B cell depletion is credible but the authors had been unsuccessful in depleting B cells with anti-mouse CD19 (1D3) alone and thus unable to address this question because of technical issue. In literature, 1D3 had been shown to effectively deplete B cells in mice in combination with other B cell targeting antibodies (Keren, Z., et al. (2011). B-cell depletion reactivates B lymphopoiesis in the BM and rejuvenates the B lineage in aging" Blood 117(11): 3104-3112). It would have been ideal if the authors could prove their hypothesis by performing optimal CD19 depletion in vivo with reference to prior literature. I understand that this is not a straightforward experiment and would require optimization.

Nonetheless, the addition of CD19+ and CD19- PC characterization from human SLE patients into Figure 3A and the extensive characterization of PC in lupus mice may justify publication.

Reviewer #3

(Remarks to the Author)

I read the revised manuscript by Van Duc Dang et al., on CD19- plasmacytes in Lupus, and I thank the authors for their revising effort.

My main remarks were :

-the fact that it is essentially a murine analysis. In the revised form, the authors performed a quantification of CD19- PCs, and determined that SLE patients do have more circulating CD19- PCs.

They analysed CD19- PCs also in patients with haematological malignancies and treated with anti-CD19 CAR T, to find that CD19- B cells were resistant to anti-CD19 CAR T although they do not show the data

They write « This data help to explain how protective antibodies ... remain stable, while autoantibodies are depleted in patients with autoimmune diseases » : I must say that I do not understand completely the link and the sentence in the context of SLE. The idea is better developed in the discussion, and suggest that PC producing antigen specific antibodies are CD19- PC. The authors suggest that early intervention should be relevant. However, concerning B cells, they produce autoantibodies a long time ago before the development of the disease (see Arbuckle MR, N Engl J Med, 2003).

-the specific place of phosphatidylcholine (PtC) autoAbs in the context of SLE. Although convincing, the added references from 27 to 31, concern mice models.

Concerning the other points :

-I maintain that the title suggests Lupus in general (including human) and that there is no demonstration of autoreactive

CD19- PC in human lupus in this manuscript so that the term is a bit large or confusing. The authors extend their results but only to demonstrate the presence of CD19- PC in patients, not their autoreactive features. I understand that the present title is more eye catching

The other points are addressed. I found very interesting the new set of experiments done in Fig4 to explore CD19 regulation in the different clusters, and the conclusion that CD19 downregulation is transcriptionally regulated

A real effort has been made to improve the writing and the understanding of the manuscript

New points

-precise diagnosis of SLE patients (active, quiescent, systemic ?) in Methods

In the new Figure 4, re-precise in the legend the CD19- clusters to facilitate understanding

Point-by-point reply:**Reviewer #1**

In this manuscript, the authors analyzed phenotypes and transcriptomes of plasma cells (PCs) in normal and lupus prone mice. They showed that CD19⁻ PCs are derived from CD19⁺ PCs and accumulate in BM with age in normal mice. In the lupus-prone Sle123 mice, CD19⁻ PCs accumulate in spleen and kidney as well as BM. Single cell transcriptome analysis of PCs from Sle123 mice showed 14 clusters of PCs. One of them expands in sick mice especially in kidney. PCs in this cluster (Cluster 2) are CD19⁻, show gene expression related to lupus and accumulation of somatic mutations in IgV genes, and are clonally different between kidney and spleen/BM. Additionally, this analysis reveals another cluster (Cluster 10) derived from B-1a cells, which expands in BM in sick Sle123 mice. Analysis is nicely performed, and the results are novel and interesting.

We appreciate the reviewer's positive and encouraging feedback on our manuscript. We are pleased to know that the analysis was considered well-performed and the results as both novel and interesting. We sincerely appreciate the time and effort the reviewer invested in evaluating our work. Additionally, we are grateful for their constructive comments, which have contributed to enhancing the quality and clarity of our paper.

1) In the discussion section, the authors suggest a pathogenic role of Cluster 2 because of expression of CXCR3 and other lupus-associated genes. However, these findings are indirect, and more direct evidence such as autoreactivity of these PCs may be required for this conclusion. The authors should discuss this point.

We agree that some findings are indirect. We have further discussed these findings in the revised manuscript. Particularly, we proposed that Cluster 2 might represent a pathogenic population because it is expanded in BM, spleen and kidney of sick mice compared to healthy ones and based on its expression of the inflammatory chemokine receptor CXCR3 and other lupus-associated genes. Supporting this hypothesis, CXCR3⁺ PC subsets have been identified in the bone marrow, spleen, and inflamed kidneys of lupus-prone mice, where they contribute to autoantibody production^{1,2}. Notably, CXCR3 expression on PCs is induced by inflammatory cytokine IFN- γ via a T-bet-dependent mechanism³.

Consistent with these findings, CXCR3⁺ PCs have been shown to accumulate in the synovial fluid of patients with rheumatoid arthritis (RA), suggesting their involvement in disease progression⁴. Moreover, CXCR3⁺ memory B cells are significantly more frequent in patients

with RA compared to healthy donors (HDs), further implicating these cells in RA pathogenesis⁵. In RA, autoreactivity is predominantly associated with CXCR3⁺ memory B cells and plasmablasts⁶. In addition, CXCR3⁺ PCs are found at an elevated frequency in the peripheral blood and lamina propria of patients with ulcerative colitis (UC) compared to HDs⁷.

Furthermore, in the revised version of the manuscript, we analyzed the clusters expression of genes that have been shown in GWAS to be associated with lupus (Fig. 5E). Cluster 2 expressed high levels of genes identified in GWAS as associated to lupus. This association is again indirect but strengthens our argumentation.

Collectively, these findings support the notion that CXCR3 expression may define a subset of PCs with pathogenic roles in SLE. We have incorporated these points into the Discussion section of the revised manuscript to strengthen the interpretation of our results.

2) Based on the results in this manuscript, Cluster 2 appears to be LLPCs. However, some studies on human SLE suggest a role of SLPCs (doi:10.1038/ni.3175, doi.org/10.1016/j.immuni.2018.08.015). Moreover, a previous study on another lupus-prone mouse line shows that CXCR3⁺ PCs are newly generated PCs but not LLPCs (Reference 53). The authors need to discuss these findings.

We thank the reviewer for raising this important point. In the originally submitted manuscript, we highlighted that Cluster 2 exhibits higher expression of MHC-II and B220 (Fig. 5B, D in the revised manuscript) compared to other clusters. Notably, the expression of MHC-II and B220 has been shown to decrease during ASC maturation^{8, 9, 10}. Therefore, it is possible that Cluster 2 includes both long-lived and shorter-lived PCs. This interpretation aligns with previous reports indicating that CXCR3⁺ ASCs are newly generated PCs¹. In our study, we show that LLPC are CD19⁻, but we know that some short-lived PCs can be CD19⁻. From our own data, in the experiments shown in Fig. 2 E-I, in which we immunized mice with NP-KLH and injected BrdU, we also analyzed some mice 6h, and 24h after BrdU and found some short-lived NP-specific cells BrdU⁺ that were CD19⁻ PCs.

Consistent with our findings in mice, studies on human SLE have also suggested a role for short-lived PCs in the pathogenesis of SLE^{11, 12}. We have incorporated these aspects into the revised manuscript, along with the two additional references the reviewer recommended. We apologize for the possible misunderstanding our wording might have caused.

3) p.6, l.12-13. The authors mention that MZ, FO, B-2 and B1-b cells could generate CD19⁻ BMPCs. However, the result shows that CD19⁻ BMPCs are derived from both B-1a cells and

non-B-1a B cells. There appears no evidence showing generation of CD19- BMPCs from, for example, MZ B cells.

We apologize for any confusion that may have arisen on this point. In the originally submitted manuscript, we demonstrated that CD19- plasma cells (PCs) partially originate from B-1a cells, as evidenced by the presence of VH11+/VK14+ cells. To investigate whether other B cell subsets could also contribute to the generation of CD19- PCs, we utilized a genetic fate-mapping system (*Cd21-cre-ROSA26-STOP-eYFP* reporter mouse). This approach was based on the observation that B-1a cells exhibit low CD21 expression compared to marginal zone (MZ) and follicular (FO) B cells from the spleen, as well as B-2 and B-1b cells from the peritoneal cavity^{13, 14}.

Our findings revealed that the nearly all MZ and FO B cells from the spleen, along with B-2 and B-1b cells from the peritoneal cavity, were eYFP+, whereas most B-1 cells from the bone marrow and spleen, as well as B-1a and VH11+/VK14+ B-1a cells from the peritoneal cavity, were eYFP-. Within the CD19- BMPC population, approximately 80% were eYFP+, suggesting that subsets beyond B-1a cells—including MZ, FO, B-2, and B-1b cells—could contribute to the generation of CD19- BMPCs.

We acknowledge that these data are indirect and appreciate the reviewer's feedback on this matter. In response to the reviewer's concern, we have revised the manuscript to clarify this point and adjusted the wording to ensure precision and alignment with the data presented.

We re-analyzed our previously published data using *Cd19cre-Notch2flox/flox* mice¹³. These mice have a specific deficiency in MZ B cells^{13, 15}. Our re-analysis revealed a reduced number of total PCs, as well as CD19+ and CD19- PCs, in the spleen (Fig. R1). These findings suggest that MZ B cells contribute to the CD19- PC pool, at least in the spleen. Unfortunately, we do not have bone marrow data for this model, and this mouse strain is no longer available. For these reasons, we opted not to include this data in the revised manuscript.

Figure R1: Decrease plasma cell subsets in *Cd19cre-Notch2flox/flox* mice

*The graph shows the number of total, CD19+ and CD19- PCs in spleen of Cd19cre-Notch2fl/fl mice compared to the controls. Data show compilation of three independent experiments (n=8 mice). Groups were compared using non-parametric Mann-Whitney (*p < 0.05). Data show mean ± SEM..*

4) p. 6, the last line. The authors mention that CD19- BMPCs are generated by T-dependent and T-independent antigens. Although they show the data on PCs generated by a T-dependent antigen in Fig. 2, I cannot find the data on the T-independent response.

We apologize again for any confusion that may have arisen on this point. In the originally submitted manuscript, we demonstrated the presence of CD19⁻ BMPCs, albeit at a lower frequency, in T cell-deficient mice (*Tcrβδ^{-/-}*). These findings led us to conclude that the generation of CD19⁻ BMPCs also occurs in a T cell-independent manner.

In response to the reviewer's feedback, we have revised the manuscript to improve precision and ensure alignment with the data presented.

Reviewer #2

This interesting study led by Dang et al. sought to characterize the different PC populations based on CD19 expression in naïve, immunized and autoimmune mouse models. The basis of the study is that CD19 CAR-T therapy has been ineffective in clinics and the authors suggest that this could be due to downregulation of CD19 in a further differentiated form of LLPC, resulting in evasion from anti-CD19 therapy and continual maintenance of autoantibody production by this CD19 subset. Indeed, their study had thoroughly characterized the CD19-

plasma cell population using conventional plasma cell and plasmablast markers. They also demonstrated using T cell dependent immunization model as well as through staining of B1 cell-associated BCR VH11/Vk14 and VH12 that these CD19- PC can derive from both T cell/GC dependent and T cell independent reactions. Using adoptive transfer model of PC into RAG KO mice, they found that CD19- PC derived unidirectionally from CD19+ PC and as such argue that timing of anti CD19 therapy before the majority of CD19+ PC turn into CD19- PC could be more efficacious. Single cell analyses of PC as well as B cell clonal analyses revealed interesting findings that kidneys of lupus mice harbour unique clones of PC of mostly IgA and IgG isotypes while spleen and BM have shared clones of mostly IgM+ PC. Lastly, they found a dominance of B1a cells derived PC in lupus mice among the top 10 PC clones found in bone marrow and spleen which were reactive to T cell independent antigen phosphatidyl choline. These PCs were mostly CD19- in 24 month old mice.

Overall, this is a comprehensive characterization of CD19- PC. The experimental design, methods and statistical analyses used are appropriate to arrive at the conclusions of the paper but there are some concerns that need to be addressed and they are as follows.

We thank the reviewer for considering that this manuscript presents a comprehensive characterization of CD19- PCs. We are also grateful to the reviewers for their highly constructive comments and suggestions that have been extremely valuable to clarify important aspects and to better identify the biology of CD19- PCs. This shall undoubtedly further enhance the impact of the study.

Major concerns

1) The use of fluorophore conjugated NP with CD138 alone to identify NP-specific PC by flow may lead to false positives. Could the authors also use lambda light chain or two different fluorophore-conjugated NP for more specific identification of NP-specific PC?

We followed the reviewer's recommendation with additional staining to clarify this point. First, we confirm that all PC subsets (CD19+ and CD19-) were gated within the total PC population (CD138+CD39+) as shown in Supplementary Fig. 1B. For quantifying CD19+ and CD19- PCs within antigen-specific PCs (NP+ or PtC+), our gating strategy involved first selecting total PCs (CD138+CD39+), followed by antigen-specific PCs (NP+ or PtC+), and then identifying CD19+ and CD19- subsets within this population. Conversely, for determining the frequency of antigen-specific PCs (NP+ or PtC+) in CD19+ or CD19- subsets, we first gated total PCs (CD138+CD39+), subsequently separated them into CD19+ and CD19- subsets, and finally assessed antigen specificity (NP+ or PtC+). While we find NP specific cells in the NP-KLH

immunized mice, in the PBS control group these cells are absent. These were indicated on top FACS plots in the main and supplemental figures and figure legends (Fig. 2F and Supplementary Fig. 2E).

Nevertheless, as suggested, we performed λ light chain staining of BM PCs at day 90 after NP-KLH immunization and added a figure (Supplementary Fig. 2E of the revised manuscript and Fig. R2 below) to show that nearly all NP-specific cells are Ig λ +, as established in previous studies^{16, 17, 18, 19}.

Additionally, we utilized ELISPOT assays with NP-OVA-coated plates to quantify NP-specific PCs in NP-KLH-immunized mice, which yielded results consistent with flow cytometry (Fig. 2I). We hope that this clarification adequately addresses the reviewer's concerns.

Figure R2: Lambda light chain is the predominant Ig light chain isotype associated with NP-specific PCs

*Representative FACS plots (left) show NP-specific and Ig λ expression by BMPCs at days 90 after NP-KLH immunization. The graph (right) shows the frequency of Ig λ + PCs in NP+ NP- BMPCs. Data show compilation of two independent experiments (n=5 mice/group). Groups were compared using non-parametric Mann-Whitney (**p < 0.01). Data show mean \pm SEM.*

2) Sca-1 has recently been demonstrated as a reliable marker to identify plasma cell population (Wilmore et al. 2018). Could the authors demonstrate that the CD138+ PC in the paper also express Sca-1?

We thank the reviewer for highlighting the reliability of Sca-1 as a marker for identifying PC populations. Yes, the PCs in the paper are Sca-1+ (Fig. R3). We agree that the combination of CD138 and Sca-1 improves the resolution of PC staining for flow cytometry compared to using CD138 alone. We appreciate the findings by Wilmore et al. and, in fact, have employed this

combination in our lab for a period of time. However, Sca-1 is not an entirely specific marker for PCs, as it is also expressed by hematopoietic progenitor cells and various other immune cell types^{20, 21, 22}. Moreover, the combination of Sca-1 and CD138 has proven less effective in identifying PCs in Peyer's patches⁹ and other lymphoid organs with respect to PC specificity²². To address these limitations, we used splenocytes and bone marrow (BM) cells isolated from Blimp1eGFP reporter mice to screen 255 surface molecules (LEGENDScreen™, BioLegend, Inc.) by flow cytometry. This screen identified several novel markers that are more stable, provide better resolution, and are applicable across various experimental contexts²². In this manuscript, PCs were identified as Blimp1eGFP+CD138+ or Blimp1eGFP+CD39+ for Blimp1eGFP mice, and as CD138+CD39+ for other mouse strains. Importantly, this combination (CD138+CD39+) identifies 99% of PCs, as validated by Blimp1eGFP expression²² (Fig. R3). Notably, CD138+CD39+ PCs in our study also exhibit high levels of Sca-1 expression compared to other cells (Fig. R3).

Figure R3: Combination of CD138 versus CD39 increases the resolution of plasma cell identification by flow cytometry

The flow cytometry plots show the identification of PCs in mouse bone marrow. The graphs show the frequency of Blimp1eGFP+ and Sca-1+ cells in CD138+CD39+ PCs in bone marrow (BM), spleen and mesenteric lymph node (mLN). Data were from two independent experiments (n=7-8 mice/group).

3) Do the two subsets of PC differ in the total amount of Ig secreted? An ELISA measurement of antibodies secreted by in vitro culture of the two different subsets would complement the current set of data.

We thank the reviewer for this suggestion. We have sorted BM CD19⁺ vs CD19⁻ PCs and cultured them in RPMI, LPS or CpG and quantified Igs secreted in culture supernatant by ELISA. In consistent to a slightly increased frequency of IgA PCs in CD19⁻ compared to CD19⁺ counterpart (Fig. 1C), we found a slight increase of secreted IgA by CD19⁻ compared to CD19⁺ PCs (Fig. R4). We included these results in Supplementary Fig. 1E of the revised manuscript.

Figure R4: Antibody secretion by bone marrow CD19⁺ and CD19⁻ plasma cells

The graph shows the concentration of IgA, IgM and IgG in the supernatants at 24 hours after cultured in RPMI, LPS or CpG of sorted BM CD19⁺ or CD19⁻ PCs. IgA, IgM and IgG concentrations were determined by ELISAs. Experiments were performed with naive C57BL/6 mice. Data show compilation of four independent experiments. Data show mean \pm SEM.

Importantly, in the context of lupus, the CD19⁻ PCs contribute more than CD19⁺ PCs to the pool of anti-dsDNA antibodies as we show in Fig. 5A of the revised manuscript.

4) In the discussion, it was mentioned that some anti ds-DNA IgG could derive from anti PtC IgM B cells. Indeed, some “natural antibodies” derived from B1 cells have been found to be polyreactive. Could the authors assess if PtC-reactive antibodies secreted by CD19⁻ subset of PCs are polyreactive?

Indeed, this statement in the discussion is based on a large body of data from the literature showing that natural antibodies produced by B-1 cells can bind a broad range of self-antigens, including phosphatidylcholine (PtC), dsDNA, phosphatidylserine, phosphorylcholine, and other lupus-associated autoantigens^{13, 23, 24, 25, 26}. B-1 cells have been shown to play a critical role in the pathogenesis of SLE^{24, 25}. In our study, anti-PtC PCs predominantly utilized the VH11 gene segment - a germ line gene that has been demonstrated to encode antibodies recognizing dsDNA by multiple independent groups^{27, 28, 29, 30, 31}. Specifically, in PCs from autoimmune FcγRIIB-deficient mice, VH11⁺ cells were found in IgG isotype and exhibited polyreactivity to dsDNA, insulin, LPS, and nucleosomes³².

In humans, PtC-specific antibodies have also been detected in the sera of individuals with autoimmune or infectious diseases^{33, 34, 35}. Notably, patients with antiphospholipid syndrome (APS) show a significant increase in the frequency of PtC-specific B cells, which correlates with elevated serum levels of anti-PtC IgM and IgG as measured by ELISA³⁶. Additionally, SLE patients with hemolytic anemia exhibit higher levels of anti-PtC IgM than those without this hematologic manifestation³⁷. Interestingly, soluble PtC has been shown to inhibit the IgM anti-dsDNA reactivity³⁸, indicating its polyreactive activity, particularly for these two specificities.

Taken together, these findings support that PtC-reactive antibodies are polyreactive and can contribute to autoimmune pathogenesis, including in SLE. To properly acknowledge the contributions of previous research, we have included a discussion of these points in the revised manuscript. At this point, however, we feel that re-visiting the polyreactivity of PtC-specific B cells experimentally would be redundant with previous literature, and take us away from the focus on CD19- PC heterogeneity, their role in autoimmunity at large, and the implication of this heterogeneity to design targeted therapies. The important question of the polyreactivity and potential origin of autoreactive PC emerging during infections or due to microbiota should be addressed in detail in further dedicated studies.

5) There is extensive profiling of the CD19- subset of PC in this study. However, there is a lack of functional characterization of the population in relation to disease state of the lupus mice. Also, if the authors could show that specific targeting of CD19- LLPC could be beneficial in the lupus mouse model or perhaps demonstrate, as they purported, that altering the timing of anti CD19 therapy could lead to better disease outcome will further enhance the value of this study.

We thank the reviewer for this constructive comment on CD19- LLPC functionality.

First, we addressed the role of CD19- LLPCs in relation to SLE. We would like to emphasize that clusters 0, 1, 2, 3, 5, 10, and 11, which are disease-associated clusters, exhibited low expression levels of CD19 (Fig. 3E and Supplementary Fig. 3J). To functionally characterize these clusters in the context of lupus, we sorted CD19- and CD19+ PCs from sick Sle123 mice and quantified anti-dsDNA IgM and IgG antibodies using an ELISPOT assay. Importantly, we observed increased numbers of dsDNA-specific IgM- and IgG-secreting cells in the CD19- PC fraction compared to CD19+ PCs (Fig. R5). This finding supports that CD19- PC have higher pathogenicity than CD19+ PCs. We have included these new results in Fig. 5A of the revised manuscript.

Figure R5: Increased numbers of dsDNA-specific IgM- and IgG-secreting cells in the CD19– PC fraction compared to CD19+ PCs

Graphs show the fold changes in the number of anti-dsDNA IgM+ (left) and IgG+ (right) CD19– relative to CD19+ PCs in sick *Sle123* mice, as measured by ELISPOT assay. Data show compilation of three independent experiments ($n=18$ mice). BM (Bone marrow).

To strengthen the data presented in Fig. 5B-D, which showed that disease-specific clusters express genes and surface markers associated with SLE, we now included the results of an analysis of genes previously reported in SLE genome-wide associated studies (GWAS). This analysis included genes involved in IFN pathway, antigen processing and presentation, NF- κ B regulation and ubiquitin-editing, B cell and Fc receptor signaling, autophagy and vesicle formation. These genes were highly expressed in several PC subsets, such as CD19– cluster 2, a predominant cluster in sick mice (Fig. R6). Furthermore, we found that within same clusters, the expression of SLE GWAS was tendentially higher in the clusters of the sick mice compared to healthy (Fig. R6). We have included these new data in Fig. 5E and Supplementary Fig. 5B of the revised manuscript.

Figure R6: GWAS analysis shows the expression of genes associated with SLE in plasma cells from Sle123 mice

Bubble plot shows the expression of the full list of top 50 mouse orthologs of human genes associated with systemic lupus erythematosus (GWAS, MONDO:0007915), ranked by minimal SNP P-value. Bubble size indicates the percentage of cells in that sample with nonzero expression of the given gene, and bubble color encodes the average scaled expression level.

Second, to test whether early intervention with anti-CD19 therapy improves treatment outcomes we set up an experiment in which we compare responses in young and old Sle123 mice. Our rationale was based on the observation that PCs progressively differentiate into long-lived CD19⁻ PCs over time, and that older mice accumulated CD19⁻ PCs that are resistant to CD19-targeted therapies, while younger mice predominantly harbor CD19⁺ PCs.

To test this and attempt to deplete CD19⁺ cells, which represent CD19⁻ precursors, we injected young and old Sle123 mice intravenously with 250 µg of anti-CD19 antibody (clone 1D3, rat IgG2a, κ) or an isotype control (clone 1D10, rat IgG2a, κ), followed by anti-rat κ light chain (clone MAR 18.5, mouse IgG2a, κ)³⁹. Mice were analyzed at days 7, 14, and 28 after treatment. On day 7, both young and old treated mice seemed to show a marked reduction in CD19⁺ PCs in the bone marrow and spleen compared to control mice (Fig. R7A). However, by day 28, CD19⁺ PC frequencies had returned to levels similar to controls (Fig. R7A).

To assess whether CD19⁺ PCs were truly depleted, we indirectly quantified CD19⁻ PCs. Given that B220 is predominantly expressed by CD19⁺ PCs and nearly absent on CD19⁻ PCs (Fig. 1E and Fig. R7B), we reasoned that if CD19⁺ PCs were completely eliminated, the number of B220⁺ PCs should also be reduced drastically. Contrary to this, we observed comparable levels of B220⁺ PCs in both anti-CD19-treated and control mice, regardless of age or tissue (Fig. R7C), suggesting that CD19⁺ PCs may have downregulated CD19 expression via internalization rather than the cells being eliminated.

To further assess this interpretation, we quantified total PCs. Since young mice primarily possess CD19⁺ PCs, effective depletion would be expected to reduce total PC numbers. However, anti-CD19-treated mice did not show reduced total PC frequencies relative to controls (Fig. R7D). Similarly, B220⁺ B cell frequencies remained unchanged (Fig. R7E). Next, we confirmed that the absence of CD19 detection was not an artifact of antibody clone 1D3, as clone 6D5 yielded consistent results (Fig. R7F). Finally, we quantified immunoglobulin (Ig) levels in the sera of anti-CD19-treated mice before and after treatment and observed no reduction in any Ig isotypes (Fig. R7G). Altogether, these findings indicate that anti-CD19 antibody treatment did not effectively deplete B cells or CD19⁺ plasma cells.

These results are interesting nonetheless since a similar phenomenon of CD19 loss has also been described in malignancies. Those reports and our findings strongly suggest that CD19-targeted therapies may be insufficient due to antigen modulation, and combination therapies are currently under investigation to mitigate this limitation^{40, 41, 42}.

In conclusion, while CD19-targeting antibodies do not deplete CD19⁻ cells, this finding is not directly relevant to our primary research question at this stage. Therefore, we have decided not to include these results in the revised manuscript.

Figure R7: anti-CD19 antibody treatment does not effectively deplete B cells or CD19+ plasma cells

Young and old Sle123 mice were intravenously injected with 250 µg of anti-CD19 antibody (clone 1D3, rat IgG2a, κ) or an isotype control (clone 1D10, rat IgG2a, κ), followed by anti-rat κ light chain (clone MAR 18.5, mouse IgG2a, κ). Mice were subsequently analyzed at days 7, 14, and 28 post-treatment.

(A) The representative FACS plots (left) and the graphs (right) show the frequency of CD19+ PCs in young and old Sle123 mice treated with anti-CD19 antibody or isotype antibody control.

(B) The representative FACS plots show the co-expression of CD19 versus B220 by PCs.

(C) The representative FACS plots (left) and the graphs (right) show the frequency of B220+ PCs in young and old Sle123 mice treated with anti-CD19 antibody or isotype antibody control.

(D) The representative FACS plots (left) and the graphs (right) show the frequency of total PCs in young and old Sle123 mice treated with anti-CD19 antibody or isotype antibody control

(E) The representative FACS plots (left) and the graphs (right) show the frequency of total B cells (B220+) in young and old Sle123 mice treated with anti-CD19 antibody or isotype antibody control

(F) The representative FACS plots show the co-staining of two anti-CD19 antibodies (clone 1D3 and 6D5) for B cells from Sle123 mice treated with anti-CD19 antibody or isotype antibody control.

(G) The graph shows the concentration of immunoglobulin in sera before and after treatment determined by LEGENDplex assay.

Data show compilation of two independent experiments (n=3 to 6 mice per group). Data show mean ± SEM.

The reviewer also raised the question whether (instead of targeting CD19+ cells) targeting CD19- PCs directly could be beneficial. We think that would be an interesting approach. Thus, we screened 255 surface molecules (LEGENDScreen™, BioLegend, Inc.) by flow cytometry to find surface molecules that were expressed by total CD19- PCs (Fig. 1E). Although we found differences between CD19+ and CD19- PCs (Fig. 1E), a pan-specific surface molecule to directly target all CD19- PCs could not be identified.

Instead of depleting the entire CD19⁻ population, a cluster-specific approach might therefore be more promising. Specifically, clusters 2 and 10, which exhibited low CD19 expression and were disease-specific, could represent particularly valuable targets for therapeutic interventions in SLE as we discussed in the Discussion. Implementing, possible therapeutic strategies is a focus of our future research program, while the focus of the present research is

to better delineate PCs clusters, their function and their respective relevance to disease. More systematic functional approaches of cluster depletion will be endeavored in future research.

6) CD19- PC is generally understood as a long-lived plasma cell subset. There is lack of reference throughout the paper to many previous studies of LLPC and how this CD19- subset fit into what is known about LLPC in the literature. More of this could be added into the discussion section.

We thank the reviewer for raising this important point. We have expanded the Discussion section accordingly, to include several recently published studies on long-lived plasma cells (LLPCs), and we discuss how our identified CD19- plasma cell subset aligns with existing knowledge about LLPCs. Furthermore, we provide a comparative analysis of LLPC phenotypes in lupus-prone mouse models, integrating findings from prior reports with our current data to contextualize the relevance and potential implications of the CD19- subset in autoimmune settings.

Minor concerns

1) There are a number of experiments that had only been done twice. It might be appropriate to do each experiment thrice to ensure reproducibility.

We have conducted additional experiments, and the majority have now been performed at least three times. For experiments, which were repeated twice, the results were clear, consistent, and reproducible across both independent replicates. Therefore, a third repetition was not conducted in these cases to adhere to animal welfare considerations, in line with the principles of the 3Rs for responsible animal use.

2) References are a different Font from the rest of the paper.

We apologize for the format issue. We have changed the Font accordingly.

Reviewer #3

The authors report a large comprehensive analysis of several plasma cell (PC) subsets in different mouse models. They based their interest on previous findings describing several subsets of PCs both in mice and humans, with different phenotypes, and functions. They argue that new therapies used in autoimmune diseases (systemic lupus) as CD19-directed CAR-T

cells when initiated could either miss some pathogenic CD19- PC or impair PCs with regulatory functions. In this view, they first study PCs in BM, spleen, tonsils and blood from human participants, mostly healthy (except for the spleens from patients with immune thrombocytopenia) and confirm the identification of CD19- PCs mostly in BM. Second, they study various lymphoid organs including bone marrow (BM) from C57BL/6 mice at different ages. They use genetic fate mapping systems to follow more precisely CD21+ cells and Ig gamma1 expressing cells. To differentiate B cells responding to T cell dependent and independent antigens, they use NP-KLH immunization. Considering their hypothesis in systemic lupus (SLE), they focus on Sle123 triple congenic mice, and study PCs in non-sick and sick mice, in lymphoid organs and kidney.

They found that CD19- BMPCs come from CD19+ cells and increase in numbers with age. These CD19- BMPCs in WT mice originate from diverse B cell subsets, including marginal zone, follicular B cells, B2 and B1-b cells. Moreover, CD19- PCs partly derive from germinal centers (GCs). Using an immunization model with NP-KLH, they demonstrate that CD19- PCs can be IgG+ NP-specific PCs.

Sick Sle123 mice develop more CD19- B cells than their non-sick or WT counterpart. They isolate BM and spleen PCs from sick and non-sick mice, and kidney PCs from sick mice to study heterogeneity of SLE PCs through single cell transcriptomic profiling. They conclude that sick mice are enriched with CD19- PC. Using a transfer model in Rag-/- mice, they conclude that CD19- PC can originate from CD19+ PCs from Sle123 mice.

Through their single cell transcriptomic profiling, CITE-seq, and screening of surface proteins, they differentiate clusters of PCs and based on previous studies, propose links with SLE pathogenesis.

They finally analysed the BCR repertoire in non-sick and sick mice, indicating overabundance or some clones in sick mice, with switch and high mutation rates in affected kidneys. Most clones show CD19 low expression. Looking to the most abundant clones, they identify VH11/Vk14 rearrangements previously associated to the recognition of phosphatidylcholine. Both B cells carrying these rearrangements and recognizing PC were more numerous in CD19- BM PCs in old mice.

Altogether, the main interesting points of this rather heavy analysis in lupus-prone mice are that some PCs can be CD19-, increasing in numbers with age, and more present in mice with lupus, both in lymphoid organs and in organs affected by the disease as kidney. This may explain the failure of targeting CD19 in some patients.

We appreciate the reviewer's positive feedback on our paper. We also thank the reviewer for the suggestions made below that have been important for us to improve our work.

Main points

-This is an important mice study both in wt and lupus-prone mice. It gives very interesting data on the different clusters of PCs and of their kinetics. However, concerning lupus, it remains a mice study and the link with human disease is not obvious. Clearly human data are missing, especially either longitudinal data whatever the authors write in their discussion line 7 ("thus our study allows a longitudinal approach that is not possible in humans") or data in patients treated with anti-CD19 therapies (Rituximab, CD19-CAR T cells). If this is not possible for the authors, then this is a very elegant mice study with still holes in what happens in patients.

We agree with the reviewer that more human data is an added value for this study. Thus, we have performed additional experiments and added a completely new figure - figure 8 in the revised manuscript, which reports human data. We analyzed peripheral blood of SLE patients vs healthy donors for CD19+ and CD19- PCs, and reported that SLE patients have more circulating CD19- PCs than healthy donors (Fig. R8).

In addition, we analyzed peripheral blood and BM PCs of NHL and ALL patients before and 90 days after CD19-directed CAR-T cell therapy. This data clearly show that BM CD19- PCs are resistant to CD19-directed CAR-T cell therapy (Fig. R8). The new human data is reported in Fig. 8 and associated text and indeed gives added relevance to our study.

Our new data helps to explain how protective antibodies (e.g., against tetanus, measles, mumps, rubella, varicella-zoster virus, and Epstein–Barr virus) remain stable, while autoantibodies are depleted in patients with autoimmune diseases^{43, 44}. It also provides insight into why some patients retain certain autoantibodies after CD19-directed CAR-T therapy^{43, 44, 45}.

Although the data have not yet been published, emerging evidence presented at EULAR 2024, ACR 2024, and EULAR 2025 suggests that, while CD19-directed CAR-T cell therapy shows promise for treating SLE, not all patients respond equally well. Specifically, some individuals fail to achieve sustained remission or relapse following initial clinical improvement. The resistance of CD19- PCs to CD19-directed CAR-T cell therapy, as demonstrated in our study, helps explain these challenges and provides insight into mechanisms underlying incomplete therapeutic responses.

Fig. R8. Human CD19- PCs are resistant to CD19-directed CAR-T cell therapy

(A) Gating strategy for identifying peripheral blood CD19⁺ and CD19⁻ PCs (top). Frequencies of CD19⁺ and CD19⁻ PCs among total B cells (including PCs) in peripheral blood from healthy donors (HDs) and patients with SLE (bottom). The BIN channel included CD3, CD14 and live/dead dye.

(B) Experimental workflow illustrating collection of BM samples from patients with acute lymphoblastic leukemia (ALL) and non-Hodgkin lymphoma (NHL), collected pre-apheresis (days -14 to -10) and at day 90 post-CD19-directed CAR-T cell therapy. The experimental workflow was created with BioRender.com.

(C) The FACS plots (top) and graph (bottom) show frequencies of CD19⁺ and CD19⁻ PCs in BM samples of patients with ALL and NHL at pre-apheresis (n=3) and at 90 days after CD19-directed CAR-T cell therapy treatment (n=8). BM samples of both time points were available for three patients only.

(D) Representative FACS plots from Patient 6 showing B cell frequencies in BM and peripheral blood at indicated time points.

Data show compilation of eight HDs and ten SLE (A), and three ALL and five NHL (C). Groups were compared using non-parametric Mann-Whitney test (*p < 0.05, **p < 0.01) (A). Data show mean ± SEM.

-The interest of phosphatidylcholine specific B cells in the context of SLE is unclear and should be more justified

We are interested in phosphatidylcholine (PtC)-specific PCs since these cells utilized VH11 gene segment and are derived from B-1 cells which have been shown to produce autoreactive antibodies and play a critical role in the pathogenesis of SLE^{13, 23, 24, 25, 26}. VH11 is a germ line gene that has been shown to encode antibodies that recognize dsDNA^{27, 28, 29, 30, 31}. In humans, PtC-specific antibodies have also been detected in the sera of individuals with autoimmune and infectious diseases^{33, 34, 35}. Additionally, SLE patients with hemolytic anemia exhibit higher levels of anti-PtC IgM than those without this hematologic manifestation³⁷. Interestingly, soluble PtC has been shown to inhibit the IgM anti-dsDNA reactivity³⁸, indicating its polyreactive activity. Notably, patients with antiphospholipid syndrome (APS) show a significant increase in the frequency of PtC-specific B cells, which correlates with elevated serum levels of anti-PtC IgM and IgG as measured by ELISA³⁶.

Taken together, these findings support that the PtC-specific PCs are polyreactive and might be involved in autoimmune pathogenesis, including SLE. We have included a discussion of these points in the revised manuscript.

Minor comments

1) Considering that data on lupus come from Sle123 mice, the title should be tempered

We thank the reviewer for this valuable suggestion. In response to the reviewer's comments, we have performed additional experiments using human samples and have now incorporated new data from patients with SLE as well as from patients who underwent CD19-CAR-T cell therapy. Given these new and relevant findings, we believe that the original title remains appropriate and would prefer to retain it.

2) in Methods, in Mouse cell staining for flow cytometry analysis : it should be precise how NP specific PCs and PtC-specific PCs are detected as PCs express usually low levels of surface Ig (surface or intracellular staining ?)

We thank the reviewer for raising this important point. We concur that PCs typically express low levels of surface immunoglobulins (Igs), particularly IgG-secreting PCs. In our study, NP- and PtC-specific PCs were identified using phycoerythrin (PE)-conjugated NP and PtC-containing liposomes labeled with Texas Red, respectively, via surface binding. This approach was necessary because NP is conjugated most effectively to PE. However, intracellular staining of PCs with NP-PE indiscriminately labeled all PCs, including those from unimmunized mice, as previously noted by others^{46, 47}. Surface binding of NP-PE proved effective for identifying both IgM⁺ and IgM⁻ PCs following NP-KLH immunization, whereas control mice injected with PBS did not exhibit NP-specific PCs (Fig. 2F of the revised manuscript). This observation is consistent with findings reported by others¹⁸ and was also by our experimental results.

Additionally, the λ light chain can reliably identify NP-specific PCs, as λ light chain is the predominant Ig light chain isotype associated with anti-NP responses in C57BL/6 mice, as established in previous studies^{16, 17, 18, 19}. We also revealed that the majority of NP-specific PCs were Ig λ ⁺, whereas only 6–7% of non-NP-specific PCs expressed Ig λ (Supplementary Fig. 2E of the revised manuscript) via surface binding. We have included these new results in Supplementary Fig. 2E of the revised manuscript. Also, to validate the NP-specific PC population identified by flow cytometry, we complemented this approach with ELISPOT assays using sorted CD19⁺ and CD19⁻ bone marrow PCs from NP-KLH-immunized mice. This analysis confirmed the flow cytometry findings, quantifying NP-specific PCs (Fig. 2I of the revised manuscript).

Similarly, PtC surface binding proved to be a reliable method, as it binds BCRs encoded by the VH11/Vk14 gene segment⁴⁸ which can be detected by surface flow cytometry in both this and an earlier report¹³. Notably, all VH11+/Vk14+ PCs identified via surface staining exclusively represented PtC-specific cells via surface binding (Fig. R9). Moreover, the surface staining of VH11+/Vk14+ BCRs on PCs aligned with independently performed single-cell RNA sequencing data (Fig. 7A, B, E, and F of the revised manuscript). Additionally, the staining patterns for PtC and VH11/Vk14 were comparable between PCs and B-1a cells, further supporting the reliability of this approach.

We have clarified the methods used to detect NP- and PtC-specific PCs in the revised manuscript to address the reviewer's concerns.

Figure R9: Surface staining of VH11Vk14 and PtC by flow cytometry

The flow cytometry plots show the surface staining of VH11VK14 and PtC. The graph shows the frequency of PtC-specific PCs in VH11+Vk14+ PCs. Data were from two independent experiments (n=6 mice).

3) Figure 1C: IgG Elispot is moderately convincing (lack a positive control to add to supplementary data), also compared to Figure 2I (NP+ IgG Elispot on CD19-)

We thank the reviewer for raising this point. As detailed in the Methods section of the originally submitted manuscript, sorted CD19⁺ and CD19⁻ PCs were plated at an initial concentration of 1.5×10^3 cells per 200 μ L per well, followed by four successive five-fold serial dilutions. The ELISPOT images presented in Fig. 1C correspond to the dilution at which the numbers of IgA-, IgM-, and IgG-secreting cells were countable and reflective of their representative frequencies within the sorted PC subpopulations. Since IgG-secreting cells comprise approximately 10–20% of both CD19⁺ and CD19⁻ PCs, the number of IgG-specific spots was less prominent compared to IgA and IgM. In response to the reviewer's comment, we have revised Fig. 1C in the updated manuscript to improve its clarity and representation (Fig. R10).

Figure R10: IgA-, IgM- and IgG-secreting cells in sorted CD19⁻ BMPCs by ELISPOT assay

The ELISPOT pictures show IgA-, IgM- and IgG-secreting cells in sorted CD19⁻ BMPCs. Results are representative of five independent experiments (n=7-15 mice/experiment).

4) Figure 1E : precise the explanation for the lower expression of B220 on CD19⁻ B cells

We thank the reviewer for this correction. In response, we have revised the manuscript to clarify the lower expression levels of both B220 and MHC-II on CD19⁻ PCs. These adjustments are reflected in the revised version.

5) As it is shown that CD19⁻ B cells have lower CD19 RNA and a lower level of CD19 protein, instead of a surface downregulation/internalisation of CD19, the authors should develop the reasons of such a decreased transcription

We appreciate the reviewer's insightful observation regarding the decreased transcription of CD19 in CD19-negative PCs. To address this further and show that CD19 is regulated on transcriptional level rather than internalisation alone, we evaluated the expression levels of a

set of activators and repressors of CD19 expression, as identified in a recent publication⁴⁹, alongside we plotted CD19 RNA and protein levels (Fig. R11).

These observations are consistent with NUDT21's role as a dynamic brake on CD19 expression:

In line with previous data showing⁴⁹ that loss of NUDT21 shortens the CD19 3'UTR, stabilizes its mRNA and elevates surface CD19, we find that NUDT21 levels are lowest in cells with maximal CD19 mRNA and protein, peak in actively proliferating PBs, and then decline again in CD19^{low} cells, suggesting that modulation of NUDT21 expression during B-cell proliferation tightly tunes CD19 output.

The co-expression of both CD19 activators within the same clusters, particularly in CD19^{high} cells (cluster 6, 9 and 7), likely reflects a tightly regulated feedback mechanism to fine-tune CD19 levels. This phenomenon suggests that activators and repressors act together to dynamically balance CD19 expression.

Interestingly in CD19^{low} clusters (cluster 0 and 10), the absence of expression of both activators and repressors likely reflects a distinct cellular identity or developmental stage where CD19 is not required or functionally relevant. Alternatively, these clusters might represent a silenced state for CD19 regulation, where neither activation nor repression pathways are actively engaged. This differential expression pattern underscores the specificity of CD19 regulation to particular cell states and suggests that the coordinated presence of activators and repressors is a hallmark of CD19-positive cells to achieve precise expression control. We added this result in Fig. 3G of the revised manuscript and pointed out that instead of internalisation, the downregulation and differential expression of CD19 across clusters are regulated on transcriptional levels.

Scores for the genset of activators and repressors were added to show an overall trend.

Figure R11: Expression of selected activator and repressor genes associated with CD19 regulation

Bubble plot shows the expression of *Cd19* mRNA (first column) and *CD19* protein (second column), as well as of selected activator and repressor genes associated with *CD19* regulation as published⁴⁹ across the identified clusters. Bubble sizes show the percentage of cells within each cluster expressing the given transcript or protein, and the color scale represents the mean expression level. The two right columns display composite activator and repressor scores for each cluster. Each module score reflects the average scaled expression of the corresponding gene set (activators or repressors) in individual cells.

6) A general remark is that the manuscript is hard to follow, especially the description of the different PCs clusters

We rewrote several parts of the manuscript to increase its readability. We restructured and rewrote the descriptions of the PC clusters completely dedicating one paragraph to each cluster. We think this part has substantially improved. Thus, we would like to thank the reviewer for this suggestion.

Overall, we would like to express our appreciation to the editor, the editorial team, and all the reviewers for their support and valuable suggestions. We believe that we have effectively addressed the previous concerns raised in the manuscript and look forward to your feedback.

Sincerely,

Andreia C. Lino, Ph.D

References

1. Lacotte S, Decossas M, Le Coz C, Brun S, Muller S, Dumortier H. Early differentiated CD138(high) MHCII+ IgG+ plasma cells express CXCR3 and localize into inflamed kidneys of lupus mice. *PLoS One* **8**, e58140 (2013).
2. Rosean TR, Loo W, Erickson LD. Identification of novel plasma cell subsets in a mouse model of SLE. *The Journal of Immunology* **198**, 54.15-54.15 (2017).
3. Serre K, *et al.* CD8 T cells induce T-bet-dependent migration toward CXCR3 ligands by differentiated B cells produced during responses to alum-protein vaccines. *Blood* **120**, 4552-4559 (2012).
4. Tsubaki T, *et al.* Accumulation of plasma cells expressing CXCR3 in the synovial sublining regions of early rheumatoid arthritis in association with production of Mig/CXCL9 by synovial fibroblasts. *Clin Exp Immunol* **141**, 363-371 (2005).
5. Henneken M, Dorner T, Burmester GR, Berek C. Differential expression of chemokine receptors on peripheral blood B cells from patients with rheumatoid arthritis and systemic lupus erythematosus. *Arthritis Res Ther* **7**, R1001-1013 (2005).
6. Reijm S, *et al.* Autoreactive B cells in rheumatoid arthritis include mainly activated CXCR3+ memory B cells and plasmablasts. *JCI Insight* **8**, (2023).
7. Hosomi S, *et al.* Increased numbers of immature plasma cells in peripheral blood specifically overexpress chemokine receptor CXCR3 and CXCR4 in patients with ulcerative colitis. *Clin Exp Immunol* **163**, 215-224 (2011).
8. Tellier J, *et al.* Blimp-1 controls plasma cell function through the regulation of immunoglobulin secretion and the unfolded protein response. *Nat Immunol* **17**, 323-330 (2016).

9. Wilmore JR, Jones DD, Allman D. Protocol for improved resolution of plasma cell subpopulations by flow cytometry. *Eur J Immunol* **47**, 1386-1388 (2017).
10. Koike T, *et al.* Progressive differentiation toward the long-lived plasma cell compartment in the bone marrow. *J Exp Med* **220**, (2023).
11. Tipton CM, *et al.* Diversity, cellular origin and autoreactivity of antibody-secreting cell population expansions in acute systemic lupus erythematosus. *Nat Immunol* **16**, 755-765 (2015).
12. Jenks SA, *et al.* Distinct Effector B Cells Induced by Unregulated Toll-like Receptor 7 Contribute to Pathogenic Responses in Systemic Lupus Erythematosus. *Immunity* **49**, 725-739 e726 (2018).
13. Lino AC, *et al.* LAG-3 Inhibitory Receptor Expression Identifies Immunosuppressive Natural Regulatory Plasma Cells. *Immunity* **49**, 120-133 e129 (2018).
14. Tung JW, Herzenberg LA. Unraveling B-1 progenitors. *Curr Opin Immunol* **19**, 150-155 (2007).
15. Saito T, *et al.* Notch2 is preferentially expressed in mature B cells and indispensable for marginal zone B lineage development. *Immunity* **18**, 675-685 (2003).
16. Allen D, *et al.* Timing, genetic requirements and functional consequences of somatic hypermutation during B-cell development. *Immunol Rev* **96**, 5-22 (1987).
17. Jacob J, Kelsoe G. In situ studies of the primary immune response to (4-hydroxy-3-nitrophenyl)acetyl. II. A common clonal origin for periarteriolar lymphoid sheath-associated foci and germinal centers. *J Exp Med* **176**, 679-687 (1992).
18. Blanc P, *et al.* Mature IgM-expressing plasma cells sense antigen and develop competence for cytokine production upon antigenic challenge. *Nat Commun* **7**, 13600 (2016).
19. Robinson MJ, *et al.* Long-lived plasma cells accumulate in the bone marrow at a constant rate from early in an immune response. *Sci Immunol* **7**, eabm8389 (2022).
20. Bradfute SB, Graubert TA, Goodell MA. Roles of Sca-1 in hematopoietic stem/progenitor cell function. *Exp Hematol* **33**, 836-843 (2005).

21. Morcos MNF, *et al.* SCA-1 Expression Level Identifies Quiescent Hematopoietic Stem and Progenitor Cells. *Stem Cell Reports* **8**, 1472-1478 (2017).
22. Dang VD, *et al.* CD39 and CD326 Are Bona Fide Markers of Murine and Human Plasma Cells and Identify a Bone Marrow Specific Plasma Cell Subpopulation in Lupus. *Front Immunol* **13**, 873217 (2022).
23. Zhong X, *et al.* A novel subpopulation of B-1 cells is enriched with autoreactivity in normal and lupus-prone mice. *Arthritis Rheum* **60**, 3734-3743 (2009).
24. Ottens K, Schneider J, Satterthwaite AB. B-1a Cells, but Not Marginal Zone B Cells, Are Implicated in the Accumulation of Autoreactive Plasma Cells in Lyn^{-/-} Mice. *Immunohorizons* **8**, 47-56 (2024).
25. Ma K, *et al.* B1-cell-produced anti-phosphatidylserine antibodies contribute to lupus nephritis development via TLR-mediated Syk activation. *Cell Mol Immunol* **20**, 881-894 (2023).
26. Gronwall C, Vas J, Silverman GJ. Protective Roles of Natural IgM Antibodies. *Front Immunol* **3**, 66 (2012).
27. Pewzner-Jung Y, Simon T, Eilat D. Structural elements controlling anti-DNA antibody affinity and their relationship to anti-phosphorylcholine activity. *J Immunol* **156**, 3065-3073 (1996).
28. Behar SM, Scharff MD. Somatic diversification of the S107 (T15) VH11 germ-line gene that encodes the heavy-chain variable region of antibodies to double-stranded DNA in (NZB x NZW)F1 mice. *Proc Natl Acad Sci U S A* **85**, 3970-3974 (1988).
29. Behar SM, Lustgarten DL, Corbet S, Scharff MD. Characterization of somatically mutated S107 VH11-encoded anti-DNA autoantibodies derived from autoimmune (NZB x NZW)F1 mice. *J Exp Med* **173**, 731-741 (1991).
30. Friedmann D, *et al.* Production of high affinity autoantibodies in autoimmune New Zealand Black/New Zealand white F1 mice targeted with an anti-DNA heavy chain. *J Immunol* **162**, 4406-4416 (1999).
31. Pewzner-Jung Y, Friedmann D, Sonoda E, Jung S, Rajewsky K, Eilat D. B cell deletion, anergy, and receptor editing in "knock in" mice targeted with a germline-encoded or somatically mutated anti-DNA heavy chain. *J Immunol* **161**, 4634-4645 (1998).
32. Tiller T, *et al.* Development of self-reactive germinal center B cells and plasma cells in autoimmune Fc gammaRIIB-deficient mice. *J Exp Med* **207**, 2767-2778 (2010).

33. Cabral AR, Cabiedes J, Alarcon-Segovia D. Hemolytic anemia related to an IgM autoantibody to phosphatidylcholine that binds in vitro to stored and to bromelain-treated human erythrocytes. *J Autoimmun* **3**, 773-787 (1990).
34. Abuaf N, *et al.* Autoantibodies to phospholipids and to the coagulation proteins in AIDS. *Thromb Haemost* **77**, 856-861 (1997).
35. Casao MA, Leiva J, Diaz R, Gamazo C. Anti-phosphatidylcholine antibodies in patients with brucellosis. *J Med Microbiol* **47**, 49-54 (1998).
36. Nitschke E, *et al.* Phosphatidylcholine-specific B cells are enriched among atypical CD11c(high) and CD21(low) memory B cells in antiphospholipid syndrome. *Front Immunol* **16**, 1585953 (2025).
37. Guzman J, Cabral AR, Cabiedes J, Pita-Ramirez L, Alarcon-Segovia D. Antiphospholipid antibodies in patients with idiopathic autoimmune haemolytic anemia. *Autoimmunity* **18**, 51-56 (1994).
38. Cabiedes J, Cabral AR, Lopez-Mendoza AT, Cordero-Esperon HA, Huerta MT, Alarcon-Segovia D. Characterization of anti-phosphatidylcholine polyreactive natural autoantibodies from normal human subjects. *J Autoimmun* **18**, 181-190 (2002).
39. Keren Z, *et al.* B-cell depletion reactivates B lymphopoiesis in the BM and rejuvenates the B lineage in aging. *Blood* **117**, 3104-3112 (2011).
40. Spiegel JY, *et al.* CAR T cells with dual targeting of CD19 and CD22 in adult patients with recurrent or refractory B cell malignancies: a phase 1 trial. *Nat Med* **27**, 1419-1431 (2021).
41. Cordoba S, *et al.* CAR T cells with dual targeting of CD19 and CD22 in pediatric and young adult patients with relapsed or refractory B cell acute lymphoblastic leukemia: a phase 1 trial. *Nat Med* **27**, 1797-1805 (2021).
42. Koh SK, *et al.* Anti-CD19 antibody cotreatment enhances serial killing activity of anti-CD19 CAR-T/-NK cells and reduces trogocytosis. *Blood* **145**, 956-969 (2025).
43. Mackensen A, *et al.* Anti-CD19 CAR T cell therapy for refractory systemic lupus erythematosus. *Nat Med* **28**, 2124-2132 (2022).
44. Muller F, *et al.* CD19 CAR T-Cell Therapy in Autoimmune Disease - A Case Series with Follow-up. *N Engl J Med* **390**, 687-700 (2024).

45. Nunez D, *et al.* Cytokine and reactivity profiles in SLE patients following anti-CD19 CART therapy. *Mol Ther Methods Clin Dev* **31**, 101104 (2023).
46. Kim MS, Kim TS. R-phycoerythrin-conjugated antibodies are inappropriate for intracellular staining of murine plasma cells. *Cytometry A* **83**, 452-460 (2013).
47. Renner P, Crone M, Kornas M, Pioli KT, Pioli PD. Intracellular flow cytometry staining of antibody-secreting cells using phycoerythrin-conjugated antibodies: pitfalls and solutions. *Antib Ther* **5**, 151-163 (2022).
48. Hardy RR, Wei CJ, Hayakawa K. Selection during development of VH11+ B cells: a model for natural autoantibody-producing CD5+ B cells. *Immunol Rev* **197**, 60-74 (2004).
49. Witkowski MT, *et al.* NUDT21 limits CD19 levels through alternative mRNA polyadenylation in B cell acute lymphoblastic leukemia. *Nat Immunol* **23**, 1424-1432 (2022).

Point-by-point reply:**Reviewer #1**

In this manuscript, the authors analyzed phenotypes and transcriptomes of plasma cells (PCs) in normal and lupus prone mice. They showed that CD19⁻ PCs are derived from CD19⁺ PCs and accumulate in BM with age in normal mice. In the lupus-prone Sle123 mice, CD19⁻ PCs accumulate in spleen and kidney as well as BM. Single cell transcriptome analysis of PCs from Sle123 mice showed 14 clusters of PCs. One of them expands in sick mice especially in kidney. PCs in this cluster (Cluster 2) are CD19⁻, show gene expression related to lupus and accumulation of somatic mutations in IgV genes, and are clonally different between kidney and spleen/BM. Additionally, this analysis reveals another cluster (Cluster 10) derived from B-1a cells, which expands in BM in sick Sle123 mice. Analysis is nicely performed, and the results are novel and interesting.

We appreciate the reviewer's positive and encouraging feedback on our manuscript. We are pleased to know that the analysis was considered well-performed and the results as both novel and interesting. We sincerely appreciate the time and effort the reviewer invested in evaluating our work. Additionally, we are grateful for their constructive comments, which have contributed to enhancing the quality and clarity of our paper.

1) In the discussion section, the authors suggest a pathogenic role of Cluster 2 because of expression of CXCR3 and other lupus-associated genes. However, these findings are indirect, and more direct evidence such as autoreactivity of these PCs may be required for this conclusion. The authors should discuss this point.

We agree that some findings are indirect. We have further discussed these findings in the revised manuscript. Particularly, we proposed that Cluster 2 might represent a pathogenic population because it is expanded in BM, spleen and kidney of sick mice compared to healthy ones and based on its expression of the inflammatory chemokine receptor CXCR3 and other lupus-associated genes. Supporting this hypothesis, CXCR3⁺ PC subsets have been identified in the bone marrow, spleen, and inflamed kidneys of lupus-prone mice, where they contribute to autoantibody production^{1,2}. Notably, CXCR3 expression on PCs is induced by inflammatory cytokine IFN- γ via a T-bet–dependent mechanism³.

Consistent with these findings, CXCR3⁺ PCs have been shown to accumulate in the synovial fluid of patients with rheumatoid arthritis (RA), suggesting their involvement in disease progression⁴. Moreover, CXCR3⁺ memory B cells are significantly more frequent in patients

with RA compared to healthy donors (HDs), further implicating these cells in RA pathogenesis⁵. In RA, autoreactivity is predominantly associated with CXCR3⁺ memory B cells and plasmablasts⁶. In addition, CXCR3⁺ PCs are found at an elevated frequency in the peripheral blood and lamina propria of patients with ulcerative colitis (UC) compared to HDs⁷.

Furthermore, in the revised version of the manuscript, we analyzed the clusters expression of genes that have been shown in GWAS to be associated with lupus (Fig. 6E and supplementary Fig. 6B). Cluster 2 expressed high levels of genes identified in GWAS as associated to lupus. This association is again indirect but strengthens our argumentation.

Collectively, these findings support the notion that CXCR3 expression may define a subset of PCs with pathogenic roles in SLE. We have incorporated these points into the Discussion section of the revised manuscript to strengthen the interpretation of our results.

2) Based on the results in this manuscript, Cluster 2 appears to be LLPCs. However, some studies on human SLE suggest a role of SLPCs ([doi:10.1038/nr3175](https://doi.org/10.1038/nr3175), doi.org/10.1016/j.immuni.2018.08.015). Moreover, a previous study on another lupus-prone mouse line shows that CXCR3⁺ PCs are newly generated PCs but not LLPCs (Reference 53). The authors need to discuss these findings.

We thank the reviewer for raising this important point. In the originally submitted manuscript, we highlighted that Cluster 2 exhibits higher expression of MHC-II and B220 (Fig. 6B, D in the revised manuscript) compared to other clusters. Notably, the expression of MHC-II and B220 has been shown to decrease during ASC maturation⁸⁻¹⁰. Therefore, it is possible that Cluster 2 includes both long-lived and shorter-lived PCs. This interpretation aligns with previous reports indicating that CXCR3⁺ ASCs are newly generated PCs¹. In our study, we show that LLPC are CD19⁻, but we know that some short-lived PCs can be CD19⁻. From our own data, in the experiments shown in Fig. 2 E-I, in which we immunized mice with NP-KLH and injected BrdU, we also analyzed some mice 6h, and 24h after BrdU and found some short-lived NP-specific cells BrdU⁺ that were CD19⁻ PCs.

Consistent with our findings in mice, studies on human SLE have also suggested a role for short-lived PCs in the pathogenesis of SLE^{11,12}. We have incorporated these aspects into the revised manuscript, along with the two additional references the reviewer recommended. We apologize for the possible misunderstanding our wording might have caused.

3) p.6, l.12-13. The authors mention that MZ, FO, B-2 and B1-b cells could generate CD19⁻ BMPCs. However, the result shows that CD19⁻ BMPCs are derived from both B-1a cells and

non-B-1a B cells. There appears no evidence showing generation of CD19- BMPCs from, for example, MZ B cells.

We apologize for any confusion that may have arisen on this point. In the originally submitted manuscript, we demonstrated that CD19- plasma cells (PCs) partially originate from B-1a cells, as evidenced by the presence of VH11+/VK14+ cells. To investigate whether other B cell subsets could also contribute to the generation of CD19- PCs, we utilized a genetic fate-mapping system (*Cd21-cre-ROSA26-STOP-eYFP* reporter mouse). This approach was based on the observation that B-1a cells exhibit low CD21 expression compared to marginal zone (MZ) and follicular (FO) B cells from the spleen, as well as B-2 and B-1b cells from the peritoneal cavity^{13,14}.

Our findings revealed that nearly all MZ and FO B cells from the spleen, along with B-2 and B-1b cells from the peritoneal cavity, were eYFP+, whereas most B-1 cells from the bone marrow and spleen, as well as B-1a and VH11+/VK14+ B-1a cells from the peritoneal cavity, were eYFP-. Within the CD19- BMPC population, approximately 80% were eYFP+, suggesting that subsets beyond B-1a cells—including MZ, FO, B-2, and B-1b cells—could contribute to the generation of CD19- BMPCs.

We acknowledge that these data are indirect and appreciate the reviewer's feedback on this matter. In response to the reviewer's concern, we have revised the manuscript to clarify this point and adjusted the wording to ensure precision and alignment with the data presented.

We re-analyzed our previously published data using *Cd19cre-Notch2flox/flox* mice¹³. These mice have a specific deficiency in MZ B cells^{13,15}. Our re-analysis revealed a reduced number of total PCs, as well as CD19+ and CD19- PCs, in the spleen (Fig. R1). These findings suggest that MZ B cells contribute to the CD19- PC pool, at least in the spleen. Unfortunately, we do not have bone marrow data for this model, and this mouse strain is no longer available. For these reasons, we opted not to include this data in the revised manuscript.

Figure R1: Decrease plasma cell subsets in *Cd19cre-Notch2flox/flox* mice

*The graph shows the number of total, CD19+ and CD19- PCs in spleen of Cd19cre-Notch2fl/fl mice compared to the controls. Data show compilation of three independent experiments (n=8 mice). Groups were compared using non-parametric Mann-Whitney (*p < 0.05). Data show mean ± SEM..*

4) p. 6, the last line. The authors mention that CD19- BMPCs are generated by T-dependent and T-independent antigens. Although they show the data on PCs generated by a T-dependent antigen in Fig. 2, I cannot find the data on the T-independent response.

We apologize again for any confusion that may have arisen on this point. In the originally submitted manuscript, we demonstrated the presence of CD19⁻ BMPCs, albeit at a lower frequency, in T cell-deficient mice (*Tcrβδ^{-/-}*). These findings led us to conclude that the generation of CD19⁻ BMPCs also occurs in a T cell-independent manner.

In response to the reviewer's feedback, we have revised the manuscript to improve precision and ensure alignment with the data presented.

Reviewer #2

This interesting study led by Dang et al. sought to characterize the different PC populations based on CD19 expression in naïve, immunized and autoimmune mouse models. The basis of the study is that CD19 CAR-T therapy has been ineffective in clinics and the authors suggest that this could be due to downregulation of CD19 in a further differentiated form of LLPC, resulting in evasion from anti-CD19 therapy and continual maintenance of autoantibody production by this CD19 subset. Indeed, their study had thoroughly characterized the CD19-

plasma cell population using conventional plasma cell and plasmablast markers. They also demonstrated using T cell dependent immunization model as well as through staining of B1 cell-associated BCR VH11/Vk14 and VH12 that these CD19- PC can derive from both T cell/GC dependent and T cell independent reactions. Using adoptive transfer model of PC into RAG KO mice, they found that CD19- PC derived unidirectionally from CD19+ PC and as such argue that timing of anti CD19 therapy before the majority of CD19+ PC turn into CD19- PC could be more efficacious. Single cell analyses of PC as well as B cell clonal analyses revealed interesting findings that kidneys of lupus mice harbour unique clones of PC of mostly IgA and IgG isotypes while spleen and BM have shared clones of mostly IgM+ PC. Lastly, they found a dominance of B1a cells derived PC in lupus mice among the top 10 PC clones found in bone marrow and spleen which were reactive to T cell independent antigen phosphatidyl choline. These PCs were mostly CD19- in 24 month old mice.

Overall, this is a comprehensive characterization of CD19- PC. The experimental design, methods and statistical analyses used are appropriate to arrive at the conclusions of the paper but there are some concerns that need to be addressed and they are as follows.

We thank the reviewer for considering that this manuscript presents a comprehensive characterization of CD19- PCs. We are also grateful to the reviewers for their highly constructive comments and suggestions that have been extremely valuable to clarify important aspects and to better identify the biology of CD19- PCs. This shall undoubtedly further enhance the impact of the study.

Major concerns

1) The use of fluorophore conjugated NP with CD138 alone to identify NP-specific PC by flow may lead to false positives. Could the authors also use lambda light chain or two different fluorophore-conjugated NP for more specific identification of NP-specific PC?

We followed the reviewer's recommendation with additional staining to clarify this point. First, we confirm that all PC subsets (CD19+ and CD19-) were gated within the total PC population (CD138+CD39+) as shown in Supplementary Fig. 1B. For quantifying CD19+ and CD19- PCs within antigen-specific PCs (NP+ or PtC+), our gating strategy involved first selecting total PCs (CD138+CD39+), followed by antigen-specific PCs (NP+ or PtC+), and then identifying CD19+ and CD19- subsets within this population. Conversely, for determining the frequency of antigen-specific PCs (NP+ or PtC+) in CD19+ or CD19- subsets, we first gated total PCs (CD138+CD39+), subsequently separated them into CD19+ and CD19- subsets, and finally assessed antigen specificity (NP+ or PtC+). While we find NP specific cells in the NP-KLH

immunized mice, in the PBS control group these cells are absent. These were indicated on top FACS plots in the main and supplemental figures and figure legends (Fig. 2F and Supplementary Fig. 2E).

Nevertheless, as suggested, we performed λ light chain staining of BM PCs at day 90 after NP-KLH immunization and added a figure (Supplementary Fig. 2E of the revised manuscript and Fig. R2 below) to show that nearly all NP-specific cells are Ig λ +, as established in previous studies¹⁶⁻¹⁹.

Additionally, we utilized ELISPOT assays with NP-OVA-coated plates to quantify NP-specific PCs in NP-KLH-immunized mice, which yielded results consistent with flow cytometry (Fig. 2I). We hope that this clarification adequately addresses the reviewer's concerns.

Figure R2: Lambda light chain is the predominant Ig light chain isotype associated with NP-specific PCs

*Representative FACS plots (left) show NP-specific and Ig λ expression by BMPCs at days 90 after NP-KLH immunization. The graph (right) shows the frequency of Ig λ + PCs in NP+ NP- BMPCs. Data show compilation of two independent experiments (n=5 mice/group). Groups were compared using non-parametric Mann-Whitney (**p < 0.01). Data show mean \pm SEM.*

2) Sca-1 has recently been demonstrated as a reliable marker to identify plasma cell population (Wilmore et al. 2018). Could the authors demonstrate that the CD138+ PC in the paper also express Sca-1?

We thank the reviewer for highlighting the reliability of Sca-1 as a marker for identifying PC populations. Yes, the PCs in the paper are Sca-1+ (Fig. R3). We agree that the combination of CD138 and Sca-1 improves the resolution of PC staining for flow cytometry compared to using CD138 alone. We appreciate the findings by Wilmore et al. and, in fact, have employed this

combination in our lab for a period of time. However, Sca-1 is not an entirely specific marker for PCs, as it is also expressed by hematopoietic progenitor cells and various other immune cell types²⁰⁻²². Moreover, the combination of Sca-1 and CD138 has proven less effective in identifying PCs in Peyer's patches⁹ and other lymphoid organs with respect to PC specificity²². To address these limitations, we used splenocytes and bone marrow (BM) cells isolated from Blimp1eGFP reporter mice to screen 255 surface molecules (LEGENDScreen™, BioLegend, Inc.) by flow cytometry. This screen identified several novel markers that are more stable, provide better resolution, and are applicable across various experimental contexts²². In this manuscript, PCs were identified as Blimp1eGFP+CD138+ or Blimp1eGFP+CD39+ for Blimp1eGFP mice, and as CD138+CD39+ for other mouse strains. Importantly, this combination (CD138+CD39+) identifies 99% of PCs, as validated by Blimp1eGFP expression²² (Fig. R3). Notably, CD138+CD39+ PCs in our study also exhibit high levels of Sca-1 expression compared to other cells (Fig. R3).

Figure R3: Combination of CD138 versus CD39 increases the resolution of plasma cell identification by flow cytometry

The flow cytometry plots show the identification of PCs in mouse bone marrow. The graphs show the frequency of Blimp1eGFP+ and Sca-1+ cells in CD138+CD39+ PCs in bone marrow (BM), spleen and mesenteric lymph node (mLN). Data were from two independent experiments (n=7-8 mice/group).

3) Do the two subsets of PC differ in the total amount of Ig secreted? An ELISA measurement of antibodies secreted by in vitro culture of the two different subsets would complement the current set of data.

We thank the reviewer for this suggestion. We have sorted BM CD19⁺ vs CD19⁻ PCs and cultured them in RPMI, LPS or CpG and quantified Igs secreted in culture supernatant by ELISA. We found a slightly increased frequency of IgA PCs in CD19⁻ compared to CD19⁺ counterpart (Fig. 1C), and a slight increase of secreted IgA by CD19⁻ compared to CD19⁺ PCs (Fig. R4). We included these results in Supplementary Fig. 1E of the revised manuscript.

Figure R4: Antibody secretion by bone marrow CD19⁺ and CD19⁻ plasma cells

The graph shows the concentration of IgA, IgM and IgG in the supernatants at 24 hours after cultured in RPMI, LPS or CpG of sorted BM CD19⁺ or CD19⁻ PCs. IgA, IgM and IgG concentrations were determined by ELISAs. Experiments were performed with naive C57BL/6 mice. Data show compilation of four independent experiments. Data show mean \pm SEM.

Importantly, in the context of lupus, the CD19⁻ PCs contribute more than CD19⁺ PCs to the pool of anti-dsDNA antibodies as we show in Fig. 6A of the revised manuscript.

4) In the discussion, it was mentioned that some anti ds-DNA IgG could derive from anti PtC IgM B cells. Indeed, some “natural antibodies” derived from B1 cells have been found to be polyreactive. Could the authors assess if PtC-reactive antibodies secreted by CD19⁻ subset of PCs are polyreactive?

Yes, natural antibodies produced by B-1 cells can bind a broad range of self-antigens, including phosphatidylcholine (PtC), dsDNA, phosphatidylserine, phosphorylcholine, and other lupus-associated autoantigens^{13,23-26}. B-1 cells have been shown to play a critical role in the pathogenesis of SLE^{24,25}. In our study, anti-PtC PCs predominantly utilized the VH11 gene segment - a germ line gene that has been demonstrated to encode antibodies recognizing dsDNA by multiple independent groups²⁷⁻³¹. Specifically, in PCs from autoimmune FcγRIIB-deficient mice, VH11⁺ cells were found in IgG isotype and exhibited polyreactivity to dsDNA, insulin, LPS, and nucleosomes³².

In humans, PtC-specific antibodies have also been detected in the sera of individuals with autoimmune or infectious diseases³³⁻³⁵. Notably, patients with antiphospholipid syndrome (APS) show a significant increase in the frequency of PtC-specific B cells, which correlates with elevated serum levels of anti-PtC IgM and IgG as measured by ELISA³⁶. Additionally, SLE patients with hemolytic anemia exhibit higher levels of anti-PtC IgM than those without this hematologic manifestation³⁷. Interestingly, soluble PtC has been shown to inhibit the IgM anti-dsDNA reactivity³⁸, indicating its polyreactive activity, particularly for these two specificities.

Taken together, these findings support that PtC-reactive antibodies are polyreactive and can contribute to autoimmune pathogenesis, including in SLE. To properly acknowledge the contributions of previous research, we have included a discussion of these points in the revised manuscript. At this point, however, we feel that re-visiting the polyreactivity of PtC-specific B cells experimentally would be redundant with previous literature, and take us away from the focus on CD19- PC heterogeneity, their role in autoimmunity at large, and the implication of this heterogeneity to design targeted therapies. The important question of the polyreactivity and potential origin of autoreactive PC emerging during infections or due to microbiota should be addressed in detail in further dedicated studies.

5) There is extensive profiling of the CD19- subset of PC in this study. However, there is a lack of functional characterization of the population in relation to disease state of the lupus mice. Also, if the authors could show that specific targeting of CD19- LLPC could be beneficial in the lupus mouse model or perhaps demonstrate, as they purported, that altering the timing of anti CD19 therapy could lead to better disease outcome will further enhance the value of this study.

We thank the reviewer for this constructive comment. To address the first part of this question about how CD19- LLPCs relate to SLE, we would like to emphasize that clusters 0, 1, 2, 3, 5, 10, and 11, which are disease-specific clusters, exhibited low expression levels of CD19 (Fig. 4A and Supplementary Fig. 4A). To further functionally characterize these clusters in the context of lupus, we sorted CD19- and CD19+ PCs from sick Sle123 mice and quantified anti-dsDNA IgM and IgG antibodies using an ELISPOT assay. Notably, we observed increased numbers of dsDNA-specific IgM- and IgG-secreting cells in the CD19- PC fraction compared to CD19+ PCs (Fig. R5). This finding supports that CD19- PC have higher pathogenicity than CD19+ PCs. We have included these new results in Fig. 6A of the revised manuscript.

Figure R5: Increased numbers of dsDNA-specific IgM- and IgG-secreting cells in the CD19– PC fraction compared to CD19+ PCs

Graphs show the fold changes in the number of anti-dsDNA IgM+ (left) and IgG+ (right) CD19– relative to CD19+ PCs in sick *Sle123* mice, as measured by ELISPOT assay. Data show compilation of three independent experiments ($n=18$ mice). BM (Bone marrow).

To strengthen the data presented in Fig. 6B-D in which we showed disease-specific clusters expressing genes and surface markers involved in SLE, we now also included analysis of genes previously reported in SLE genome-wide associated studies (GWAS) such as IFN related genes, genes associated with antigen processing and presentation, NF- κ B regulators and ubiquitin-editing, B cell and Fc receptor signaling, autophagy and vesicle formation. These genes were highly expressed in several PC subsets, such as CD19– cluster 2, a predominant cluster in sick mice (Fig. R6). Furthermore, we found that within same clusters, the expression of SLE GWAS was tendentially higher in the clusters of the sick mice compared to healthy (Fig. R6).

We have included these new data in Fig. 6E and Supplementary Fig. 6B of the revised manuscript.

Figure R6: GWAS analysis shows the expression of genes associated with SLE in plasma cells from Sle123 mice

Bubble plot shows the expression of the full list of top 50 mouse orthologs of human genes associated with systemic lupus erythematosus (GWAS, MONDO:0007915), ranked by minimal SNP P-value. Bubble size indicates the percentage of cells in that sample with nonzero expression of the given gene, and bubble color encodes the average scaled expression level.

To test whether early intervention with anti-CD19 therapy improves treatment outcomes we set up an experiment in which we compare responses in young and old Sle123 mice. Our rationale was based on the observation that PCs progressively differentiate into long-lived CD19⁻ PCs over time, and that older mice accumulated CD19⁻ PCs that are resistant to CD19-targeted therapies, while younger mice predominantly harbor CD19⁺ PCs.

To test this and attempt to deplete CD19⁺ cells, which represent CD19⁻ precursors, we intravenously injected young and old Sle123 mice with 250 µg of anti-CD19 antibody (clone 1D3, rat IgG2a, κ) or an isotype control (clone 1D10, rat IgG2a, κ), followed by anti-rat κ light chain (clone MAR 18.5, mouse IgG2a, κ)³⁹. Mice were analyzed at days 7, 14, and 28 after treatment. At day 7, both young and old treated mice seemed to show a marked reduction in CD19⁺ PCs in the bone marrow and spleen compared to control mice (Fig. R7A). However, by day 28, CD19⁺ PC frequencies had returned to levels similar to controls (Fig. R7A).

To assess whether CD19⁺ PCs were truly depleted, we indirectly quantified CD19⁻ PCs. Given that B220 is predominantly expressed by CD19⁺ PCs and nearly absent on CD19⁻ PCs (Fig. 1E and Fig. R7B), we reasoned that if CD19⁺ PCs were completely eliminated, the number of B220⁺ PCs should also be reduced drastically. Contrary to this, we observed comparable levels of B220⁺ PCs in both anti-CD19-treated and control mice, regardless of age or tissue (Fig. R7C), suggesting that CD19⁺ PCs may have downregulated CD19 expression via internalization rather than the cells being eliminated.

To further assess this interpretation, we quantified total PCs. Since young mice primarily possess CD19⁺ PCs, effective depletion would be expected to reduce total PC numbers. However, anti-CD19-treated mice did not show reduced total PC frequencies relative to controls (Fig. R7D). Similarly, B220⁺ B cell frequencies remained unchanged (Fig. R7E). Next, we confirmed that the absence of CD19 detection was not an artifact of antibody clone 1D3, as clone 6D5 yielded consistent results (Fig. R7F). Finally, we quantified immunoglobulin (Ig) levels in the sera of anti-CD19-treated mice before and after treatment and observed no reduction in any Ig isotypes (Fig. R7G). Altogether, these findings indicate that anti-CD19 antibody treatment did not effectively deplete B cells or CD19⁺ plasma cells.

These results are interesting nonetheless since similar phenomenon of CD19 loss has also been described in malignancies. Those reports and our findings strongly suggest that CD19-targeted therapies may be insufficient due to antigen modulation, and combination therapies are currently under investigation to mitigate this limitation⁴⁰⁻⁴².

In conclusion, while CD19-targeting antibodies do not deplete CD19⁻ cells, this finding is not directly relevant to our primary research question at this stage. Therefore, we have decided not to include these results in the revised manuscript.

Figure R7: anti-CD19 antibody treatment does not effectively deplete B cells or CD19+ plasma cells

Young and old Sle123 mice were intravenously injected with 250 µg of anti-CD19 antibody (clone 1D3, rat IgG2a, κ) or an isotype control (clone 1D10, rat IgG2a, κ), followed by anti-rat κ light chain (clone MAR 18.5, mouse IgG2a, κ). Mice were subsequently analyzed at days 7, 14, and 28 post-treatment.

(A) The representative FACS plots (left) and the graphs (right) show the frequency of CD19+ PCs in young and old Sle123 mice treated with anti-CD19 antibody or isotype antibody control.

(B) The representative FACS plots show the co-expression of CD19 versus B220 by PCs.

(C) The representative FACS plots (left) and the graphs (right) show the frequency of B220+ PCs in young and old Sle123 mice treated with anti-CD19 antibody or isotype antibody control.

(D) The representative FACS plots (left) and the graphs (right) show the frequency of total PCs in young and old Sle123 mice treated with anti-CD19 antibody or isotype antibody control

(E) The representative FACS plots (left) and the graphs (right) show the frequency of total B cells (B220+) in young and old Sle123 mice treated with anti-CD19 antibody or isotype antibody control

(F) The representative FACS plots show the co-staining of two anti-CD19 antibodies (clone 1D3 and 6D5) for B cells from Sle123 mice treated with anti-CD19 antibody or isotype antibody control.

(G) The graph shows the concentration of immunoglobulin in sera before and after treatment determined by LEGENDplex assay.

Data show compilation of two independent experiments (n=3 to 6 mice per group). Data show mean ± SEM.

The reviewer also raised the question whether (instead of targeting CD19+ cells) targeting CD19- PCs directly could be beneficial. We think that would be an interesting approach. Thus, we screened 255 surface molecules (LEGENDScreen™, BioLegend, Inc.) by flow cytometry to find surface molecules that were expressed by total CD19- PCs (Fig. 1E). Although we found differences between CD19+ and CD19- PCs (Fig. 1E), a pan-specific surface molecule to directly target all CD19- PCs could not be identified.

Instead of depleting the entire CD19⁻ population, a cluster-specific approach might therefore be more promising. Specifically, clusters 2 and 10, which exhibited low CD19 expression and were disease-specific, could represent particularly valuable targets for therapeutic interventions in SLE as we discussed in the Discussion. Implementing, possible therapeutic strategies is a focus of our future research program, while the focus of the present research is

to better delineate PCs clusters, their function and their respective relevance to disease. More systematic functional approaches of cluster depletion will be endeavored in future research.

6) CD19- PC is generally understood as a long-lived plasma cell subset. There is lack of reference throughout the paper to many previous studies of LLPC and how this CD19- subset fit into what is known about LLPC in the literature. More of this could be added into the discussion section.

We thank the reviewer for raising this important point. We have expanded the Discussion section accordingly, to include several recently published studies on long-lived plasma cells (LLPCs), and we discuss how our identified CD19- plasma cell subset aligns with existing knowledge about LLPCs. Furthermore, we provide a comparative analysis of LLPC phenotypes in lupus-prone mouse models, integrating findings from prior reports with our current data to contextualize the relevance and potential implications of the CD19- subset in autoimmune settings.

Minor concerns

1) There are a number of experiments that had only been done twice. It might be appropriate to do each experiment thrice to ensure reproducibility.

We have conducted additional experiments, and the majority have now been performed at least three times. For experiments, which were repeated twice, the results were clear, consistent, and reproducible across both independent replicates. Therefore, a third repetition was not conducted in these cases to adhere to animal welfare considerations, in line with the principles of the 3Rs for responsible animal use.

2) References are a different Font from the rest of the paper.

We apologize for the format issue. We have changed the Font accordingly.

Reviewer #3

The authors report a large comprehensive analysis of several plasma cell (PC) subsets in different mouse models. They based their interest on previous findings describing several subsets of PCs both in mice and humans, with different phenotypes, and functions. They argue that new therapies used in autoimmune diseases (systemic lupus) as CD19-directed CAR-T

cells when initiated could either miss some pathogenic CD19- PC or impair PCs with regulatory functions. In this view, they first study PCs in BM, spleen, tonsils and blood from human participants, mostly healthy (except for the spleens from patients with immune thrombocytopenia) and confirm the identification of CD19- PCs mostly in BM. Second, they study various lymphoid organs including bone marrow (BM) from C57BL/6 mice at different ages. They use genetic fate mapping systems to follow more precisely CD21+ cells and Ig gamma1 expressing cells. To differentiate B cells responding to T cell dependent and independent antigens, they use NP-KLH immunization. Considering their hypothesis in systemic lupus (SLE), they focus on Sle123 triple congenic mice, and study PCs in non-sick and sick mice, in lymphoid organs and kidney.

They found that CD19- BMPCs come from CD19+ cells and increase in numbers with age. These CD19- BMPCs in WT mice originate from diverse B cell subsets, including marginal zone, follicular B cells, B2 and B1-b cells. Moreover, CD19- PCs partly derive from germinal centers (GCs). Using an immunization model with NP-KLH, they demonstrate that CD19- PCs can be IgG+ NP-specific PCs.

Sick Sle123 mice develop more CD19- B cells than their non-sick or WT counterpart. They isolate BM and spleen PCs from sick and non-sick mice, and kidney PCs from sick mice to study heterogeneity of SLE PCs through single cell transcriptomic profiling. They conclude that sick mice are enriched with CD19- PC. Using a transfer model in Rag-/- mice, they conclude that CD19- PC can originate from CD19+ PCs from Sle123 mice.

Through their single cell transcriptomic profiling, CITE-seq, and screening of surface proteins, they differentiate clusters of PCs and based on previous studies, propose links with SLE pathogenesis.

They finally analysed the BCR repertoire in non-sick and sick mice, indicating overabundance or some clones in sick mice, with switch and high mutation rates in affected kidneys. Most clones show CD19 low expression. Looking to the most abundant clones, they identify VH11/Vk14 rearrangements previously associated to the recognition of phosphatidylcholine. Both B cells carrying these rearrangements and recognizing PC were more numerous in CD19- BM PCs in old mice.

Altogether, the main interesting points of this rather heavy analysis in lupus-prone mice are that some PCs can be CD19-, increasing in numbers with age, and more present in mice with lupus, both in lymphoid organs and in organs affected by the disease as kidney. This may explain the failure of targeting CD19 in some patients.

We appreciate the reviewer's positive feedback on our paper. We also thank the reviewer for the suggestions made below that have been important for us to improve our work.

Main points

-This is an important mice study both in wt and lupus-prone mice. It gives very interesting data on the different clusters of PCs and of their kinetics. However, concerning lupus, it remains a mice study and the link with human disease is not obvious. Clearly human data are missing, especially either longitudinal data whatever the authors write in their discussion line 7 ("thus our study allows a longitudinal approach that is not possible in humans") or data in patients treated with anti-CD19 therapies (Rituximab, CD19-CAR T cells). If this is not possible for the authors, then this is a very elegant mice study with still holes in what happens in patients.

We agree with the reviewer that more human data is an added value for this study. Thus, we have performed additional experiments and added human data on SLE patients on figure 3A in the revised manuscript. We analyzed peripheral blood of SLE patients vs healthy donors for CD19+ and CD19- PCs, and reported that SLE patients have more circulating CD19- PCs than healthy donors (Fig. 3A).

In addition, we analyzed peripheral blood and bone marrow plasma cells of NHL and ALL patients before and 90 days after CD19-directed CAR-T cell therapy. As expected, these analyses showed that BM CD19⁻ plasma cells are resistant to CD19-directed CAR-T therapy. Because such sequential BM sampling can only be obtained in the setting of a clinical trial, and the primary and secondary outcomes of the trial are not yet published, these data cannot be included in the manuscript. Nevertheless, they provide strong support for the translational significance of our findings and will be reported separately in due course.

This data helps to explain how protective antibodies (e.g., against tetanus, measles, mumps, rubella, varicella-zoster virus, and Epstein–Barr virus) remain stable, while autoantibodies are depleted in patients with autoimmune diseases^{43,44}. It also provides insight into why some patients retain certain autoantibodies after CD19-directed CAR-T therapy⁴³⁻⁴⁵.

Although the data have not yet been published, emerging evidence presented at EULAR 2024, ACR 2024, and EULAR 2025 suggests that, while CD19-directed CAR-T cell therapy shows promise for treating SLE, not all patients respond equally well. Specifically, some individuals fail to achieve sustained remission or relapse following initial clinical improvement. The resistance of CD19- PCs to CD19-directed CAR-T cell therapy helps explain these challenges and provides insight into mechanisms underlying incomplete therapeutic responses.

-The interest of phosphatidylcholine specific B cells in the context of SLE is unclear and should be more justified

We are interested in phosphatidylcholine (PtC)-specific PCs since these cells utilized VH11 gene segment and are derived from B-1 cells which have been shown to produce autoreactive antibodies and play a critical role in the pathogenesis of SLE^{13,23-26}. VH11 is a germ line gene that has been shown to encode antibodies that recognize dsDNA²⁷⁻³¹. In humans, PtC-specific antibodies have also been detected in the sera of individuals with autoimmune and infectious diseases³³⁻³⁵. Additionally, SLE patients with hemolytic anemia exhibit higher levels of anti-PtC IgM than those without this hematologic manifestation³⁷. Interestingly, soluble PtC has been shown to inhibit the IgM anti-dsDNA reactivity³⁸, indicating its polyreactive activity. Notably, patients with antiphospholipid syndrome (APS) show a significant increase in the frequency of PtC-specific B cells, which correlates with elevated serum levels of anti-PtC IgM and IgG as measured by ELISA³⁶.

Taken together, these findings support that the PtC-specific PCs are polyreactive and might be involved in autoimmune pathogenesis, including SLE. We have included a discussion of these points in the revised manuscript.

Minor comments

1) Considering that data on lupus come from Sle123 mice, the title should be tempered

We thank the reviewer for this valuable suggestion. In response to the reviewer's comments, we have performed additional experiments using human samples and have now incorporated new data from patients with SLE. We have also analyzed BM of patients who underwent CD19-CAR-T cell therapy. Although the later analysis cannot be included in the manuscript at this point (please see above), our observations further support the translational significance of our findings. Given these new and relevant findings, we believe that the original title remains appropriate and would prefer to retain it.

2) in Methods, in Mouse cell staining for flow cytometry analysis : it should be precise how NP specific PCs and PtC-specific PCs are detected as PCs express usually low levels of surface Ig (surface or intracellular staining ?)

We thank the reviewer for raising this important point. We concur that PCs typically express low levels of surface immunoglobulins (Igs), particularly IgG-secreting PCs. In our study, NP-

and PtC-specific PCs were identified using phycoerythrin (PE)-conjugated NP and PtC-containing liposomes labeled with Texas Red, respectively, via surface binding. This approach was necessary because NP conjugates most effectively with PE. However, intracellular staining of PCs with NP-PE indiscriminately labeled all PCs, including those from unimmunized mice, as previously noted by others^{46,47}. Surface binding of NP-PE proved effective for identifying both IgM⁺ and IgM⁻ PCs following NP-KLH immunization, whereas control mice injected with PBS did not exhibit NP-specific PCs (Fig. 2F of the revised manuscript). This observation is consistent with findings reported by others¹⁸ and was also by our experimental results.

Additionally, the λ light chain can reliably identify NP-specific PCs, as λ light chain is the predominant Ig light chain isotype associated with anti-NP responses in C57BL/6 mice, as established in previous studies¹⁶⁻¹⁹. We also revealed that the majority of NP-specific PCs were Ig λ ⁺, whereas only 6–7% of non-NP-specific PCs expressed Ig λ (Supplementary Fig. 2E of the revised manuscript) via surface binding. We have included these new results in Supplementary Fig. 2E of the revised manuscript. Also, to validate the NP-specific PC population identified by flow cytometry, we complemented this approach with ELISPOT assays using sorted CD19⁺ and CD19⁻ bone marrow PCs from NP-KLH-immunized mice. This analysis confirmed the flow cytometry findings, quantifying NP-specific PCs (Fig. 2I of the revised manuscript).

Similarly, PtC surface binding proved to be a reliable method, as it binds BCRs encoded by the VH11/Vk14 gene segment⁴⁸ which can be detected by surface flow cytometry in both this and an earlier report¹³. Notably, all VH11⁺/Vk14⁺ PCs identified via surface staining exclusively represented PtC-specific cells via surface binding (Fig. R8). Moreover, the surface staining of VH11⁺/Vk14⁺ BCRs on PCs aligned with independently performed single-cell RNA sequencing data (Fig. 8A, B, E, and F of the revised manuscript). Additionally, the staining patterns for PtC and VH11/Vk14 were comparable between PCs and B-1a cells, further supporting the reliability of this approach.

We have clarified the methods used to detect NP- and PtC-specific PCs in the revised manuscript to address the reviewer's concerns.

Figure R8: Surface staining of VH11Vk14 and PtC by flow cytometry

The flow cytometry plots show the surface staining of VH11VK14 and PtC. The graph shows the frequency of PtC-specific PCs in VH11+Vk14+ PCs. Data were from two independent experiments (n=6 mice).

3) Figure 1C: IgG Elispot is moderately convincing (lack a positive control to add to supplementary data), also compared to Figure 2I (NP+ IgG Elispot on CD19-)

We thank the reviewer for raising this point. As detailed in the Methods section of the originally submitted manuscript, sorted CD19⁺ and CD19⁻ PCs were plated at an initial concentration of 1.5×10^3 cells per 200 μ L per well, followed by four successive five-fold serial dilutions. The ELISPOT images presented in Fig. 1C correspond to the dilution at which the numbers of IgA-, IgM-, and IgG-secreting cells were countable and reflective of their representative frequencies within the sorted PC subpopulations. Since IgG-secreting cells comprise approximately 10–20% of both CD19⁺ and CD19⁻ PCs, the number of IgG-specific spots was less prominent compared to IgA and IgM. In response to the reviewer's comment, we have revised Fig. 1C in the updated manuscript to improve its clarity and representation (Fig. R9).

Figure R9: IgA-, IgM- and IgG-secreting cells in sorted CD19– BMPCs by ELISPOT assay

The ELISPOT pictures show IgA-, IgM- and IgG-secreting cells in sorted CD19- BMPCs. Results are representative of five independent experiments (n=7-15 mice/experiment).

4) Figure 1E : precise the explanation for the lower expression of B220 on CD19- B cells

We thank the reviewer for this correction. In response, we have revised the manuscript to clarify the lower expression levels of both B220 and MHC-II on CD19- PCs. These adjustments are reflected in the revised version.

5) As it is shown that CD19- B cells have lower CD19 RNA and a lower level of CD19 protein, instead of a surface downregulation/internalisation of CD19, the authors should develop the reasons of such a decreased transcription

We appreciate the reviewer's insightful observation regarding the decreased transcription of CD19 in CD19-negative PCs. To address this further and show that CD19 is regulated on transcriptional level rather than internalisation alone, we evaluated the expression levels of a set of activators and repressors of CD19 expression, as identified in a recent publication⁴⁹, alongside we plotted CD19 RNA and protein levels (Fig. R10).

These observations are consistent with NUDT21's role as a dynamic brake on CD19 expression:

In line with previous data showing⁴⁹ that loss of NUDT21 shortens the CD19 3'UTR, stabilizes its mRNA and elevates surface CD19, we find that NUDT21 levels are lowest in cells with maximal CD19 mRNA and protein, peak in actively proliferating PBs, and then decline again in CD19^{low} cells, suggesting that modulation of NUDT21 expression during B-cell proliferation tightly tunes CD19 output.

The co-expression of both CD19 activators within the same clusters, particularly in CD19^{high} cells (cluster 6, 9 and 7), likely reflects a tightly regulated feedback mechanism to fine-tune CD19 levels. This phenomenon suggests that activators and repressors act together to dynamically balance CD19 expression.

Interestingly in CD19^{low} clusters (cluster 0 and 10), the absence of expression of both activators and repressors likely reflects a distinct cellular identity or developmental stage where CD19 is not required or functionally relevant. Alternatively, these clusters might represent a transcriptionally silenced state for CD19 regulation, where neither activation nor repression pathways are actively engaged. This differential expression pattern underscores the specificity of CD19 regulation to particular cell states and suggests that the coordinated presence of activators and repressors is a hallmark of CD19-positive cells to achieve precise expression control.

We added this result in Fig. 4C of the revised manuscript and pointed out that instead of internalisation, the downregulation and differential expression of CD19 across clusters are regulated on transcriptional levels.

Scores for the genset of activators and repressors were added to show an overall trend.

Figure R10: Expression of selected activator and repressor genes associated with CD19 regulation

Bubble plot shows the expression of *Cd19* mRNA (first column) and *CD19* protein (second column), as well as of selected activator and repressor genes associated with *CD19* regulation as published⁴⁹

across the identified clusters. Bubble sizes show the percentage of cells within each cluster expressing the given transcript or protein, and the color scale represents the mean expression level. The two right columns display composite activator and repressor scores for each cluster. Each module score reflects the average scaled expression of the corresponding gene set (activators or repressors) in individual cells.

6) A general remark is that the manuscript is hard to follow, especially the description of the different PCs clusters

We rewrote several parts of the manuscript to increase its readability. We restructured and rewrote the descriptions of the PC clusters completely dedicating one paragraph to each cluster. We think this part has substantially improved. Thus, we would like to thank the reviewer for this suggestion.

Overall, we would like to express our appreciation to the editor, the editorial team, and all the reviewers for their support and valuable suggestions. We believe that we have effectively addressed the previous concerns raised in the manuscript and look forward to your feedback.

Sincerely,

Andreia C. Lino, Ph.D

References

1. Lacotte, S., *et al.* Early differentiated CD138(high) MHCII+ IgG+ plasma cells express CXCR3 and localize into inflamed kidneys of lupus mice. *PLoS One* **8**, e58140 (2013).
2. Rosean, T.R., Loo, W. & Erickson, L.D. Identification of novel plasma cell subsets in a mouse model of SLE. *The Journal of Immunology* **198**, 54.15–54.15 (2017).
3. Serre, K., *et al.* CD8 T cells induce T-bet-dependent migration toward CXCR3 ligands by differentiated B cells produced during responses to alum-protein vaccines. *Blood* **120**, 4552–4559 (2012).
4. Tsubaki, T., *et al.* Accumulation of plasma cells expressing CXCR3 in the synovial sublining regions of early rheumatoid arthritis in association with production of Mig/CXCL9 by synovial fibroblasts. *Clin Exp Immunol* **141**, 363–371 (2005).
5. Henneken, M., Dorner, T., Burmester, G.R. & Berek, C. Differential expression of chemokine receptors on peripheral blood B cells from patients with rheumatoid arthritis and systemic lupus erythematosus. *Arthritis Res Ther* **7**, R1001–1013 (2005).
6. Reijm, S., *et al.* Autoreactive B cells in rheumatoid arthritis include mainly activated CXCR3+ memory B cells and plasmablasts. *JCI Insight* **8**(2023).
7. Hosomi, S., *et al.* Increased numbers of immature plasma cells in peripheral blood specifically overexpress chemokine receptor CXCR3 and CXCR4 in patients with ulcerative colitis. *Clin Exp Immunol* **163**, 215–224 (2011).
8. Tellier, J., *et al.* Blimp-1 controls plasma cell function through the regulation of immunoglobulin secretion and the unfolded protein response. *Nat Immunol* **17**, 323–330 (2016).
9. Wilmore, J.R., Jones, D.D. & Allman, D. Protocol for improved resolution of plasma cell subpopulations by flow cytometry. *Eur J Immunol* **47**, 1386–1388 (2017).
10. Koike, T., *et al.* Progressive differentiation toward the long-lived plasma cell compartment in the bone marrow. *J Exp Med* **220**(2023).
11. Tipton, C.M., *et al.* Diversity, cellular origin and autoreactivity of antibody-secreting cell population expansions in acute systemic lupus erythematosus. *Nat Immunol* **16**, 755–765 (2015).
12. Jenks, S.A., *et al.* Distinct Effector B Cells Induced by Unregulated Toll-like Receptor 7 Contribute to Pathogenic Responses in Systemic Lupus Erythematosus. *Immunity* **49**, 725–739 e726 (2018).
13. Lino, A.C., *et al.* LAG-3 Inhibitory Receptor Expression Identifies Immunosuppressive Natural Regulatory Plasma Cells. *Immunity* **49**, 120–133 e129 (2018).

14. Tung, J.W. & Herzenberg, L.A. Unraveling B-1 progenitors. *Curr Opin Immunol* **19**, 150–155 (2007).
15. Saito, T., *et al.* Notch2 is preferentially expressed in mature B cells and indispensable for marginal zone B lineage development. *Immunity* **18**, 675–685 (2003).
16. Allen, D., *et al.* Timing, genetic requirements and functional consequences of somatic hypermutation during B-cell development. *Immunol Rev* **96**, 5–22 (1987).
17. Jacob, J. & Kelsoe, G. In situ studies of the primary immune response to (4-hydroxy-3-nitrophenyl)acetyl. II. A common clonal origin for periarteriolar lymphoid sheath-associated foci and germinal centers. *J Exp Med* **176**, 679–687 (1992).
18. Blanc, P., *et al.* Mature IgM-expressing plasma cells sense antigen and develop competence for cytokine production upon antigenic challenge. *Nat Commun* **7**, 13600 (2016).
19. Robinson, M.J., *et al.* Long-lived plasma cells accumulate in the bone marrow at a constant rate from early in an immune response. *Sci Immunol* **7**, eabm8389 (2022).
20. Bradfute, S.B., Graubert, T.A. & Goodell, M.A. Roles of Sca-1 in hematopoietic stem/progenitor cell function. *Exp Hematol* **33**, 836–843 (2005).
21. Morcos, M.N.F., *et al.* SCA-1 Expression Level Identifies Quiescent Hematopoietic Stem and Progenitor Cells. *Stem Cell Reports* **8**, 1472–1478 (2017).
22. Dang, V.D., *et al.* CD39 and CD326 Are Bona Fide Markers of Murine and Human Plasma Cells and Identify a Bone Marrow Specific Plasma Cell Subpopulation in Lupus. *Front Immunol* **13**, 873217 (2022).
23. Zhong, X., *et al.* A novel subpopulation of B-1 cells is enriched with autoreactivity in normal and lupus-prone mice. *Arthritis Rheum* **60**, 3734–3743 (2009).
24. Ottens, K., Schneider, J. & Satterthwaite, A.B. B-1a Cells, but Not Marginal Zone B Cells, Are Implicated in the Accumulation of Autoreactive Plasma Cells in Lyn^{-/-} Mice. *Immunohorizons* **8**, 47–56 (2024).
25. Ma, K., *et al.* B1-cell-produced anti-phosphatidylserine antibodies contribute to lupus nephritis development via TLR-mediated Syk activation. *Cell Mol Immunol* **20**, 881–894 (2023).
26. Gronwall, C., Vas, J. & Silverman, G.J. Protective Roles of Natural IgM Antibodies. *Front Immunol* **3**, 66 (2012).
27. Pewzner-Jung, Y., Simon, T. & Eilat, D. Structural elements controlling anti-DNA antibody affinity and their relationship to anti-phosphorylcholine activity. *J Immunol* **156**, 3065–3073 (1996).

28. Behar, S.M. & Scharff, M.D. Somatic diversification of the S107 (T15) VH11 germ-line gene that encodes the heavy-chain variable region of antibodies to double-stranded DNA in (NZB x NZW)F1 mice. *Proc Natl Acad Sci U S A* **85**, 3970–3974 (1988).
29. Behar, S.M., Lustgarten, D.L., Corbet, S. & Scharff, M.D. Characterization of somatically mutated S107 VH11-encoded anti-DNA autoantibodies derived from autoimmune (NZB x NZW)F1 mice. *J Exp Med* **173**, 731–741 (1991).
30. Friedmann, D., *et al.* Production of high affinity autoantibodies in autoimmune New Zealand Black/New Zealand white F1 mice targeted with an anti-DNA heavy chain. *J Immunol* **162**, 4406–4416 (1999).
31. Pewzner-Jung, Y., *et al.* B cell deletion, anergy, and receptor editing in "knock in" mice targeted with a germline-encoded or somatically mutated anti-DNA heavy chain. *J Immunol* **161**, 4634–4645 (1998).
32. Tiller, T., *et al.* Development of self-reactive germinal center B cells and plasma cells in autoimmune Fc gammaRIIB-deficient mice. *J Exp Med* **207**, 2767–2778 (2010).
33. Cabral, A.R., Cabiedes, J. & Alarcon-Segovia, D. Hemolytic anemia related to an IgM autoantibody to phosphatidylcholine that binds in vitro to stored and to bromelain-treated human erythrocytes. *J Autoimmun* **3**, 773–787 (1990).
34. Abuaf, N., *et al.* Autoantibodies to phospholipids and to the coagulation proteins in AIDS. *Thromb Haemost* **77**, 856–861 (1997).
35. Casao, M.A., Leiva, J., Diaz, R. & Gamazo, C. Anti-phosphatidylcholine antibodies in patients with brucellosis. *J Med Microbiol* **47**, 49–54 (1998).
36. Nitschke, E., *et al.* Phosphatidylcholine-specific B cells are enriched among atypical CD11c(high) and CD21(low) memory B cells in antiphospholipid syndrome. *Front Immunol* **16**, 1585953 (2025).
37. Guzman, J., Cabral, A.R., Cabiedes, J., Pita-Ramirez, L. & Alarcon-Segovia, D. Antiphospholipid antibodies in patients with idiopathic autoimmune haemolytic anemia. *Autoimmunity* **18**, 51–56 (1994).
38. Cabiedes, J., *et al.* Characterization of anti-phosphatidylcholine polyreactive natural autoantibodies from normal human subjects. *J Autoimmun* **18**, 181–190 (2002).
39. Keren, Z., *et al.* B-cell depletion reactivates B lymphopoiesis in the BM and rejuvenates the B lineage in aging. *Blood* **117**, 3104–3112 (2011).
40. Spiegel, J.Y., *et al.* CAR T cells with dual targeting of CD19 and CD22 in adult patients with recurrent or refractory B cell malignancies: a phase 1 trial. *Nat Med* **27**, 1419–1431 (2021).

41. Cordoba, S., *et al.* CAR T cells with dual targeting of CD19 and CD22 in pediatric and young adult patients with relapsed or refractory B cell acute lymphoblastic leukemia: a phase 1 trial. *Nat Med* **27**, 1797–1805 (2021).
42. Koh, S.K., *et al.* Anti-CD19 antibody cotreatment enhances serial killing activity of anti-CD19 CAR-T/-NK cells and reduces trogocytosis. *Blood* **145**, 956–969 (2025).
43. Mackensen, A., *et al.* Anti-CD19 CAR T cell therapy for refractory systemic lupus erythematosus. *Nat Med* **28**, 2124–2132 (2022).
44. Muller, F., *et al.* CD19 CAR T-Cell Therapy in Autoimmune Disease - A Case Series with Follow-up. *N Engl J Med* **390**, 687–700 (2024).
45. Nunez, D., *et al.* Cytokine and reactivity profiles in SLE patients following anti-CD19 CART therapy. *Mol Ther Methods Clin Dev* **31**, 101104 (2023).
46. Kim, M.S. & Kim, T.S. R-phycoerythrin-conjugated antibodies are inappropriate for intracellular staining of murine plasma cells. *Cytometry A* **83**, 452–460 (2013).
47. Renner, P., Crone, M., Kornas, M., Pioli, K.T. & Pioli, P.D. Intracellular flow cytometry staining of antibody-secreting cells using phycoerythrin-conjugated antibodies: pitfalls and solutions. *Antib Ther* **5**, 151–163 (2022).
48. Hardy, R.R., Wei, C.J. & Hayakawa, K. Selection during development of VH11+ B cells: a model for natural autoantibody-producing CD5+ B cells. *Immunol Rev* **197**, 60–74 (2004).
49. Witkowski, M.T., *et al.* NUDT21 limits CD19 levels through alternative mRNA polyadenylation in B cell acute lymphoblastic leukemia. *Nat Immunol* **23**, 1424–1432 (2022).

Point-by-point reply:

Reviewer #1 (Remarks to the Author):

In the revised manuscript, the authors appropriately addressed the concerns raised by this reviewer.

We appreciate the Reviewer's positive feedback on the revised manuscript.

Reviewer #2 (Remarks to the Author):

The authors have addressed all my major concerns in a satisfactory manner except for the *in vivo* CD19 depletion experiment to support their hypothesis that altering the timing of anti-CD19 therapy could lead to better disease outcome if their further differentiation into CD19- PC could be prevented.

We are very grateful to the reviewer for their careful reading and positive assessment of the revised manuscript and for acknowledging the value of the new human patient data (Figure 3A) and the extensive plasma cell characterization.

The attempt to perform *in vivo* B cell depletion is credible but the authors had been unsuccessful in depleting B cells with anti-mouse CD19 (1D3) alone and thus unable to address this question because of technical issues. In literature, 1D3 had been shown to effectively deplete B cells in mice in combination with other B cell targeting antibodies (Keren, Z., et al. (2011). B-cell depletion reactivates B lymphopoiesis in the BM and rejuvenates the B lineage in aging'' Blood 117(11): 3104-3112). It would have been ideal if the authors could prove their hypothesis by performing optimal CD19 depletion *in vivo* with reference to prior literature. I understand that this is not a straightforward experiment and would require optimization.

We agree that the complete *in vivo* depletion of the CD19- PC subset would be the gold standard to fully test our hypothesis regarding therapeutic timing. However, we would like to politely clarify the interpretation of the results from our anti-CD19 treatment experiment (detailed in our previous response, Fig. R7).

Our initial experiment focused specifically on the anti-CD19 monoclonal antibody (clone 1D3) alone because our central hypothesis revolved around the efficacy of CD19-targeted therapy against early CD19+ PCs versus late CD19- PCs, mirroring the use of single-target therapies in a clinical context.

As detailed in our prior response, the lack of effective depletion (no reduction in total PCs, B220+ PCs, or serum Ig levels) was not merely a non-optimized depletion method. Instead, the rapid and transient loss of surface CD19 (with preserved B220 expression and total cell numbers) strongly suggested antigen modulation/internalization as the mechanism of therapeutic failure. This finding, while unexpected, is highly interesting and aligns with observations of CD19 loss in certain malignancies, reinforcing the challenge of using anti-CD19 alone.

The reviewer correctly refers to the work by Keren et al. (2011), where effective B cell depletion was achieved using clone 1D3. We respectfully point out that Keren and colleagues utilized a combination therapy consisting of three antibodies (anti-CD19, anti-B220, and anti-CD22) to achieve complete B cell depletion in that study. This approach, while highly effective

for global B cell ablation, differs fundamentally from our initial, specific test of a monotherapy targeting only CD19. Although most B220⁺ PCs are CD19⁺, approximately 20% are CD19⁻. Thus, adding anti-B220 could inadvertently deplete a fraction of CD19⁻ PCs, complicating downstream interpretation.

So here, to make it comparable to currently used anti-CD19 treatments we focused on the single CD19 targeting approach. Using the suggested multi-target approach, although a very interesting idea, would have complicated the interpretation of our scientific question, since with that approach we would not be able to clearly discriminate between the effect of targeting CD19 expressing B cells and PC or if co-targeting B220⁺CD19⁻ PCs would influence the outcome.

We are currently working on optimizing a murine CAR-T cell or multi-target approach to achieve comprehensive B cell and CD19⁺ PC depletion without directly targeting any CD19⁻ PCs. However, given the complexity and time required for fulfilling the legal requirements for those experiments, we do not think this could be achieved in a reasonable timeframe during the current revision. Overall, we think the results are interesting and generate new scientific questions we would like to address in future projects.

Nonetheless, the addition of CD19⁺ and CD19⁻ PC characterization from human SLE patients into Figure 3A and the extensive characterization of PC in lupus mice may justify publication.

We agree with the reviewer that the robust characterization of the CD19-PC subset in both human SLE patients and the Sle123 model, which is the core finding of the manuscript, already justifies publication at this stage.

Reviewer #3 (Remarks to the Author):

I read the revised manuscript by Van Duc Dang et al., on CD19⁻ plasmacytes in Lupus, and I thank the authors for their revising effort.

We would like to thank reviewer #3 for appreciating our efforts. Please find below a point-by-point reply to the remaining comments.

My main remarks were: the fact that it is essentially a murine analysis. In the revised form, the authors performed a quantification of CD19⁻ PCs, and determined that SLE patients do have more circulating CD19⁻ PCs. They analysed CD19⁻ PCs also in patients with haematological malignancies and treated with anti-CD19 CAR T, to find that CD19⁻ B cells were resistant to anti-CD19 CAR T although they do not show the data. They write « This data help to explain how protective antibodies ... remain stable, while autoantibodies are depleted in patients with autoimmune diseases » : I must say that I do not understand completely the link and the sentence in the context of SLE. The idea is better developed in the discussion, and suggest that PC producing antigen specific antibodies are CD19⁻ PC. The authors suggest that early intervention should be relevant. However, concerning B cells, they produce autoantibodies a long time ago before the development of the disease (see Arbuckle MR, N Engl J Med, 2003).

We thank the reviewer for their comments on the revised manuscript and for acknowledging our efforts to incorporate human data.

We understand the reviewer's curiosity regarding the human BM data from patients treated with anti-CD19 CAR T cells. The data showing the persistence of CD19⁻ LLPCs in the BM of these patients were obtained in the context of a clinical trial. While highly relevant, these data

could not be included in the revised manuscript as external clinical trial results were editorially excluded. However, as correctly surmised by the reviewer, our findings indicate that patients treated with anti-CD19 CAR T cells retained CD19- LLPCs in the BM, while the CD19+ compartment was effectively depleted. This result confirms our central finding that once the plasma cell phenotype shifts to CD19-, they become resistant to CD19-targeted therapies. This is particularly relevant given that anti-CD19 CAR T cell therapy is being explored as an end-of-line treatment for refractory lupus, suggesting that earlier intervention may be necessary for better efficacy. We are actively working towards the publication of these complete clinical findings as soon as possible.

We acknowledge the reviewer's comment regarding the sentence on protective antibodies and thank the reviewer for pointing out the need for greater clarity. The intended meaning is that LLPCs are critical for both protective immunity and sustained autoimmunity. We agree that autoantibodies can be detected years before clinical symptoms of SLE develop, as demonstrated by Arbuckle et al. (2003) and others (e.g. Tan EM et al. *Arthritis Rheum.* 1997;40(9):1601-11). We interpret this observation to mean that the initial, low-level autoantibody production occurring years before disease onset is likely derived from short-lived plasma cells or early-stage B cells that are not yet established in the as LLPC. These initial autoantibodies may not be sufficient to drive sustained, organ-damaging pathology. Our hypothesis is that the transition to clinically relevant, high-titer autoantibody production, which characterizes active SLE, requires the establishment of a stable pool of pathogenic, autoantibody-producing CD19- LLPCs. Therefore, our suggestion for early therapeutic intervention aims to target the CD19+ PC and B cell before they can differentiate, migrate, and settle into the long-lived, drug-resistant CD19- pool. Preventing the establishment of this pathogenic CD19- memory pool is precisely why early intervention could lead to better long-term disease outcome.

the specific place of phosphatidylcholine (PtC) autoAbs in the context of SLE. Although convincing, the added references from 27 to 31, concern mice models.

We apologize for any misunderstanding that may have occurred in the previous revision. In our prior point-by-point response, we addressed the role of PtC autoantibodies in humans (References 33–38) and incorporated this discussion into the revised manuscript (References 87–92).

Additionally, in August 2025, an independent group led by Carola G. Vinuesa (Francis Crick Institute, London, UK) reported in *Nature* the identification of PtC-specific B cells in humans.

To ensure clarity, we have included our discussion on PtC autoantibodies in humans below, now updated to incorporate this most recent publication:

“In humans, PtC-specific antibodies have also been detected in the sera of individuals with autoimmune or infectious diseases^{1, 2, 3}. PtC-specific B cells were also recently identified in humans⁴. Notably, patients with antiphospholipid syndrome (APS) show a significant increase in the frequency of PtC-specific B cells, which correlates with elevated serum levels of anti-PtC IgM and IgG as measured by ELISA⁵. Additionally, SLE patients with hemolytic anemia exhibit higher levels of anti-PtC IgM than those without this hematologic manifestation⁶. Interestingly, soluble PtC has been shown to inhibit the IgM anti-dsDNA reactivity⁷, indicating its polyreactive activity, particularly for these two specificities”.

Concerning the other points: I maintain that the title suggests Lupus in general (including human) and that there is no demonstration of autoreactive CD19- PC in human lupus in this manuscript so that the term is a bit large or confusing. The authors extend their results but only to demonstrate the presence of CD19- PC in patients, not their autoreactive features. I understand that the present title is more eye catching.

We understand the reviewer's point; thus, we changed the title to *Distinct Autoreactive CD19- Plasma Cell Subsets Accumulate in Lupus-prone mice*.

The other points are addressed. I found very interesting the new set of experiments done in Fig4 to explore CD19 regulation in the different clusters, and the conclusion that CD19 downregulation is transcriptionally regulated. A real effort has been made to improve the writing and the understanding of the manuscript

We thank the reviewer for the very insightful comments provided in the previous round of review. We greatly appreciate the recognition of our efforts to address the raised points, as well as the positive feedback on the new experiments in Figure 4 that CD19 regulation is transcriptionally regulated. We also sincerely appreciate the reviewer's recognition of the improvements made in both the writing and the overall clarity of the manuscript.

New points:

precise diagnosis of SLE patients (active, quiescent, systemic ?) in Method

We now added the Systemic Lupus Erythematosus Disease Activity Index (SLEDAI) and clinical SLEDAI (cSLEDAI) to the table with the patient information which can be found in the Supplementary Information file (Supplementary Table1). We also grouped the patients based on the SLEDAI as follows: active, SLEDAI ≥ 6 ; mild, SLEDAI 1–5 and quiescent, SLEDAI 0. Most of our patients fall into the mild to active category. Our cohort therefore represents the wide range of different disease activity stages found in patients. We were not aiming at a specific subgroup of patients within the heterogenic group of SLE patients. We hope this is the information reviewer 3 was asking for. In alignment with the editorial request, we updated the table and are now reporting the average age instead of the individual age.

In the new Figure 4, re-precise in the legend the CD19- clusters to facilitate understanding

We thank the reviewer for this request to facilitate better understanding for the readers. We have introduced the requested changes by adding the following to the figure legend:

“(A) [...] Based on CD19 RNA and protein expression, Clusters 4, 6, 7, 8, 9, 12 and 13 were identified as CD19^{hi} clusters while Clusters 0, 1, 2, 3, 5, 10 and 11 expressed low amounts of CD19 and were therefore identified as CD19^{lo} clusters. (B) Pie charts show the frequency of CD19^{hi} (blue; representing concatenation of Clusters 4, 6, 7, 8, 9, 12 and 13) and CD19^{lo} (orange; representing concatenation of Clusters 0, 1, 2, 3, 5, 10 and 11) PCs in indicated organs from healthy and sick Sle123 mice quantified by scRNA seq.

Overall, we would like to express our appreciation to the editor, the editorial team, and all the reviewers for their support and valuable suggestions.

Sincerely,

Andreia C. Lino, Ph.D

References

1. Cabral AR, Cabiedes J, Alarcon-Segovia D. Hemolytic anemia related to an IgM autoantibody to phosphatidylcholine that binds in vitro to stored and to bromelain-treated human erythrocytes. *J Autoimmun* **3**, 773–787 (1990).
2. Abuaf N, *et al.* Autoantibodies to phospholipids and to the coagulation proteins in AIDS. *Thromb Haemost* **77**, 856–861 (1997).
3. Casao MA, Leiva J, Diaz R, Gamazo C. Anti-phosphatidylcholine antibodies in patients with brucellosis. *J Med Microbiol* **47**, 49–54 (1998).
4. Shen Q, *et al.* TCF1 and LEF1 promote B-1a cell homeostasis and regulatory function. *Nature*, (2025).
5. Nitschke E, *et al.* Phosphatidylcholine-specific B cells are enriched among atypical CD11c(high) and CD21(low) memory B cells in antiphospholipid syndrome. *Front Immunol* **16**, 1585953 (2025).
6. Guzman J, Cabral AR, Cabiedes J, Pita-Ramirez L, Alarcon-Segovia D. Antiphospholipid antibodies in patients with idiopathic autoimmune haemolytic anemia. *Autoimmunity* **18**, 51–56 (1994).
7. Cabiedes J, Cabral AR, Lopez-Mendoza AT, Cordero-Esperon HA, Huerta MT, Alarcon-Segovia D. Characterization of anti-phosphatidylcholine polyreactive natural autoantibodies from normal human subjects. *J Autoimmun* **18**, 181–190 (2002).